# Boosting the stability of perovskites with exsolved nanoparticles by B-site supplement mechanism

Bo-Wen Zhang [1,5], Meng-Nan Zhu [1,5], Min-Rui Gao[1], Xiuan Xi[2], Nanqi Duan[1], Zhou Chen[3], Ren-Fei Feng [4], Hongbo Zeng [1] & Jing-Li Luo [1,2] ✉

Perovskites with exsolved nanoparticles (P-eNs) have immense potentials for carbon dioxide ($CO_2$) reduction in solid oxide electrolysis cell. Despite the recent achievements in promoting the B-site cation exsolution for enhanced catalytic activities, the unsatisfactory stability of P-eNs at high voltages greatly impedes their practical applications and this issue has not been elucidated. In this study, we reveal that the formation of B-site vacancies in perovskite scaffold is the major contributor to the degradation of P-eNs; we then address this issue by fine-regulating the B-site supplement of the reduced $Sr_2Fe_{1.3}Ni_{0.2}Mo_{0.5}O_{6-\delta}$ using foreign Fe sources, achieving a robust perovskite scaffold and prolonged stability performance. Furthermore, the degradation mechanism from the perspective of structure stability of perovskite has also been proposed to understand the origins of performance deterioration. The B-site supplement endows P-eNs with the capability to become appealing electrocatalysts for $CO_2$ reduction and more broadly, for other energy storage and conversion systems.

Perovskites are attractive candidates for high-temperature solid oxide electrolysis/fuel cell (SOEC/SOFC)[1–4]. Benefiting from the excellent catalytic activities of well-exposed metallic nanoparticles, alongside the unique nanoparticle-perovskite interface with high resistance to agglomeration, perovskites with exsolved nanoparticles (P-eNs) provide an appealing platform for the large-scale energy storage and conversion technique compared to their nanoparticle-free counterparts[5–9]. Over the past decades, there have been significant advances in promoting exsolution via additional driving forces[10–12]. However, challenges remain concerning the stability of P-eNs, especially when being used as a cathode for $CO_2$ electrolysis in SOEC[13,14]. Although the thermal stability of nanoparticles has been literally enhanced by the exsolution process, the rapid degradation issue still exists at high voltages, which results in a lower energy efficiency[15–17]. Since steady high-voltage $CO_2$ electrolysis can lead to higher CO yields[18], new strategies are urgently required to enhance the stability of P-eNs at high voltages while promoting nanoparticle exsolution.

To tackle this challenge, we started by examining the fundamental process of nanoparticle exsolution on the perovskite[10], as shown in Eq. 1.

$$ABO_{3-\delta} \xrightarrow{\text{Exsolution}} A_{1-\alpha'}B_{1-\alpha}O_{3-\delta'} + \alpha B + \alpha' AO \qquad (1)$$

It is noteworthy that the exsolution of B-site reducible cations inevitably leaves behind many B-site vacancies within the perovskite bulk, thus causing the detrimental A-site segregation and the slow-down of the electron transfer rate via the $B^{(n-1)+}$-O-$B^{n+}$ pathway[19–21]. The external potentials would drive the continuous exsolution during $CO_2$ electrolysis, especially at higher voltages, which could potentially hamper the stability of P-eNs[22,23]. However, the B-site vacancy-dictated stability issues are often veiled by the reactivity enhancement brought

[1]Department of Chemical and Materials Engineering, University of Alberta, Edmonton, AB T6G 1H9, Canada. [2]College of Materials Science and Engineering, Shenzhen University, Shenzhen 518060, P.R. China. [3]College of Materials, Xiamen University, Xiamen 361005, P.R. China. [4]Canadian Light Source Inc., Saskatoon, SK S7N 0X4, Canada. [5]These authors contributed equally: Bo-Wen Zhang, Meng-Nan Zhu. ✉e-mail: jingli.luo@ualberta.ca

by concurrent B-site exsolution. Very few studies have attempted to correlate the perovskite structure evolution dominated by B-site vacancy to the stability of P-eNs.

In this study, the promising double perovskite $Sr_2Fe_{1.3}Ni_{0.2}Mo_{0.5}O_{6-\delta}$ (SFNM) was selected as a prototype example to elaborate on the effects of the structure evolution of the perovskite scaffold on the stability of P-eNs[15]. Either controlling the A-site deficiency or implementing the topotactic ion exchange (TIE) is expected to be a pathway to regulate the concentration of the B-site vacancies in the reduced SFNM (Eqs. 2 and 3)[10,24–26]. However, the limited Sr-site deficiency (<5% mol) in the SFNM makes it fail to refill the B-site vacancies after the exsolution by controlling the A-site deficiency[27]. Therefore, the TIE-assisted exsolution was employed to fine-tune the B-site occupation of perovskite scaffold while promoting the formation of nanoparticles.

$$A_{1-\alpha}BO_{3-\delta} \xrightarrow{\text{Exsolution}} (1-\alpha)ABO_{3-\delta'} + \alpha B \qquad (2)$$

$$ABO_{3-\delta} + \alpha B_{\text{guest}} \xrightarrow{\text{Exsolution}} AB_{1-\alpha}B_{\text{guest}_\alpha}O_{3-\delta'} + \alpha B \qquad (3)$$

To initiate the TIE-assisted exsolution, the foreign Fe ion is a good choice as the B-site filling agent due to its relatively high redox stability[28]. Our density functional theory (DFT) results demonstrated the feasibility of swapping guest Fe with host Ni in SFNM driven by the difference in co-segregation energy. Therefore, a uniform guest Fe

overlayer was introduced on the surface of SFNM scaffold by a facile lyophilization method, followed by the supplement of Fe into B-site vacancies in the reduced SFNM bulk. Consequently, a robust $Sr_2Fe_{1.5}Mo_{0.5}O_{6-\delta}$ (SFM) scaffold with decreased B-site vacancies and Ni incorporation has been implemented, accompanied by the exsolution of FeNi alloy nanoparticles[29]. The extensive electrochemical and structural characterizations confirm that the incorporation of the guest Fe into B-site vacancies of the reduced SFNM scaffold delivers higher catalytic activity and stability of the reduced perovskite scaffold, especially when external voltages ≥1.6 V. Furthermore, from the short/long-term stability performances and post characterizations, we have uncovered the degradation mechanism of the reduced SFNM with/without B-site supplement at high potentials from the perspective of structural stability. These results highlight the indispensable contribution of structurally robust mother perovskite to the higher stability of P-eNs for $CO_2$ reduction in SOEC, and also provide extensive prospects towards the rational design of advanced heterogeneous P-eNs and other catalytic systems.

## Results

### DFT calculations and B-site supplement

To reoccupy the B-site vacancies of reduced SFNM scaffold during exsolution, the appropriate filling agent should be selected to initiate the TIE process. For this purpose, the co-segregation energies of Ni, Fe, and Mo at B-site of SFNM were calculated by DFT simulation, which are

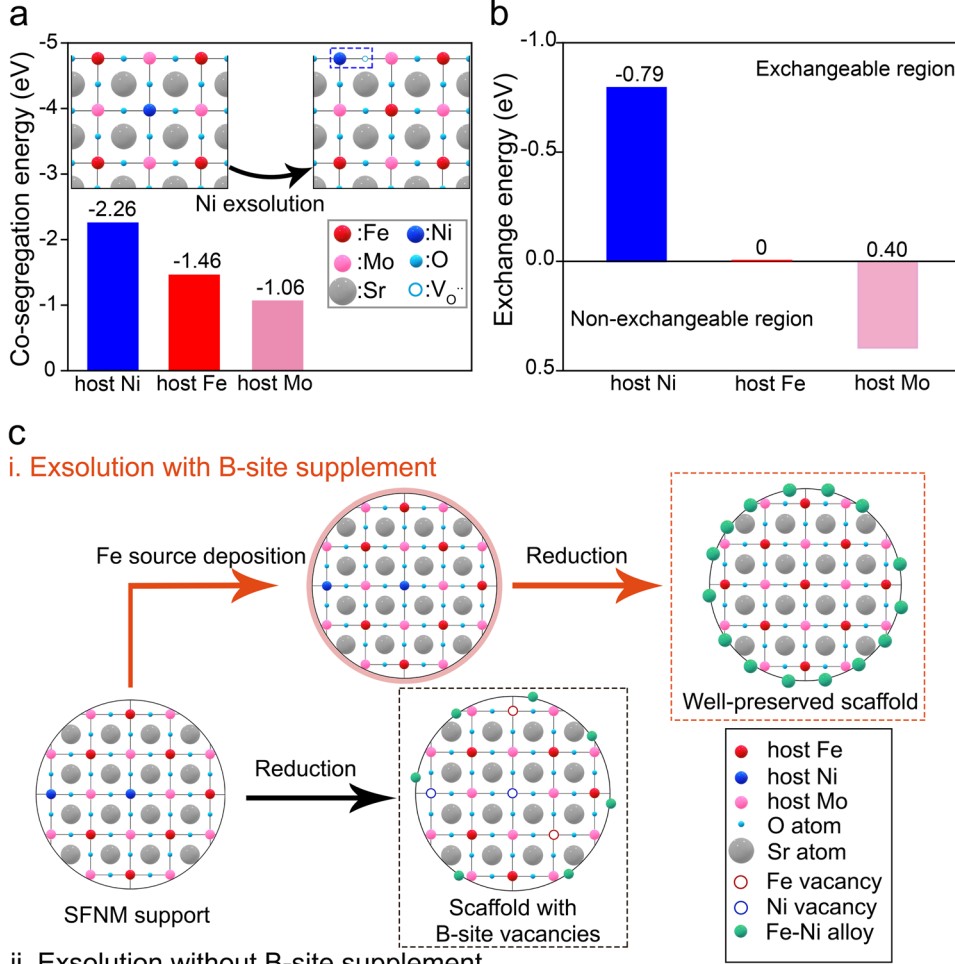

**Fig. 1 | Feasibility of B-site supplement calculated by DFT calculations and schematic illustrations of two exsolution process. a** Co-segregation energy and schematic illustrations of the DFT models for co-segregation by conventional exsolution. **b** Exchange energy comparison of B-site cations of SFNM with guest Fe. **c** Schematic illustration of exsolution with and without B-site supplement on SFNM.

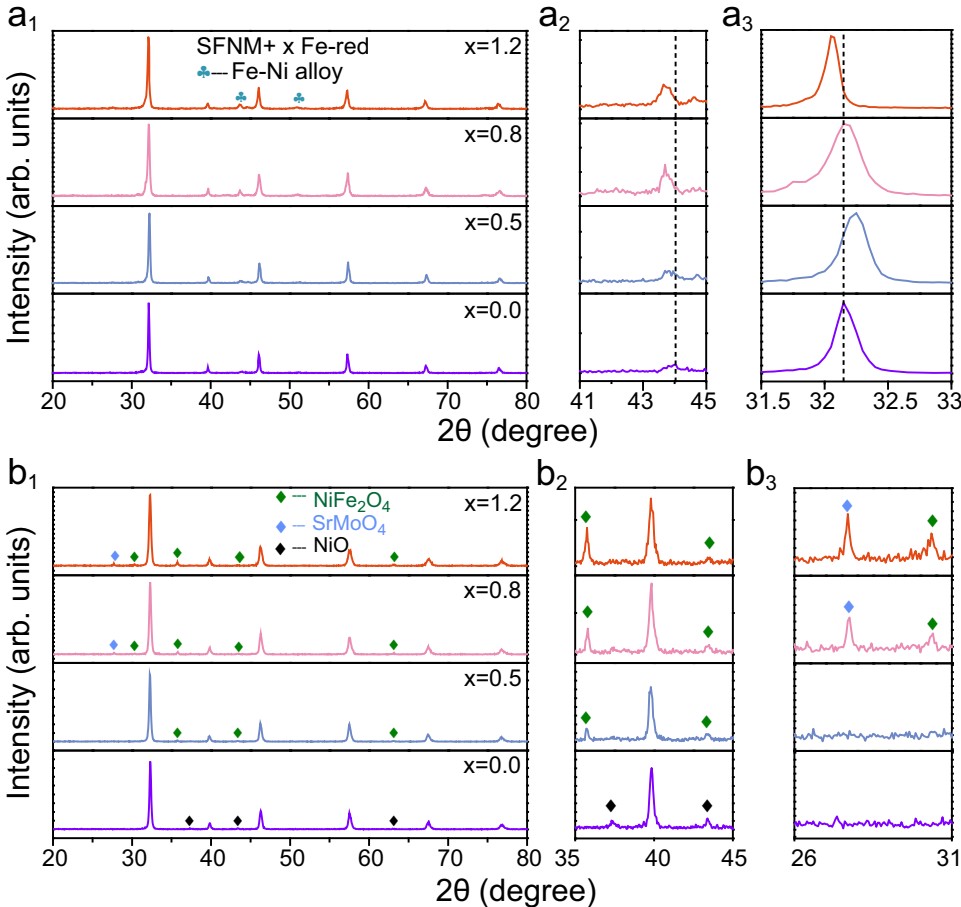

**Fig. 2 | Structure characterizations after reduction and reoxidation. $a_1$** XRD patterns of SFNM+$x$Fe ($x$ = 0.0, 0.5, 0.8, 1.2) reduced at 800 °C for 2 h in 5% $H_2$/$N_2$ from 20 to 80°. Amplified XRD patterns in the range of **$a_2$** 41–45° and **$a_3$** 31.5–33°. **$b_1$** XRD patterns of re-oxidized SFNM+$x$Fe-red at 1000 °C in air for 10 h. Amplified XRD patterns in the range of **$b_2$** 35–45° and **$b_3$** 26–31°.

−2.26 eV, −1.46 eV, and −1.06 eV, respectively (Fig. 1a). It indicates that Fe exsolves more favorably than Mo but less favorably than Ni[30]. Considering that TIE is driven by the difference in co-segregation between guest ion and host ion[25], the exchanges of guest Fe↔ host Ni (−0.79 eV) and guest Fe↔ host Fe (0 eV) are thermodynamically more favorable than that of guest Fe↔ host Mo (0.40 eV) when external Fe ion is introduced on the surface of SFNM substrate (Fig. 1b), suggesting the feasibility of refilling of guest Fe into the B-site vacancies and exsolution of Fe-Ni alloy.

With the guidance of DFT calculations, two different exsolution routes are recapitulated in Fig. 1c for schematic illustrations. Route i depicts the TIE-assisted exsolution initiated on the guest Fe-SFNM complex while the conventional counterpart is shown as route ii. In the TIE-assisted approach, the foreign Fe ions are initially deposited on the surface of perovskite by freeze drying (Supplementary Fig. 1), which will then be incorporated into the remaining B-site vacancies, accompanied by the egression of host Ni and Fe upon reduction (Eq. 4).

TIE-assisted exsolution:

$$Fe^X_{Fe} + Ni^X_{Ni} + 2O^X_O + 2Fe_{guest} \rightarrow Fe_{guest}{}^X_{Fe} + Fe_{guest}{}^X_{Ni} + 2V^{\cdot\cdot}_O + 4e' + Fe - Ni + O_2 \quad (4)$$

For the conventional exsolution, Fe- and Ni-site vacancies emerge in perovskite matrix with the formation of Fe-Ni alloy nanoparticles at the surface (Eq. 5).

Conventional exsolution:

$$Fe^X_{Fe} + Ni^X_{Ni} + 2O^X_O \rightarrow V''_{Fe} + V''_{Ni} + 2V^{\cdot\cdot}_O + Fe - Ni + O_2 \quad (5)$$

## Examination of the exsolved nanoparticles and perovskite scaffold

To determine the optimal Fe loading amount on surface of SFNM, a series of SFNM+$x$Fe-red samples ($x$ = 0.0, 0.5, 0.8, 1.2, which refer to the molar ratios of guest Fe to host Ni) were prepared. X-ray diffraction (XRD) analysis confirms that the Fe-Ni alloy phases have emerged in all SFNM+$x$Fe-red samples with the well-preserved perovskite phase (Fig. 2$a_1$). Nevertheless, further increasing $x$ to 1.5 gives rise to metallic Fe phase (PDF# 06-0696) (Supplementary Fig. 2), suggesting that the excessive surface Fe seeds trigger the formation of Fe clusters at this deposition level. Therefore, the interplay between the nanoparticle formation and perovskite structure evolution among SFNM+$x$Fe-red ($x$ = 0.0, 0.5, 0.8, 1.2) are further elucidated. The characteristic Fe-Ni alloy peaks in SFNM+0.0Fe-red can be well indexed to FeNi₃ phase (PDF#03-065-3244) (Supplementary Fig. 3), while the peak located at 44.1° among SFNM+$x$Fe-red ($x$ = 0.5, 0.8, 1.2) slightly shifts leftward (Fig. 2$a_2$). Transmission electron microscopy (TEM) with energy-dispersive X-ray spectroscopy (EDS) element mappings on randomly selected nanoparticles of SFNM+0.0Fe-red and SFNM+1.2Fe-red reveal the significant increase of Fe proportion in the exsolved nanoparticles of SFNM+1.2Fe-red (Supplementary Fig. 4 and Supplementary Table 1). It may be presumably explained by the involvement of guest Fe in the nanoparticle growth, and the slightly smaller electronegativity of Fe

than that of Ni causes the lattice expansion of Fe-Ni alloy[31–33]. More importantly, distinct peak-shifting of parent perovskites can be observed from the XRD patterns of SFNM+$x$Fe-red. As shown in Fig. 2a$_3$, the magnified diffraction peak at about 32° of SFNM+0.5Fe-red shifts slightly to a higher angle with regards to SFNM+0.0Fe-red, which may be ascribed to the partial filling of B-site vacancies with the guest Fe. Oppositely, the peak shifts leftward when further increasing the deposited Fe content to $x = 0.8$ and 1.2, mainly due to the loss of lattice oxygen after the accelerated exsolution[34]. It in turn indicates the promotion of exsolution and substantial occupation of B-site vacancies by the guest Fe in SFNM+0.8Fe-red and SFNM+1.2Fe-red.

Field emission-scanning electron microscopy (FE-SEM) results demonstrate that the nanoparticle exsolution among SFNM+$x$Fe-red ($x = 0.5, 0.8, 1.2$) is more efficient compared with that on SFNM+0.0Fe-red, as illustrated by the larger average size, wider size distribution and larger population (Supplementary Figs. 5a–d, 6). SFNM+$x$Fe-red samples ($x = 0.0, 0.5, 0.8, 1.2$) were subsequently subjected to O 1$s$ X-ray photoelectron spectroscopy (XPS) analysis to inspect the surface oxygen species. As shown in Supplementary Fig. 7, the concentration of surface-adsorbed oxygen gradually increases as the surface Fe deposition increases, which is consistent with the promoted exsolution observed by SEM.

To further verify the reoccupation of B-site by the guest Fe, SFNM+$x$Fe-red samples ($x = 0.0, 0.5, 0.8, 1.2$) were re-oxidized to inspect the ability of exsolved nanoparticles to reintegrate into the parent perovskite lattice[35–37]. After reoxidation at 1000 °C for 10 h, XRD analysis shows that the additional NiO peaks have clearly emerged in re-oxidized SFNM+0.0Fe-red, while NiFe$_2$O$_4$ or SrMoO$_4$ can be detected on re-oxidized SFNM+$x$Fe-red ($x = 0.5, 0.8, 1.2$) (Figs. 2b$_1$–b$_3$). It can be concluded that the redissolution of the exsolved FeNi$_3$ nanoparticles is partially reversible on the SFNM+0.0Fe-red. Moreover, Fe atoms in FeNi$_3$ nanoparticles preferentially dissolve into perovskite compared to Ni atoms, which is in accord with the fact that Ni exsolves more favorably than Fe (Fig. 1a). As a result, the residual Ni on the surface transforms into NiO after heating in air[38]. However, for SFNM+$x$Fe-red ($x = 0.5, 0.8, 1.2$), the B-site vacancies of parent perovskite have been occupied by the guest Fe partially/entirely; the exsolved Fe and Ni on the surface are prone to self-assembly into binary oxide NiFe$_2$O$_4$ in the air rather than dissolve into perovskite lattice. In addition, the secondary phase SrMoO$_4$ detected on re-oxidized SFNM+0.8Fe-red and SFNM+1.2Fe-red is due to the fact that B-sites were almost completely occupied by the high-valence Fe and Mo, exceeding the tolerance of perovskite structure. This is consistent with the emergence of SrMoO$_4$ in the air-sintered SFM[39] (Supplementary Fig. 8). Therefore, it suggests that the target P-eNs have been successfully synthesized where Fe-Ni alloy nanoparticles are on the SFM substrate with decreased B-site vacancies and Ni incorporation.

## Electrocatalytic performance

The electrocatalytic capacities of SFNM+$x$Fe-red for CO$_2$ reduction were subsequently examined by current density-voltage ($j$–$V$) curves at 800 and 850 °C, as shown in Supplementary Fig. 9. The cathode electrodes were prepared by mixing SFNM+$x$Fe-red with gadolinium doped ceria (GDC), and the full cells for the measurements of ((La$_{0.6}$Sr$_{0.4}$)$_{0.95}$Co$_{0.2}$Fe$_{0.8}$O$_{3-\delta}$-GDC|GDC|YSZ|GDC|SFNM+$x$Fe-red-GDC) are denoted as SFNM+$x$Fe-red-GDC for simplicity. Apparently, SFNM+1.2Fe-red-GDC exhibits superior electrocatalysis performance compared with others, indicative of its better catalytic activity towards CO$_2$ conversion. Specifically, the $j$ reaches 0.97 and 1.13 A cm$^{-2}$ for SFNM+0.0Fe-red-GDC and SFNM+1.2Fe-red-GDC, respectively, at 1.6 V and 850 °C (Fig. 3a).

To further gain insights into the CO$_2$ electrolysis in SOEC, SFNM+0.0Fe-red-GDC and SFNM+1.2Fe-red-GDC were subjected to the electrochemical impedance spectra (EIS) analysis. Figure 3b shows the Nyquist plots of both cells at the applied potentials of 1.0, 1.2, 1.4,

1.6 and 1.8 V at 850 °C and the corresponding equivalent circuit model LR(Q$_H$R$_H$)(Q$_L$R$_L$) (inserted image). The simulated ohmic resistance (R$_S$), polarization resistances at high and low frequencies (R$_H$ and R$_L$) are summarized in Supplementary Table 2. Both cells show the similar R$_S$ due to their identical cell assembly (Supplementary Fig. 10), while the R$_P$ values of SFNM+1.2Fe-red-GDC are comparably smaller than that of the SFNM+0.0Fe-red-GDC at all the monitored potentials, suggesting the faster cathode kinetics for CO$_2$ reduction over SFNM+1.2Fe-red-GDC. Interestingly, R$_P$ of SFNM+0.0Fe-red-GDC firstly drops with the increase of voltage (from 1.0 to 1.6 V), following an increase at 1.8 V, while R$_P$ drops monotonously as the voltage increases over SFNM+1.2Fe-red-GDC. This confirms that the exceptional kinetic performance of SFNM+1.2Fe-red-GDC is well maintained even under harsh poling conditions. Remarkably, the R$_P$ at 1.8 V for SFNM+1.2Fe-red-GDC reaches an appreciably low value of 0.09 Ω cm$^2$, almost half the value of that for SFNM+0.0Fe-red-GDC (0.17 Ω cm$^2$). In addition, the Nyquist plots and the fitted R$_p$ of SFNM+1.2Fe-red-GDC at 800 °C are obtained (Supplementary Fig. 11 and Supplementary Table 3). It shows the lowest R$_p$ at high voltages (≥1.6 V) among the state-of-art P-eNs-based cathode candidates as listed in Supplementary Table 4, indicative of the higher catalytic activities of SFNM+1.2Fe-red-GDC at higher negative potentials.

In conjunction with the EIS data, distribution of relaxation times analysis was used to discern the contributions of underlying kinetic processes to the polarization resistances over these two cells[40]. As shown in Supplementary Fig. 12, the electrode process can be deconvoluted into four peaks, which are denoted as (1) oxygen ion transfer through electrolyte and oxygen evolution at anode (P$_1$), (2) charge transfer (P$_2$), (3) surface CO$_2$ adsorption and activation (P$_3$), (4) gas diffusion process (P$_4$) at cathode from high to low frequency[37,41]. Notably, the integral areas of P$_2$ and P$_3$ peaks on SFNM+0.0Fe-red increase as voltage rises from 1.6 to 1.8 V, as opposite to that on the SFNM+1.2Fe-red (Fig. 3c). It can be inferred that the higher surface reactivity, the faster charge transfer on SFNM+1.2Fe-red, whereas the passivated surface would deteriorate the charge transfer on SFNM+0.0Fe-red at 1.8 V.

## The origin of the enhanced catalytic activity

To understand the origin of the difference in catalytic performances between these two cells for CO$_2$ reduction at high voltages, further characterizations were carried out on SFNM+0.0Fe-red and SFNM+1.2Fe-red. First, Fourier transform infrared spectroscopy (FTIR) measurement was performed to compare their CO$_2$ adsorption capacities[14,42]. As shown in Fig. 3d and Supplementary Fig. 13e–f, SFNM+1.2Fe-red exhibits a higher chemical CO$_2$ adsorption signal with a quicker response time at 600 °C than the SFNM+0.0Fe-red, implying the superiority of SFNM+1.2Fe-red in terms of surface CO$_2$ adsorption and reactivity. However, SFNM+0.0Fe-red presents more intensive physical adsorption peaks compared with SFNM+1.2Fe-red at all the examined temperatures (Fig. 3d and Supplementary Fig. 13b). It may be ascribed to its higher surface alkalinity caused by more severe Sr-segregation. As shown in Eq. 6, the Fe-Ni nanoparticle exsolution, together with the exhaustion of B-site cations and lattice oxygen, in a conventional way would trigger the stripping out of Sr cations to re-establish stoichiometry.

Conventional exsolution:

$$Sr_2Fe_{1.3}Ni_{0.2}Mo_{0.5}O_{6-\delta} \xrightarrow{\text{Reduction}} Sr_{2-x-y}Fe_{1.3-x}Ni_{0.2-y}Mo_{0.5}O_{6-\delta'} + (x+y)SrO + xFe - yNi$$

(6)

The Sr-rich surface would reorganize into SrCO$_3$ at a high CO$_2$ concentration, which has a detrimental effect on the surface catalytic activity for CO$_2$ reduction[43]. Furthermore, high voltages would impose high reducing potentials at the cathode, the concomitant exsolution

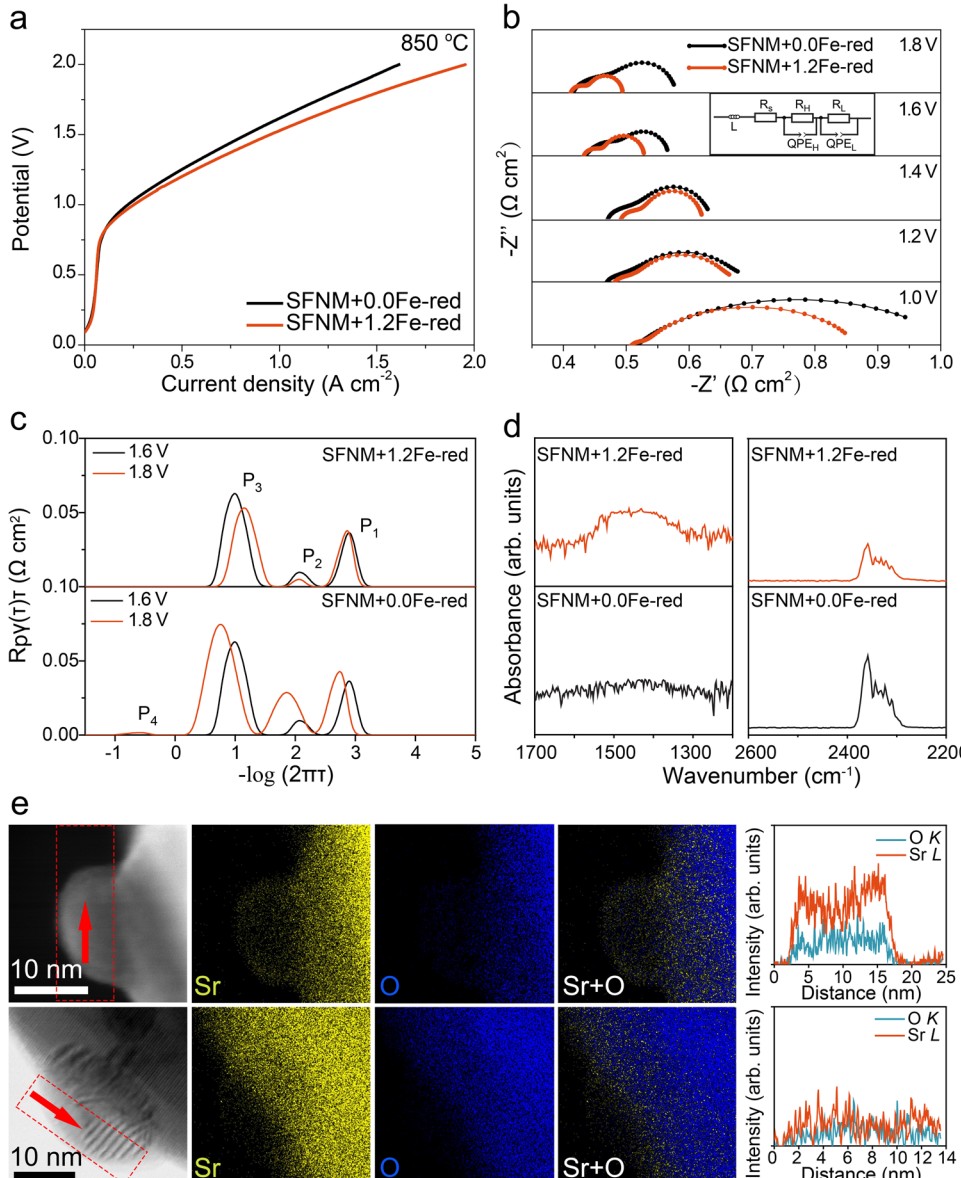

**Fig. 3 | Electrocatalysis performances and surface activity characterization.**
**a** Current density-voltage curves and **b** EIS of SFNM+0.0Fe-red-GDC and
SFNM+1.2Fe-red-GDC for pure $CO_2$ electrolysis at 850 °C. **c** DRT analyses of the
EIS for SFNM+0.0Fe-red-GDC and SFNM+1.2Fe-red-GDC at 1.6 and 1.8 V. **d** FTIR
spectra of $CO_2$ chemisorption and physisorption for SFNM+0.0Fe-red and
SFNM+1.2Fe-red at 600 °C. **e** Overlap of Sr- and O- EDS signals for SFNM+0.0Fe-red
and SFNM+1.2Fe-red.

and Sr segregation would be sustained by the voltage driving force. On
the contrary, the undesired Sr segregation could be efficiently alle-
viated when the B-site vacancies are occupied by the guest Fe; the
active sites can be well exposed even at high potentials (Eq. 7).

TIE-assisted exsolution:

$$Sr_2Fe_{1.3}Ni_{0.2}Mo_{0.5}O_{6-\delta} + (x+y)Fe_{guest} \xrightarrow{Reduction} Sr_2Fe_{1.3-x}Ni_{0.2-y}Fe_{guest_{x+y}}Mo_{0.5}O_{6-\delta'}$$
$$+ xFe - yNi$$

(7)

The difference in Sr segregation between SFNM+0.0Fe-red and
SFNM+1.2Fe-red has also been verified by comparable EDS signal
intensity of Sr in the form of thin shell around the exsolved
nanoparticles[19] (Fig. 3e). Combined with the facilitated formation of
Fe-Ni alloy nanoparticles and surface-active oxygen vacancies, it in
turn persuasively confirms the evolution of initial surface of
SFNM+1.2Fe-red into a more catalytically active surface by B-site filling,

which contributes to enhanced $CO_2$ adsorption and activation at high
operating voltages[14].

Thermogravimetric analysis reveals that the lattice oxygen loss
increases with the guest Fe deposition amount among SFNM+$x$Fe-red
($x$ = 0.0, 0.5, 0.8, 1.2), indicative of higher oxygen vacancy concentra-
tion within perovskite scaffold of SFNM+1.2Fe-red than that of SFNM
+0.0Fe-red (Supplementary Fig. 14). Since the oxygen ion transfer in
perovskite is realized by reverse jumping of oxygen vacancies, the
oxygen ion conductivity is also dependent on the mobility of oxygen
vacancies in addition to oxygen vacancy concentration[20]. Apparently,
the association between neighboring B-site vacancy and oxygen
vacancy on SFNM+0.0Fe-red is likely to trap the oxygen vacancy. The
calculated binding energy of the $V_{Fe}'' - V_O^{..}$ defect pair is −1.73 eV
(Supplementary Fig. 15 and Supplementary Table 5), indicating that the
additional association barrier needs to be overcome to achieve the
jumping of oxygen vacancies[20]. In contrast, the full occupation of
B-sites on SFNM+1.2Fe-red allows the elimination of constraint of B-site

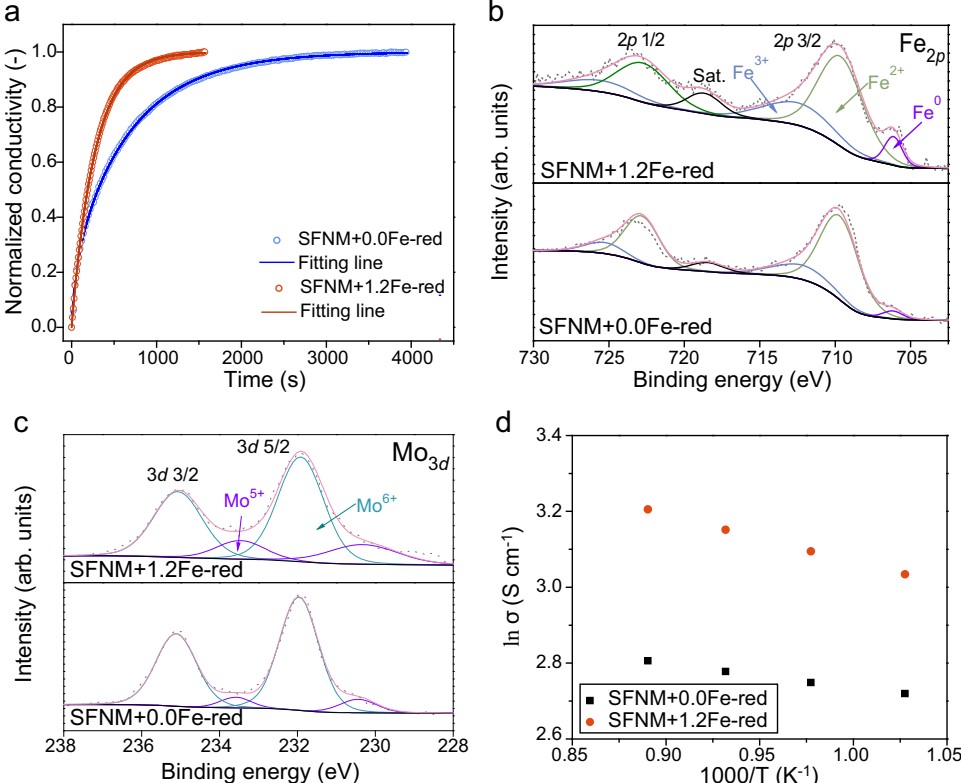

**Fig. 4 | Charge transport characterizations and X-ray photoelectron spectroscopy for SFNM+0.0Fe-red and SFNM+1.2Fe-red. a** Normalized electrical conductivity relaxation curves for SFNM+0.0Fe-red and SFNM+1.2Fe-red at 850 °C with the switching of the gas stream from 2:1 CO–$CO_2$ to 1:1 CO–$CO_2$. XPS of **b** Fe 2$p$ and **c** Mo 3$d$ for SFNM+0.0Fe-red and SFNM+1.2Fe-red. **d** Temperature-dependent electrical conductivities of SFNM+0.0Fe-red and SFNM+1.2Fe-red under 50% $CO_2$/50% CO atmosphere.

defect to surrounding oxygen vacancies. The electrical conductivity relaxation (ECR) experiments for SFNM+0.0Fe-red and SFNM+1.2Fe-red were carried out to study the oxygen ion bulk diffusion[44,45] (Fig. 4a). The bulk diffusion constant ($D_{chem}$) of SFNM+1.2Fe-red obtained by fitting the ECR curve is $3.747 \times 10^{-4}$ cm$^2$ s$^{-1}$, much higher than $1.439 \times 10^{-5}$ cm$^2$ s$^{-1}$ of the SFNM+0.0Fe-red[46]. It suggests that there are more available oxygen ion transfer pathways in SFNM+1.2Fe-red, which lead to the higher oxygen ion conductivity and lower charge transport resistance.

Since the valence states of B-site cations have a great influence on the electronic transfer within perovskite bulk, the electronic environments of B-site cations on SFNM+0.0Fe-red and SFNM+1.2Fe-red were examined by the XPS, X-ray absorption near edge structure (XANES) and extended X-ray absorption fine structure (EXAFS) spectra[47–49] (Fig. 4b–c and Supplementary Figs. 16–17). As shown in Fig. 4b, higher proportion of $Fe^{3+}$ (23.30%) in SFNM+1.2Fe-red can be observed than that in SFNM+0.0Fe-red (16.29%). In parallel, there is a distinct reduction in the average oxidation state of the Mo atom in SFNM+1.2Fe-red (+5.67) with respect to that in SFNM+0.0Fe-red (+5.89), which was derived from partial reduction of $Mo^{6+}$ to $Mo^{5+}$ (Fig. 4c), indicating the coexistence of $Fe^{3+}$-$Mo^{5+}$ and $Fe^{2+}$-$Mo^{6+}$ electronic configurations to achieve charge neutrality ($Fe^{3+}$+$Mo^{5+}$ = $Fe^{2+}$+$Mo^{6+}$)[50]. More $Fe^{2+}$-$Fe^{3+}$ and $Mo^{5+}$-$Mo^{6+}$ charge pairs should endow SFNM+1.2Fe-red with the improved electronic conductivity owing to the sufficient $B^{(n-1)+}$-O-$B^{n+}$ conduction pathways in SFNM+1.2Fe-red. The temperature-dependent electrical conductivity results show that SFNM+1.2Fe-red performs better than SFNM+0.0Fe-red in the 50% $CO_2$/50% CO and 5% $H_2$/95% Ar atmospheres (Fig. 4d and Supplementary Fig. 18). In the 50% $CO_2$/50% CO atmosphere, the electrical conductivities of SFNM+0.0Fe-red and SFNM+1.2Fe-red increase as the temperature increases, reaching 16.5 and 24.6 S cm$^{-1}$ at 850 °C,

respectively (Fig. 4d). SFNM+1.2Fe-red shows the higher conductivity in the prospective operation atmosphere of $CO_2$ electrolysis.

In summary, the B-site supplement of the reduced SFNM by external Fe source plays a crucial role in preserving the high surface activity, high oxygen ion conductivity and electronic conductivity at high voltages, thus leading to the faster cathode reaction kinetics.

## Short-term/long-term stability performances

The cathode stability under operational conditions is a vital criterion to evaluate the $CO_2$ electrocatalysis performances in SOEC. The 15 min potential step chronoamperometry was firstly performed at 850 °C to gain initial indication of the stability of SFNM+0.0Fe-red-GDC and SFNM+1.2Fe-red-GDC. Online gas chromatography (GC) was employed to monitor the CO formation during the short-term stability tests. These two cells deliver quite similar $j$ at negative potentials of 1.0, 1.2, and 1.4 V (Fig. 5a). Upon increasing the potential to 1.6 V and then to 1.8 V stepwise, stark differences in $j$ have emerged. The significant decay of $j$ can be observed over SFNM+0.0Fe-red-GDC, whereas there is negligible $j$ decline over SFNM+1.2Fe-red-GDC under both polarization conditions. SFNM+1.2Fe-red-GDC shows the competitive short-term stability at the voltages ≥1.6 V in comparison with the SFNM+0.0Fe-red-GDC and other state-of-art P-eNs-based SOECs (Supplementary Table 6). Additionally, both cells could steadily generate CO with appreciable Faraday efficiency ($FE_{CO}$) (~90%) at all the potentials (Fig. 5b). As expected, the CO production rate for SFNM+1.2Fe-red-GDC gradually increased with increasing the external voltages, while the production rate for SFNM+0.0Fe-red-GDC peaked at 1.6 V and then decreased upon further raising the potential to 1.8 V. Considering that the nanoparticles closely socketed on the surface of SFNM+0.0Fe-red and SFNM+1.2Fe-red can retain the virtue of high resistance to agglomeration[51] (Supplementary Fig. 19), the

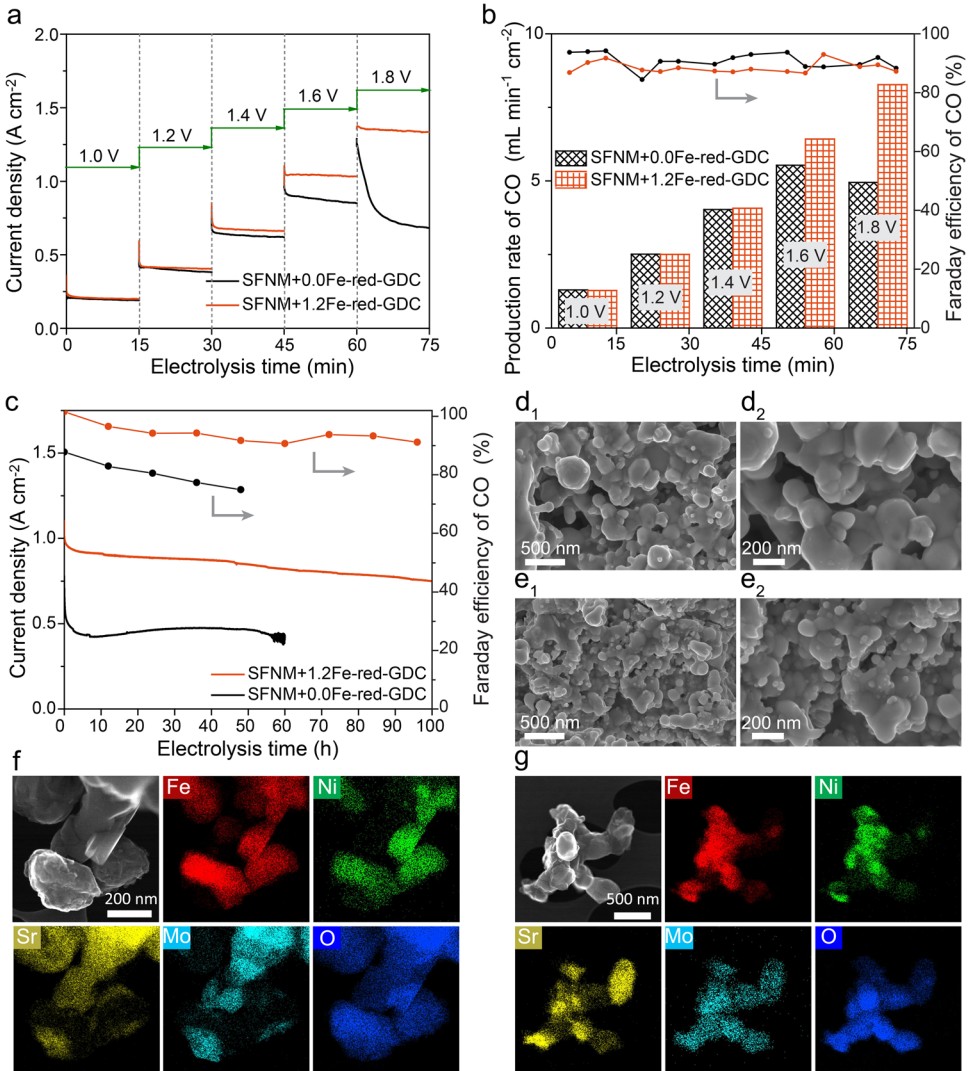

**Fig. 5 | Stability performances and post-mortem surface microstructural characterizations. a** The current density response curves for the 15 min potential step chronoamperometry and **b** corresponded CO productivity and $FE_{CO}$ for SFNM+0.0Fe-red-GDC and SFNM+ 1.2Fe-red-GDC at 850 °C with pure $CO_2$ (Gray shaded areas indicate the applied voltages). **c** 100 h long-term stability testing for SFNM+0.0Fe-red-GDC and SFNM+ 1.2Fe-red-GDC at 1.6 V and 850 °C in pure $CO_2$. SEM images of cathode surface microstructure of **$d_1$–$d_2$** SFNM+0.0Fe-red-GDC and **$e_1$–$e_2$** SFNM+1.2Fe-red-GDC after long-term stability at 1.6 V and 850 °C. SE-STEM and STEM-EDS images of cathode on **f** SFNM+0.0Fe-red-GDC, **g** SFNM+1.2Fe-red-GDC after long-term stability at 1.6 V and 850 °C.

discrepancies in $j$ decay and CO productivity appear to be raising from the differences in the structure evolution of perovskite matrix at higher negative potentials.

It should be worth noting that the stable SOEC operation at high voltages has significant implications on improving the energy efficiency and industrial scale of applications[18]. Furthermore, high cathode potentials are applied to intentionally accelerate electrode degradation to help us understand the cathode evolution and further shed lights on the degradation mechanisms. This in turn is of great significance for designing P-eNs materials in line with the goal for industrialization. Figure 5c presents the prolonged stability performances of SFNM+0.0Fe-red-GDC and SFNM+1.2Fe-red-GDC at 850 °C and 1.6 V. Interestingly, SFNM+0.0Fe-red-GDC experiences the current density output instability at around 55 h. Moreover, despite the apparent $j$ decrease over the entire operation, a trend of "degradation–reactivation–degradation" emerges that can be roughly categorized to Phase I (0–8 h), Phase II (8–35 h), and Phase III (35–55 h). Within Phase I, $j$ experiences dramatic decrease with a degradation rate of 44 mA cm$^{-2}$ h$^{-1}$. As the reaction proceeds and enters Phase II, the $j$ slowly climbs up, peaking at 35 h with a maximum $j$ of 0.47 A cm$^{-2}$.

In the following Phase III, the $j$ drops again until it becomes unstable at around 55 h. Especially, $j$ declines at an attenuation rate of 6 mA cm$^{-2}$ h$^{-1}$ from 50 to 55 h. In sharp contrast to the complicated $j - t$ profile on SFNM+0.0Fe-red-GDC, SFNM+1.2Fe-red-GDC manifests a rather simple and mild degradation profile throughout the long-term stability measurement without observable $j$ fluctuations, only having an average degradation rate of 3 mA cm$^{-2}$ h$^{-1}$. The $j$ remains at a high value of 0.75 A cm$^{-2}$ at 100 h. In addition, the recorded $FE_{CO}$ profiles show that SFNM+0.0Fe-red-GDC experienced a significant $FE_{CO}$ decline, only having 75% at 48 h. In contrast, the $FE_{CO}$ of SFNM+1.2Fe-red-GDC has remained at above 90% throughout the 100-h stability test, indicating its superiority in preventing the yielding of by-products.

## Post-mortem characterization and degradation mechanism

After the long-term stability tests, the significant morphology changes and coarsening of exsolved particles can be observed on the both cathode surfaces, and the coarsening of surface nanoparticles on SFNM+1.2Fe-red-GDC is less severe than that on SFNM+0.0Fe-red-GDC (Fig. 5d–e). Both cathodes were scratched from the cells for the element distribution detection by secondary electron-scanning TEM

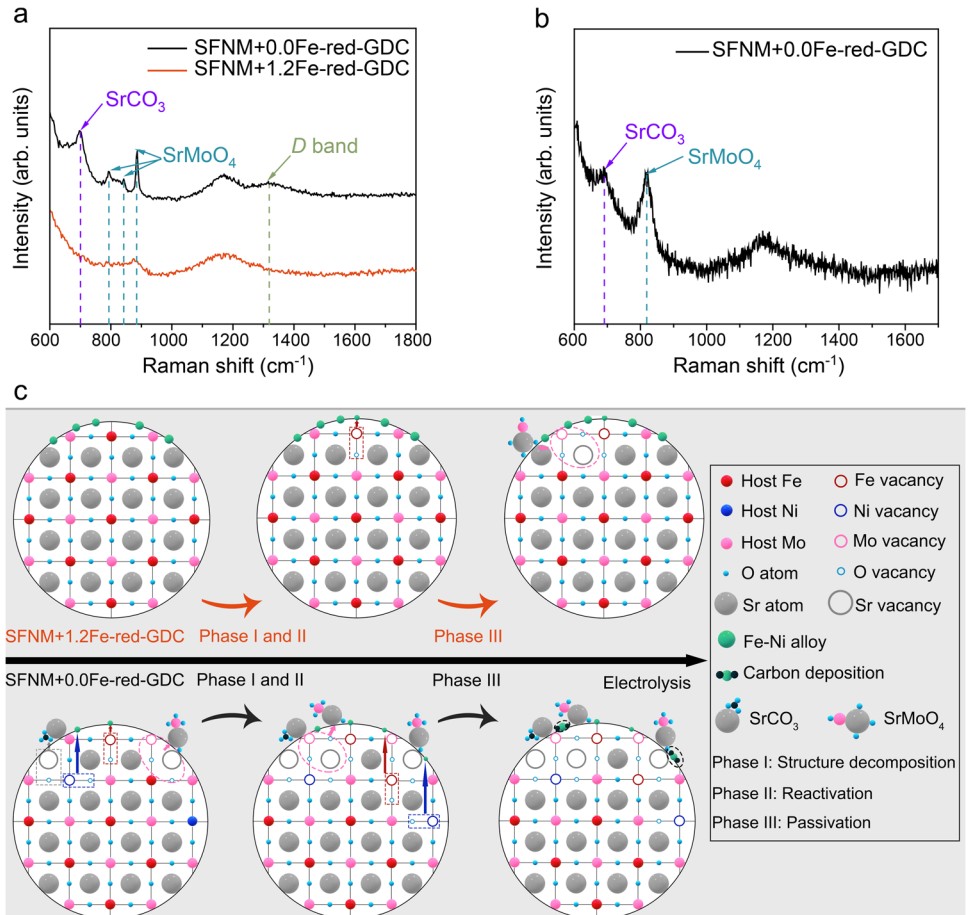

**Fig. 6 | Post-mortem Raman spectra characterizations and illustrations of degradation mechanisms. a** Raman spectra collected from cathode surface of SFNM+0.0Fe-red-GDC and SFNM+1.2Fe-red-GDC after the long-term stability test. **b** Raman spectra collected from cathode surface of SFNM+0.0Fe-red-GDC whose stability test was interrupted before entering Phase II. **c** Structure evolution-driven degradation mechanisms for SFNM+0.0Fe-red-GDC and SFNM+1.2Fe-red-GDC at high negative potentials.

(SE-STEM) and STEM-EDS. More pronounced Sr- and Mo-derived phase separation in the form of irregular particles appear on the surfaces of SFNM+0.0Fe-red-GDC, (Fig. 5f–g), suggesting that SFNM+0.0Fe-red-GDC undergoes the more significant phase decomposition than SFNM+1.2Fe-red-GDC under the synergistic effect of the high voltage and $CO_2/CO$ mixed atmosphere during the $CO_2$ electrolysis. The Raman spectra from cathode surface of both cells were collected (Fig. 6a). As shown, the distinct Raman feature peaks of $SrCO_3$ (located at 701 cm$^{-1}$), $SrMoO_4$ (located at 795, 842, and 887 cm$^{-1}$), carbon with lattice defect (D band, located at 1319 cm$^{-1}$) can be found on the cathode surface of SFNM+0.0Fe-red-GDC[39,52,53], whereas only trace amounts of $SrMoO_4$ peaks were detected on the cathode surface of SFNM+1.2Fe-red-GDC. The emergence of Sr-containing impurities on both spectra confirms the structure decomposition and surface reconstruction on SFNM+0.0Fe-red and SFNM+1.2Fe-red. Obviously, the phase decomposition of SFNM+0.0Fe-red is more thorough. Furthermore, a small amount of carbon deposition on SFNM+0.0Fe-red-GDC may be ascribed to the newly born Ni nanoparticles driven by the applied voltage. Subsequently, SFNM+0.0Fe-red-GDC was subjected to the stability test again and was interrupted before entering the Phase II, and the Raman spectrum of the cathode surface was collected. As shown in Fig. 6b, only the Raman feature peaks of $SrCO_3$ and $SrMoO_4$ were detected but no peaks of carbon, which confirms that the perovskite scaffold of SFNM+0.0Fe-red primarily underwent the bulk structural decomposition before the exposure of newly grown nanoparticles and carbon deposition. This also explains the j evolution over SFNM+0.0Fe-red-GDC.

To verify the exsolution scenario and the structural decomposition during the electrolysis process, the SFNM without pre-reduction treatment was fabricated as the composite cathode material (SFNM-oxi-GDC) for the electrochemical testing (Supplementary Fig. 20) and the post-mortem characterizations (Supplementary Figs. 21–23). As shown in Supplementary Fig. 20d, the SFNM-oxi-GDC shows the similar long-term stability profile as the SFNM+0.0Fe-red-GDC. The SEM, SE-STEM, and STEM-EDS results confirm that partial Ni elements segregate from the SFNM bulk and the newly born fibrous phases appear on the cathode surface (Supplementary Figs. 21–22). Meanwhile, the Raman feature peaks of $SrCO_3$, carbon with lattice defect (D band) can be observed (Supplementary Fig. 23), suggesting that the Ni element in situ exsolves from the SFNM bulk and subsequently results in the carbon fiber growth during $CO_2$ electrolysis. This can be ascribed to the lower co-segregation energy of Ni in SFNM, which causes the structural decomposition under the synergistic effect of 1.6 V and 850 °C.

Accordingly, the severe degradation may be effectively mitigated by preserving the structural integrity and enhancing the resistance to exsolution of perovskite scaffold of SFNM+0.0Fe-red. In SFNM+1.2Fe-red, the robustness of perovskite scaffold has been significantly enhanced by B-site supplement using redox-stable Fe ions, which postpones the decomposition of perovskite bulk and reassembly of Sr-based insulators on surface. Furthermore, the continuous exsolution has been greatly suppressed due to the decreased Ni content in perovskite bulk. This also well accounts for the mild degradation rate, high $FE_{CO}$, no intuitive reactivation process, decreased

Sr-based matters and negligible deposited carbon over SFNM+1.2Fe-red-GDC.

In view of the above analyses, a perspective on the degradation mechanism of P-eNs from the structure stability of perovskite substrate can be proposed. Since the proper negative potential acts as the driving force to drain out B-site reducible cations from perovskite, the high voltage on cathode will not only accelerate $CO_2$ reduction reaction, but also give rise to the continuous and slow exsolution during the electrolysis[23], starting from breaking of weak B–O bonds, diffusion of B-site cations, followed by nucleation and growth of nanoparticles[54]. In parallel, A-site segregation occurs, leading to surface reassembly and perovskite bulk reconstruction. As below, we can tentatively discuss the differences in structure evolutions of SFNM+0Fe-red-GDC and SFNM+1.2Fe-red-GDC in light of their degradation processes (Fig. 6c). For SFNM+0Fe-red-GDC, the profile presents a high $j$ but a faster decay rate in the Phase I (Fig. 5c). This is presumably ascribed to the breaking of B–O bonds and subsequent diffusion of ions from bulk to surface driven by the external voltage[55], which causes deteriorated charge transfer capability of perovskite scaffold and surface assembly of detrimental Sr-based matters (Fig. 6b). This stage is named as "Structure decomposition" regime. However, such a declining profile has been gradually counterbalanced by the newly exposed active nanoparticles derived from diffusion of active cations in the deeper region of perovskite (Phase II in Fig. 5c). It can be seen that this process is very mild, which means that exsolution is limited by the concentration of exsolved cations in the perovskite and proceeds slowly[54]. This stage is termed as the "Reactivation" regime. As electrolysis proceeds, the structural decomposition progresses slowly, and a balance between surface reorganization and bulk reconstruction has gradually reached. Meanwhile, carbon deposition around the nanoparticles becomes increasingly distinct (Fig. 6a). Consequently, $j$ drops again or even becomes unstable (Phase III in Fig. 5c), this period can be named as the "Passivation" zone. Nevertheless, for SFNM+1.2Fe-red-GDC, the "structure decomposition" has been significantly alleviated by the reoccupation of relatively redox-stable Fe ions at B-site vacancies of perovskite. The high applied potential offers a higher initial transient $j$, however, the amplitude of $j$ attenuation has been remarkably reduced at the initial stage compared with that for SFNM+0.0Fe-red-GDC, which proves that the structural decomposition has been greatly alleviated (Fig. 5c). Although the "reactivation" stage is not visible from the $j$ curve, the change in $FE_{CO}$ and phase analysis from Raman results suggest the existence of a small amount of exsolution and surface reconstruction (Figs. 5c and 6a). Moreover, carbon deposition near active sites has been circumvented (Fig. 6a), resulting in a moderate degradation process in the following reaction and a longer operating life. In addition, the deteriorations of the electrolyte and the anode under the harsh conditions seem to be inevitable (Supplementary Fig. 24), which are also responsible for the significant attenuation of the current density during $CO_2$ electrocatalysis at 1.6 V and 850 °C.

## Discussion

In summary, this study has highlighted that the B-site supplement should be an alternative strategy to boost the stability of P-eNs for $CO_2$ electrocatalysis in SOEC. By incorporating the guest Fe ions into the residual B-site vacancies of perovskite scaffold, SFNM+1.2Fe-red exhibits higher surface activity, charge conductivities and structural robustness compared to SFNM+0.0Fe-red, thus leading to both the high catalytic activity and the high stability performances, especially at high voltages (i.e., ≥1.6 V). Furthermore, the degradation mechanisms of the reduced SFNM with/without B-site supplement have been clarified from the perspective of structure stability of perovskite scaffold, revealing the important role of the robust perovskite scaffold in enhancing the stability of P-eNs. This work provides an approach to rational design of the P-eNs with improved activity and stability and also paves the way toward their wide industrial applications as the

efficient electrocatalysts for various energy storage and conversion systems.

## Methods

### DFT computational details

DFT were performed by Vienna Ab initio Simulation Package[56,57]. Perdew-Burke-Ernzer (PBE) function and projector-augmented plane-wave method were employed in order to treat the exchange-interaction effect and electron-ion interaction, respectively[58]. For plane-wave expansion, the cutoff energy was set at 520 eV. A Gaussian smearing with a width of 0.2 eV was used to determine partial occupancies. A $4 \times 2 \times 4$ supercell was built based on the optimized unit cell and a vacuum layer of 15 Angstroms was added in order to avoid the interaction between neighboring slabs. Monkhorst-Pack ($3 \times 6 \times 1$) was set for Brillouim zone sampling[59]. During geometry optimization, the top four atomic layers were fully relaxed while the other atomic layers were fixed. Relaxation of degree of ions was not terminated until a maximum force component of 0.05 eV/Angstrom was achieved. In order to describe the formation of SFNM perovskite, we replaced one Fe atom with one Ni atom in SFM. A PBE+U strategy was used to take the exchange and on-site Coulomb interaction in Fe, Ni into consideration. The values of U were set at 6.2 eV and 5.3 eV for Ni and Fe, respectively. Spin polarization was considered during all calculations. The (001) plane was chosen for the calculation of co-segregation by conventional exsolution and exchange energy between guest Fe and host B-site cations since the (001) lattice plane had been used as the active surface of SFNM in the previous work[44,60]. Vesta software was employed for model construction and visualization[61].

The co-segregation energy is defined as the energy difference between the total energies of the supercell with exsolved metal ion plus its nearest oxygen vacancy in the 1st and 3rd atomic layers of perovskite, as following equation (Taking M as the exsolved metal):

$$E_{co-segregation(M)} = E_{(M-V_O^{\cdot\cdot}@1st\,layer)} - E_{(M-V_O^{\cdot\cdot}@3rd\,layer)}$$

where $E_{co-segregation(M)}$, $E_{(M-V_O^{\cdot\cdot}@1st\,layer)}$, $E_{(M-V_O^{\cdot\cdot}@3rd\,layer)}$ are the co-segregation energy, the total energy of the supercell with exsolved ion plus its nearest oxygen vacancy in the 1st atomic layer, and the total energy of the supercell with exsolved ion plus its nearest oxygen vacancy in the 3rd atomic layer, respectively.

The exchange energy is defined as the difference between the co-segregation energies of the two metal ions involved in exchanging, as shown in following equation:

$$E_{exchange(M_1-M_2)} = E_{co-segregation(M_1)} - E_{co-segregation(M_2)}$$

### Synthesis of materials

SFNM powder was synthesized by a modified sol-gel method. Stoichiometric amounts of $Sr(NO_3)_2$, $Fe(NO_3)_3 \cdot 9H_2O$, $Ni(NO_3)_2 \cdot 6H_2O$ and $(NH_4)_6Mo_7O_{24} \cdot 4H_2O$ were dissolved in EDTA-28% $NH_3 \cdot H_2O$ mixture under continuous stirring, followed by the addition of citric acid. The molar ratio of total metal ions: EDTA: citric acid was 1:1:1.5. Subsequently, 28% $NH_3 \cdot H_2O$ was added dropwise to maintain a pH at ~8. The solution was slowly evaporated at 80 °C until a homogeneous organic/cation resins was achieved. The as-obtained gel was further decomposed at 300 °C for 4 h to remove the organic components and remnant nitrates, then the obtained powder was finally fired at 1000 °C for 5 h in air to yield pristine SFNM.

### Foreign Fe deposition and TIE-assisted exsolution process

The guest Fe ion was introduced on the surface of as-prepared SFNM porous support by an immerse-freeze drying process. Certain amount of $Fe(NO_3)_3 \cdot 9H_2O$ was firstly dissolved in 10 mL deionized water at room temperature and stirred for 2 h. Then, the pre-prepared SFNM

powders were added to the Fe-containing solutions and ultrasound-treated for 3 h. The suspension precursor was then rapidly frozen with liquid nitrogen and freeze dried overnight to achieve the homogeneous Fe-SFNM complex with Fe seeds uniformly coated on perovskite scaffold. The fluffy powder was collected and heated in a tube furnace at 800 °C for 2 h in 5% $H_2/N_2$, yielding the final catalysts SFNM+$x$Fe-red ($x$ = 0.0, 0.5, 0.8, 1.2, which referring to the molar ratio of guest Fe to host Ni). Particularly, SFNM+0.0Fe-red refers to the P-eN after conventional exsolution. Electrolyte material yttria-stabilized zirconia (YSZ), buffer layer material Gadolinium doped Ceria (GDC) and anode material 50% wt. $(La_{0.6}Sr_{0.4})_{0.95}Co_{0.2}Fe_{0.8}O_{3-\delta}-$ 50% wt. GDC (LSCF-GDC) powders are purchased from Fuel Cell Materials.

## Characterizations

The crystalline structures of all the synthesized powders were identified by XRD with Rigaku Ultima IV equipped with a Cu Kα radiation (40 kV, 44 mA). XPS (Kratos AXIS Ultra) was used to examine the electron structure of the reduced perovskites with C 1s at binding energy of 284.8 eV as the reference. The microstructures of reduced samples were revealed by high-resolution filed emission-scanning electron microscopy performed on Zeiss Sigma FE-SEM. The atomic-scale microstructural feature of the Fe-Ni alloy nanoparticles anchored on the perovskite surface was analyzed using JEOL JEM-ARM200CF equipped with the EDS detector. Thermogravimetric analysis (TA SDT Q500) was performed from 50 to 800 °C at a heating/cooling rate of 20 °C min$^{-1}$ in 5% $H_2/N_2$ to characterize the lattice oxygen loss. FTIR was used to investigate the $CO_2$ physical/chemical absorption capacity of reduced samples at elevated temperatures. Firstly, the background peaks were collected at the test temperatures, the FTIR spectra were obtained by subtracting the corresponding background peaks from the overall absorption and then normalizing it with respect to the noise peaks. X-ray absorption measurements were conducted at Canadian Light Source (CLS). Fe K-edge and Ni K-edge XANES spectra were collected using the Very Sensitive Elemental and Structural Probe Employing Radiation from a Synchrotron (VESPERS, 07B2-1) beamline. Temperature-dependent electrical conductivities were carried out in the 50% $CO_2$/50% CO and 5% $H_2$/95% Ar atmospheres. ECR experiments were implemented at 850 °C with switching of the gas stream from 2:1 CO−$CO_2$ ($P_{O_2}$ = 1.246 × 10$^{-18}$ atm at 850 °C) to 1:1 CO−$CO_2$ ($P_{O_2}$ = 4.985 × 10$^{-18}$ atm at 850 °C). Detailed descriptions of the ECR technique and the model used for data fitting are given elsewhere[46]. Raman spectra were recorded by the Renishaw inVia Qontor Confocal Raman Microscope with a laser of 532 nm.

## Cell fabrication

SOEC with a configuration of SFNM-red+$x$Fe ($x$ = 0.0, 0.5, 0.8, 1.2)-GDC|GDC|YSZ|GDC|LSCF-GDC was fabricated. Cathode materials were prepared by mixing SFNM+$x$Fe-red ($x$ = 0.0, 0.5, 0.8, 1.2) and GDC with a weight ratio of 3:2. The cathode slurry was prepared by mixing the SFNM-red+$x$Fe ($x$ = 0.0, 0.5, 0.8, 1.2)-GDC with a glue containing 1-butanol, benzyl butyl phthalate, ethyl cellulose, and α-terpineol in a weight ratio of 1.7:1, followed by ball milling for 3 h. Anode slurry was prepared using the same method as the cathode slurry. GDC buffer layer was screen printed on both sides of YSZ electrolyte, then sintered at 1300 °C for 5 h and lastly, the cathode and anode slurries were printed on both GDC sides. The assembled cell was finally sintered at 1100 °C for 2 h. Gold paste was painted onto the surfaces of both anode and cathode to serve as current collectors.

## Catalytic activity of $CO_2$ reduction in SOEC

The $CO_2$ electrolysis cell was built by fixing the cell between coaxial pairs of alumina tubes with a sealant, which was fastened in a vertical tubular furnace. The 5% $H_2/N_2$ reducing gas flow is continuously pumped into the SFNM+$x$Fe-red-GDC ($x$ = 0.0, 0.5, 0.8, 1,2) cathode compartment during the temperature ramping of SOEC.

The temperature is maintained at 800 °C for 2 h to complete the further reduction and exsolution of SFNM+$x$Fe-red-GDC. During the electrolysis, pure $CO_2$ was fed to the cell with a flow rate of 100 mL min$^{-1}$ via the cathode compartment located at the bottom, while the anode was placed on the top and exposed to air. The cell was mounted in the testing apparatus and the temperature was slowly increased to 800 °C for 2 h to achieve the formation of reduced perovskite support with exsolved nanoparticles.

The electrochemical measurement was conducted using a Solartron 1255 frequency response analyzer and a Solartron 1286 electrochemical interface instrument. The outlet gas from the cathode compartment was analyzed using a Hewlett-Packard model HP5890 GC equipped with a packed column (Porapak QS) operated at 80 °C with a thermal conductivity detector and a flame ionization detector. The polarization resistance of the $CO_2$ electrolysis cell was determined by EIS measured under an ac potential with a frequency range of 1 MHz to 0.1 Hz and an amplitude of 10 mV at applied potentials.

The Faradaic efficiency of CO is calculated by the following equation[34,62]:

$$FE_{CO} = \frac{0.1315 \times V(\frac{mL}{min}) \times v(vol\%)}{I_{total}(A)} \times 100\%$$

$V(\frac{mL}{min})$ = Gas flow rate measured by a flow meter at the exit of the cell at room temperature and under ambient pressure.

$v$ (vol%) = Volume concentration of CO in the exhaust gas from the cell (obtained by GC).

$I_{total}$ (A) = cell current during short-term stability experiments.

The equation calculating the production rate of CO is as follows[63]:

$$\text{Formation rate of CO} = \frac{q_{total} \times FE_{CO} \times 22.4(L/mol) \times 1000 \times 60(\frac{h}{min})}{FntS \times 100}$$

$q_{total}$ = total charge passed.
$FE_{CO}$ = Faraday efficiency of CO.
F = Faraday constant.
$n$ = the number of exchanged electrons to produce CO from $CO_2$.
$t$ = electrolysis time (h).
$S$ = the geometric area of the electrode (cm$^2$).

## Data availability

The data measured, simulated, and analyzed in this study are available from the corresponding author upon request.

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

## Acknowledgements

This work is supported by the Natural Sciences and Engineering Research Council (NSERC) of Canada, Discovery Grant (GRPIN-2016-05494), and Strategic Research Projects of Alberta Innovates Technology Futures (#G2016000655). As a part of the University of Alberta's Future Energy Systems research initiative, this research was made possible in part thanks to funding from the Canada First Research Excellence Fund (CFREF-2015-00001). Bo-Wen Zhang acknowledges the financial support from China Scholarship Council (Grant no. 201806450022).

## Author contributions

B.W.Z. conceived the idea and performed the experiments, and wrote the manuscript. M.N.Z. contributed to the DFT calculations, TEM analysis, and Raman analysis. M.R.G. performed the DFT calculations. X.X. contributed to ECR experiments and temperature-dependent electrical conductivity data. R.F.F. contributed to XANES data. N.D., Z.C., and H.Z. gave constructive comments. J.L.L. supervised the project and edited the manuscript.

## Competing interests

The authors declare no competing interests.

## Additional information

**Supplementary information** The online version contains

supplementary material available at https://doi.org/10.1038/s41467-022-32393-y.

