## [Peer Review File · Nature Communications]

REVIEWER COMMENTS

Reviewer #1 (Remarks to the Author):

In this manuscript, the authors developed highly stable exsolution SOEC for CO₂ reduction under high voltage ($V > 1.6$ V). The authors used recently developed topotactic ion exchange method for developing the highly stable exsolution SOEC. The authors' work has two important points. Firstly, they pointed out the poor stability issue of exsolution SOEC for CO₂ reduction under high voltage. Secondly, they proposed a solution for the poor stability issue of exsolution SOEC for CO₂ reduction by applying the recently developed topotactic ion exchange method. However, they ignored the reason why should we use the exsolved SOEC instead of the other conventional perovskite SOEC which have much higher current density with higher stability. Also, for revealing the degradation mechanism, more detailed characterizations are required. Therefore, I recommend following major revisions should be clear before this manuscript will be published.

1. Without exsolution, SFMM (Sr₂Fe_{1.4}Mn_{0.1}Mo_{0.5}O₆) perovskite based SOEC showed two times higher current density than SFNM (Sr₂Fe_{1.3}Ni_{0.2}Mo_{0.5}O₆) based exsolution SOEC in this manuscript [J. Mater. Chem. A, 2019, 7, 11967].

A. This manuscript should discuss the essential advantages of exsolved nanoparticles for SOEC application and compare its performance (current density, CO conversion) with other state-of-art SOEC for CO₂ reduction.

B. Only two exsolution SOECs were compared in this manuscript. If the authors want to clearly show the degradation by exsolution, they need to additionally show the performance and stability of SFNM (Sr₂Fe_{1.3}Ni_{0.2}Mo_{0.5}O₆) based SOEC without exsolution.

2. The HR-TEM results for SFNM+0.0Fe-red and SFNM+0.0Fe-red were clearly showed the difference between the conventional and TIE exsolved nanoparticles. What are the morphologies like for each of them after stability tests?

3. (minor comment) The concept of vacancy trapping in fig. 3e needs experimental or theoretical verifications. Diffusion barrier calculation using DFT would be best to verify it.

4. (minor comment) The caption of Figure 4d and 4e was reversed.

5. (minor comment) The number of figures is too small. Move important supplementary figures to main figures.

Reviewer #2 (Remarks to the Author):

In this work titled “B-site supplement mechanism: A new approach to boosting stability of perovskites with exsolved nanoparticles”, the authors describe the effect of B-site supplement (or topotactic ion exchange) of Fe nitrate guest in double perovskite $\text{Sr}_2\text{Fe}_{1.3}\text{Ni}_{0.2}\text{Mo}_{0.5}\text{O}_{6-\delta}$ (SFMN). Unfortunately, I have to say that the electrochemical part of this paper is poor and there are too many flaws in this manuscript. Thus, I can't recommend this manuscript to be published in Nature Communications. Some major comments and questions are listed as below:

1. The main point of this work is to promote topotactic ion exchange (TIE) of Fe guest source in $\text{Sr}_2\text{Fe}_{1.3}\text{Ni}_{0.2}\text{Mo}_{0.5}\text{O}_{6-\delta}$ (SFMN) double perovskite. In my opinion, this work lacks novelty since the topotactic exchange has been already reported in reference 25 and 26 in the authors' work. Moreover, Figure 1c and 1d in this work is too much similar with the Figure 1a and 1b in the reference 26.

Reference 26: Joo, S. et al. Cation-swapped homogeneous nanoparticles in perovskite oxides for high power density. *Nat. Commun.* 10, 697 (2019).

2. In Supplementary figure 2, there seems to be unknown peaks in my position. I recommend designating all the peaks for Supplementary Figure 2 to remove confusion. In addition, why did the authors only put SrMoO_4 as the PDF card for air-sintered SFM? If SrMoO_4 exists, where are the Fe source?

3. The same or similar host material has been already reported (reference 27 and the below □ additional reference).

Reference 27: Lv, H. et al. In Situ Investigation of Reversible Exsolution/Dissolution of CoFe Alloy Nanoparticles in a Co-Doped $\text{Sr}_2\text{Fe}_{1.5}\text{Mo}_{0.5}\text{O}_{6-\delta}$ Cathode for CO_2 Electrolysis. *Adv. Mater.* 32, e1906193 (2020)

Additional reference: Wang, Y., Liu, T., Li, M., Xia, C., Zhou, B., & Chen, F. (2016). Exsolved Fe–Ni nanoparticles from $\text{Sr}_2\text{Fe}_{1.3}\text{Ni}_{0.2}\text{Mo}_{0.5}\text{O}_6$ perovskite oxide as a cathode for solid oxide steam electrolysis cells. *Journal of Materials Chemistry A*, 4(37), 14163-14169.

4. Why is there non-linearity from the voltage range of 0 to about 0.8 V in the I-V curves in Supplementary Figure 9? Also, why is the open-circuit voltage too low?

5. Are there any Nyquist plots for the EIS measurements in Supplementary figure 9 and 10? In general, the DRT plot (Supplementary figure 10) is originated from the Nyquist plot for the EIS measurement.

6. There are too many slope changes in the TGA plot in Supplementary Figure 12. What could be the possible reason for many slope changes? Also, the TGA measurement condition is not listed in the caption of Supplementary figure 12.
7. Are there any additional supporting data (e.g., X-ray absorption fine structure (XAFS) measurement) to further support the XPS data in Supplementary figure 13?
8. In my opinion, this work did not follow the Nature Communications format. Is the font for this work "times" or "times new roman"?
9. Why did the authors use electrode-GDC composite for SOEC measurements? Since the GDC could significantly affect the exsolution and/or TIE properties, the authors should compare the electrochemical performance measurements with only electrode material (SOEC cathode).
10. In page 7 line 15, the authors stated as "X-ray diffraction (XRD) analysis confirms that secondary Fe-Ni alloy phases have emerged ...". In general, secondary phase could also imply unnecessary or impurity phase. However, the Fe-Ni alloy exsolution would positively affect the electro-catalytic properties, as also listed in this work.
11. What is the possible reason for ultrasound (ultrasonication) and freeze-drying Fe nitrate nonahydrate in the host material (Supplementary figure 1)? In addition, how much weight percent of Fe guest did the authors immerse in the SFMN material to promote TIE phenomenon?
12. In Figure 2, there seems to be secondary phases (NiO, NiFe₂O₄, and SrMoO₄) after re-oxidation in air. The formation of secondary phases could negatively affect the electrochemical properties. Moreover, the reason for conducting re-oxidation in air is not clearly explained in this work.
13. Why did the authors conduct reduction at 5% H₂/N₂ balance instead of real experimental conditions for SOEC cathode (Figure 2)?
14. There seems to be SrO segregation in Figure 3d. In general, the A-site segregation hampers the exsolution capability and could also negatively affect the electro-catalytic properties.

15. The degradation is observable in Figure 4c. In the case of SFNM+1.2Fe-red-GDC (Figure 4c), there seems to be about 30% degradation after 100-hour stability test.

17. In the abstract part, the authors stated as “In this study, we reveal that the formation of B-site vacancies in perovskite scaffold is the major contributor to the degradation of P-eNs; ...”. Instead of B-site vacancies in perovskite scaffold, A-site segregation and/or the formation of impurity phases (SrCO₃ and SrMoO₄) in Figure 4e appears to be much significant factor for the degradation properties. This part should be clearly elucidated.

18. The current density for SFNM+0.0Fe-red-GDC at 1.8 V much decreases just after 15 minutes. Hence, the production rate of CO and the Faradaic efficiency of CO should be much different between 60 min and 75 min.

19. How did the authors calculate the Faradaic efficiency of CO and the production rate of CO via gas chromatography measurements? I couldn't find clear explanations and equations on how the authors calculated the Faradaic efficiency of CO and the production rate of CO.

20. Did the authors calculate the Gibbs free energy for Fe-Ni alloy formation? Furthermore, is there any Gibbs free energy calculations on whether the Fe-Ni alloy formation is favorable at the surface or the bulk? This part should be also explained in this paper since the title in this work is “B-site supplement mechanism: A new approach to boosting stability of perovskites with exsolved nanoparticles”.

21. The overall thesis design and/or English proficiency should be much complemented in this work.

In these regards, “B-site supplement mechanism: A new approach to boosting stability of perovskites with exsolved nanoparticles” is not enough to be published in Nature Communications in my position. Please consider my comments and requests for the next submission.

Reviewer #3 (Remarks to the Author):

This work proposed B site supplement strategy to compensate the occurrence of B site vacancies caused by exsolution process before and during the operation, especially at high reduction potential. Mostly,

nanoparticle exsolution of B site cations in perovskite lattice leaves many B site vacancies and causes detrimental A site segregation, leading to the losses of B(n+1)-O-Bn+ pathway. In this work, authors have insisted that the guest Fe metal coated on Sr₂Fe_{1.3}Ni_{0.2}Mo_{0.5}O_{6-δ} (SFNM) would fill the B site vacancies by topotactic ion exchange (TIE) mechanism and mitigate the Sr segregation on the surface, thus enhancing structure stability and facilitating stable CO₂ electrolysis performance at high voltage. Several characterizations such as XRD, SEM, TEM, and elemental mappings were performed to verify the role of guest Fe, filling the B site vacancies during exsolution process and suppressing Sr segregation on perovskite surface. In potential step chronoamperometry, SFNM without guest Fe (SFNM+0.0Fe-red) showed significant decay of current density (j), whereas SFNM with guest Fe metals (SFNM+1.2Fe-red) showed stable j profiles, especially under high operating voltage (1.8V).

In this work, the results are noteworthy and would be useful to develop a highly active CO₂ electrolysis cathode catalyst with superior stability in SOEC system. Authors have provided detailed mechanisms of perovskite structure evolution during the exsolution and TIE-assisted exsolution, and most of conclusions are supported by the results. It might be suitable to be published in this journal after addressing below issues and comments:

1. In page 3~4, authors have mentioned that the exsolution process on the perovskite leaves many B site vacancies within perovskite bulk and thus A site segregation on the surface as below equation:

However, it is well known that these defects (i.e., B_(1-α) and AO) can be relieved by controlling A site non-stoichiometry on perovskite lattice as followed:

A-site deficient design has been adopted in numerous other researches of SOEC CO₂ cathodes, especially in [D. Neagu et al., Nat. Chem., 2013, 5, 11, 916-923]. It might be helpful to mention on your manuscript that what is the strong point of your strategy to coat additional guest Fe metals on perovskite surface to induce TIE-assisted exsolution compared to the one, which controls the A site non-stoichiometry of perovskite oxide.

2. In page 6, co-segregation energies of Ni, Fe and Mo at B site of SFNM were calculated by DFT method. From the calculation results, it seems to be reasonable to choose Ni as a guest ions to induce vigorous TIE-assisted exsolution because of its relatively low segregation energies (Ni: -1.46 eV, Fe: -1.06 eV, Mo: 2.26 eV). However, authors have chosen Fe as guest metals due to its relatively high redox stability (in page 5). Is there any additional calculation results that can support the choice of Fe as a guest ions?

3. Authors have insisted that TIE-assisted exsolution would decrease B site vacancies that could be produced during exsolution, thus help to maintain the electrical conductivity of bulk perovskite. XPS analysis, which showed more Fe²⁺-Fe³⁺ and Mo⁵⁺-Mo⁶⁺ charge pairs in SFNM+1.2Fe-red, was

conducted to support the opinions. Is it possible to measure electrical conductivities of SFNM+0.0Fe-red and SFNM+1.2Fe-red under reducing and operating atmosphere?

4. In Figure 3, two electrodes, SFNM+0.0Fe-red and SFNM+1.2Fe-red, have shown different behaviors in ohmic resistance when the external voltage is applied. It would be better to explain the change of ohmic resistance under applied voltage conditions for each electrodes.

5. Performance comparison tables of cathode catalysts with exsolved-metal nanoparticles from other researches under similar conditions would be proper to show novelty of your works. Below papers would be helpful to use.

S. Park et al., *Energy Technol.*, 2021, 9, 2100116.

L. Houfu et al., *J. Mater. Chem. A*, 2019, 7, 11967

J. Choi et al., *J. Mater. Chem. A*, 2021, 9, 8740

6. What is pre-reduction conditions of SFNM+1.2-red electrode before CO₂ electrolysis test?

This work is recommended for publication with major revisions.

Responses to Reviewers' Comments

Note: Line X and Page Y are abbreviated as LX and PY in the following responses for simplicity.

Reviewer #1 (Remarks to the Author):

In this manuscript, the authors developed highly stable exsolution SOEC for CO₂ reduction under high voltage (V > 1.6 V). The authors used recently developed topotactic ion exchange method for developing the highly stable exsolution SOEC. The authors' work has two important points. Firstly, they pointed out the poor stability issue of exsolution SOEC for CO₂ reduction under high voltage. Secondly, they proposed a solution for the poor stability issue of exsolution SOEC for CO₂ reduction by applying the recently developed topotactic ion exchange method. However, they ignored the reason why should we use the exsolved SOEC instead of the other conventional perovskite SOEC which have much higher current density with higher stability. Also, for revealing the degradation mechanism, more detailed characterizations are required. Therefore, I recommend following major revisions should be clear before this manuscript will be published.

Authors' response:

We appreciate the reviewer's valuable comments and recommendation for publication. We agree with the reviewer's viewpoint that the perovskites with exsolved nanoparticles have great potentials for the large-scale energy storage and conversion in comparison to their nanoparticle-free counterparts^{1,2}. However, the exsolved SOECs still suffer from severe instability under the synergistic effect of the high voltages and CO₂/CO mixed atmospheres during CO₂ electrocatalysis³. As the reviewer suggested, we have added the electrochemical performances of the Sr₂Fe_{1.3}Ni_{0.2}Mo_{0.5}O₆-base SOEC without exsolution and the additional post-mortem characterizations in the revised manuscript to reveal the degradation mechanism.

Comment #1. Without exsolution, SFMM (Sr₂Fe_{1.4}Mn_{0.1}Mo_{0.5}O₆) perovskite based SOEC showed two times higher current density than SFNM (Sr₂Fe_{1.3}Ni_{0.2}Mo_{0.5}O₆) based exsolution SOEC in this manuscript [J. Mater. Chem. A, 2019, 7, 11967]. A. This manuscript should discuss the essential advantages of exsolved nanoparticles for SOEC application and compare its performance (current density, CO conversion) with other state-of-art SOEC for CO₂ reduction.

Authors' response to Comment #1A:

We thank the reviewer for this excellent suggestion. As discussed in the Introduction (L2-8, P3) in the revised manuscript, the exsolved nanoparticles on the perovskite scaffold can significantly increase the number of the reaction sites and enhance the catalytic activity for CO₂ electrocatalysis in SOEC^{1,4}. Furthermore, the nanoparticles grown by exsolution have higher resistance to agglomeration and hydrocarbon coking at the

elevated temperatures⁵. Therefore, the perovskites with exsolved nanoparticles (P-eNs) provide an appealing platform for the large-scale energy storage and conversion technology compared to their nanoparticle-free counterparts.

It is well known that many factors such as the types of cathode, electrolyte and anode could contribute to the higher current density of Sr₂Fe_{1.4}Mn_{0.1}Mo_{0.5}O₆ perovskite based SOEC in reference [J. Mater. Chem. A, 2019, 7, 11967] than Sr₂Fe_{1.3}Ni_{0.2}Mo_{0.5}O₆ (SFNM)-based exsolved SOEC in our work. In the reference [J. Mater. Chem. A, 2019, 7, 11967], LSCF-SDC/LSGM/SFMM0.1-SDC was assembled for electrochemical experiments, while LSCF-GDC/GDC/YSZ/GDC/ SFNM+xFe (x =0.5, 0.8, 1.2)-GDC were fabricated in our study. It is known that the LSGM-supported electrolysis cells perform better than the YSZ-supported ones under the same condition. Therefore, the direct comparison of the cell current density is only meaningful when all the other variables are kept identical. For example, under the same condition, the current density of SFNM+1.2Fe-red-GDC is higher than that of SFNM+0.0Fe-red-GDC, this indicates that the catalytic activity can be enhanced by B-site supplement strategy.

Alternatively, the total polarization resistance (R_p) may be a better indicator for evaluating the catalytic activities of cathode materials since R_p is dominated by the cathode reaction⁶. The Nyquist plots and the fitted R_s , R_p (equivalent circuit model LR(Q_HR_H)(Q_LR_L)) of SFNM+1.2Fe-red-GDC at 800 °C are provided in Fig. R1 (also Supplementary Fig. 11) and Table R1 (also Supplementary Table 3). The current density, polarization resistance and CO productivity of SFNM+1.2Fe-red-GDC in our work are compared with those of other state-of-art P-eNs-based SOECs, as shown in Table R2 (also Supplementary Table 4). SFNM+1.2Fe-red-GDC shows the lowest R_p at high voltages (≥ 1.6 V) among the state-of-art P-eNs-based cathode candidates as listed in Table R2, indicative of the higher catalytic activities of SFNM+1.2Fe-red-GDC at higher negative potentials. In addition, the short-term stability performances of SFNM+0.0Fe-red-GDC and SFNM+1.2Fe-red-GDC are compared with that of other state-of-art P-eNs-based SOECs, as shown in Table R3 (also Supplementary Table 6). SFNM+1.2Fe-red-GDC shows the competitive short-term stability at the voltages ≥ 1.6 V compared with the SFNM+0.0Fe-red-GDC and other state-of-art P-eNs-based SOECs listed in Table R3.

Most importantly, SFNM with varied B-site occupation were selected as the prototypes in our study to highlight the significance of robust perovskite scaffold for enhancing the reactivity and stability of the P-eNs. The faster reaction kinetics and more stable performances of SFNM+1.2Fe-red-GDC in comparison to SFNM+0.0Fe-red-GDC, especially at high voltages, confirm the feasibility of the B-site supplement strategy. It suggests that this strategy would be applied to other P-eNs listed in the Tables R2-R3 to improve their electrochemical performances for CO₂ electrocatalysis and more broadly, for other energy storage and conversion systems.

Figure R1 (also Supplementary Figure 11). Nyquist plots of SFNM+1.2Fe-red-GDC at 800 °C (Equivalent circuit model $LR(Q_H R_H)(Q_L R_L)$).

Table R1 (also Supplementary Table 3). EIS fitting values (R_s , R_H , R_L) of SFNM+1.2Fe-red-GDC at 800 °C.

Potential (V)	R_s ($\Omega \text{ cm}^2$)	R_H ($\Omega \text{ cm}^2$)	R_L ($\Omega \text{ cm}^2$)	R_p ($\Omega \text{ cm}^2$)
1	0.7375	0.0714	0.6480	0.7194
1.2	0.6901	0.1796	0.1871	0.3667
1.4	0.6847	0.1254	0.1090	0.2344
1.6	0.6743	0.0910	0.1064	0.1974
1.8	0.6441	0.0751	0.0700	0.1451

Table R2 (also Supplementary Table 4). Comparison of the current density, polarization resistance, CO productivity with other state-of-art SOECs for CO₂ electrocatalysis.

Perovskite cathode (Exsolved nanoparticles)	Electrolyte Anode	Current density at 1.6 V (A cm ⁻²)	Polarization resistance (Ω cm ²)	CO productivity (mL min ⁻¹ cm ⁻²)	Ref.
			0.14/0.10/0.08 (1.4/1.6/1.8 V and 850 °C)		
SFNM+1.2Fe-red (Fe-Ni alloy)	YSZ LSCF- GDC	1.12 (850 °C)	0.23/0.19/0.14 (1.4/1.6/1.8 V and 800 °C)	6.43 (1.6 V and 850 °C)	This work
Sr ₂ Fe _{1.58} Mo _{0.5} O _{6-δ} (Metallic Fe)	LSGM LSM-SDC	0.95 (850 °C)	0.29 (1.8 V and 850 °C)	-	[7]
(La _{0.2} Sr _{0.8}) _{0.85} Ti _{0.8} Cr _{0.1} Ni _{0.1} O _{3-δ} (Metallic Ni)	LSGM LSM-SDC	0.68 (850 °C)	0.43 (1.6 V and 850 °C)	3.27 (1.6 V and 850 °C)	[8]
Sr ₂ Fe _{1.35} Mo _{0.45} Co _{0.2} O _{6-δ} (Co-Fe alloy)	LSGM BSCF-GDC	1.20 (800 °C)	0.28 (1.6 V and 800 °C)	ca. 8.50 (1.6 V and 800 °C)	[9]
(La _{0.2} Sr _{0.8}) _{0.95} Ti _{0.85} Mn _{0.1} Ni _{0.05} O _{3-δ} (Metallic Ni)	YSZ LSM	0.54 (800 °C)	0.51 (1.6 V and 800 °C)	3.67 (1.6 V and 800 °C)	[10]
Sr ₂ Fe _{1.35} Mo _{0.45} Ni _{0.2} O _{6-δ} (FeNi ₃ alloy)	LSGM LSCF-GDC	0.93 (800 °C)	0.20 (1.6 V and 800 °C)	ca. 7.50 (1.6 V and 800 °C)	[3]
La _{1.2} Sr _{0.8} Mn _{0.4} Fe _{0.6} O _{4-a} (Metallic Fe)	LSGM LSCF-GDC	1.43 (1.5 V, 800 °C)	0.326 (1.5 V and 800 °C)	-	[11]
La _{0.6} Ca _{0.4} Fe _{0.8} Ni _{0.2} O _{3-δ} (Fe-Ni alloy)	YSZ La _{0.6} Ca _{0.4} Fe _{0.8} Ni _{0.2} O _{3-δ}	1.18 (800 °C)	0.399 (1.3 V and 800 °C)	-	[12]
Sr ₂ Fe _{1.25} Cu _{0.25} Mo _{0.5} O _{6-δ} (Fe-Cu alloy)	LSGM LSCF-SDC	5.40 (850 °C)	0.46 (ca. 0.92 V and 850 °C)	12.80 (1.4 V and 800 °C)	[13]

Note: Some values of the current density at 1.6 V of the cited literatures are determined by the Digitizer function of Origin software.

Table R3 (also Supplementary Table 6). Comparison of the short-term stability with the state-of-art P-eNs for pure CO₂ electrocatalysis in SOEC.

Perovskite cathode (Exsolved nanoparticles)	Current density loss rate at 1.6 V (A cm ⁻² min ⁻¹)	Current density loss rate at 1.8 V (A cm ⁻² min ⁻¹)	Ref.
SFNM+1.2Fe-red (Fe-Ni alloy)	0.003 (850 °C, 15 min)	0.001 (850 °C, 15 min)	This work
SFNM+0.0Fe-red (Fe-Ni alloy)	0.008 (850 °C, 15 min)	0.040 (850 °C, 15 min)	This work
Sr ₂ Fe _{1.58} Mo _{0.5} O _{6-δ} (Metallic Fe)	0.002 (850 °C, 15 min)	0.002 (850 °C, 15 min)	[7]
(La _{0.65} Sr _{0.3} Ce _{0.05}) _{0.9} (Cr _{0.5} Fe _{0.5}) _{0.85} Ni _{0.15} O _{3-δ} (Fe-Ni alloy)	0.022 (850 °C, 15 min)	0.030 (850 °C, 15 min)	[14]
Sr ₂ Fe _{1.35} Mo _{0.45} Ni _{0.2} O _{6-δ} (FeNi ₃ alloy)	0.005 (800 °C, 20 min)	-	[3]
(La _{0.2} Sr _{0.8}) _{0.95} Ti _{0.85} Mn _{0.1} Ni _{0.05} O _{3-δ} (Metallic Ni)	0.004 (800 °C, 15 min)	0.005 (800 °C, 15 min)	[10]

Note: The values of the current density in the short-term stability of the cited literatures are determined by the Digitizer function of Origin software.

We have revised the manuscript (L4-9, P13 and L1-3, P19) and Supplementary Information (SI) (P35 and P37-38) accordingly, as shown below:

Revised manuscript (L4-9, P13): “In addition, the Nyquist plots and the fitted R_p of SFNM+1.2Fe-red-GDC at 800 °C are obtained (Supplementary Fig. 11 and Supplementary Table 3). It shows the lowest R_p at high voltages (≥ 1.6 V) among the state-of-art P-eNs-based cathode candidates as listed in Supplementary Table 4, indicative of the higher catalytic activities of SFNM+1.2Fe-red-GDC at higher negative potentials.”

(L1-3, P19): “SFNM+1.2Fe-red-GDC shows the competitive short-term stability at the voltages ≥ 1.6 V compared with the SFNM+0.0Fe-red-GDC and other state-of-art P-eNs-based SOECs (Supplementary Table 6).”

Revised SI (P37-38): “The short-term stability performances of SFNM+0.0Fe-red-GDC and SFNM+1.2Fe-red-GDC are compared with those of other state-of-art P-eNs-based SOECs, as shown in Supplementary Table 6. In particular, Sr₂Fe_{1.58}Mo_{0.5}O_{6-δ} with the exsolved Fe nanoparticles in reference 27 shows the comparable stability

to SFNM+1.2Fe-red-GDC, which can be attributed to the super-stoichiometric Fe occupation at the B-site. It leads to the sufficient B-site occupation and robust perovskite structure after the reduction, which is consistent with our experimental results and the B-site supplement mechanism proposed in this work.”

B. Only two exsolution SOECs were compared in this manuscript. If the authors want to clearly show the degradation by exsolution, they need to additionally show the performance and stability of SFNM ($\text{Sr}_2\text{Fe}_{1.3}\text{Ni}_{0.2}\text{Mo}_{0.5}\text{O}_6$) based SOEC without exsolution.

Authors' response to Comment #1B:

Thanks to the reviewer for this constructive comment. As suggested by the reviewer, the electrochemical performances and post-mortem characterizations of $\text{Sr}_2\text{Fe}_{1.3}\text{Ni}_{0.2}\text{Mo}_{0.5}\text{O}_6$ (SFNM) based SOEC without exsolution have been added for comparison [Figs. R2-R5 (also Supplementary Figs. 20-23 in the revised SI)]. As can be seen from the current density-voltage (j -V) curves at 800 and 850 °C, SFNM-based SOEC without exsolution (SFNM-oxi-GDC) exhibits the lower j than the j values of SFNM+0.0Fe-red-GDC and SFNM+1.2Fe-red-GDC under the same condition. Additionally, the SFNM-oxi-GDC shows the similar short-term/long-term stability profiles as the SFNM+0.0Fe-red-GDC. In term of the short-term stability, the SFNM-oxi-GDC shows satisfactory stability at 1.0-1.4 V, while the SOEC experiences rapid degradation when the voltage exceeds 1.6 V. For the long-term stability, the SFNM-oxi-GDC shows a visible degradation after 60 h electrolysis, and finally destabilized. The SEM, secondary electron-scanning TEM (SE-STEM) and STEM-EDS results confirm that partial Ni elements segregate from the SFNM bulk and the newly born fibrous phases appear on the cathode surface. Meanwhile, the Raman feature peaks of SrCO_3 and carbon with lattice defect (D band) suggest that the Ni element in-situ exsolves from the SFNM bulk and subsequently result in the carbon fiber growth during CO_2 electrolysis, demonstrating the structural decomposition under the synergistic effect of 1.6 V and 850 °C. The electrochemical performances and stability results of SFNM-based SOEC without exsolution consolidate the degradation mechanisms discussed in our manuscript.

Figure R2 (also Supplementary Figure 20). Comparison of electrochemical performances of SFNM-oxi-GDC with that of SFNM+0.0Fe-red-GDC and SFNM+1.2Fe-red-GDC. I-V curves of SFNM-oxi-GDC, SFNM+0.0Fe-red-GDC and SFNM+1.2Fe-red-GDC at (a) 800 °C and (b) 850 °C. Comparison of (c) Short-term stability and (d) long-term stability performances at 1.6 V and 850 °C of SFNM-oxi-GDC with that of SFNM+0.0Fe-red-GDC and SFNM+1.2Fe-red-GDC. (Note: the break point at 47 h of the long-term stability curve of SFNM-oxi-GDC in Figure R2d is caused by updating the CO₂ cylinder.)

Figure R3 (Supplementary Figure 21). SEM images of cathode surface microstructures of SFNM-oxi-GDC after long-term stability at 1.6 V and 850 °C.

Figure R4 (Supplementary Figure 22). SE-STEM and STEM-EDS images of cathode surface microstructures of SFNM-oxi-GDC after long-term stability at 1.6 V and 850 °C.

Figure R5 (also Supplementary Figure 23). Raman spectrum collected from cathode surface of SFNM-oxi-GDC after the long-term stability test at 1.6 V and 850 °C.

We have revised the manuscript (L11-19, P22 and L1-5, P23) and SI (P26-27) accordingly, as shown below:

Revised manuscript (L11-19, P22 and L1-5, P23): “To verify the exsolution scenario and the structural decomposition during the electrolysis process, the SFNM without pre-reduction treatment was fabricated as the

composite cathode material (SFNM-oxi-GDC) for the electrochemical testing (Supplementary Fig. 20) and the post-mortem characterizations (Supplementary Figs. 21-23). As shown in Supplementary Fig. 20d, the SFNM-oxi-GDC shows the similar long-term stability profile as the SFNM+0.0Fe-red-GDC. The SEM, SE-STEM and STEM-EDS results confirm that partial Ni elements segregate from the SFNM bulk and the newly born fibrous phases appear on the cathode surface (Supplementary Figs. 21-22). Meanwhile, the Raman feature peaks of SrCO₃, carbon with lattice defect (D band) can be observed (Supplementary Fig. 23), suggesting that the Ni element in-situ exsolves from the SFNM bulk and subsequently results in the carbon fiber growth during CO₂ electrolysis. This can be ascribed to the lower co-segregation energy of Ni in SFNM, which causes the structural decomposition under the synergistic effect of 1.6 V and 850 °C.”

Revised SI (P26-27): “To further clarify the degradation by exsolution, the catalytic activity and stability performances of Sr₂Fe_{1.3}Ni_{0.2}Mo_{0.5}O₆ (SFNM) based SOEC without exsolution have been evaluated for comparison (Supplementary Fig. 20). As can be seen from the current density-voltage (*j*-V) curves at 800 and 850 °C, SFNM-based SOEC without exsolution (SFNM-oxi-GDC) exhibits the lower *j* than those of the SFNM+0.0Fe-red-GDC and SFNM+1.2Fe-red-GDC under the same condition. Additionally, the SFNM-oxi-GDC shows the similar short-term/long-term stability profiles as the SFNM+0.0Fe-red-GDC. In term of the short-term stability, the SFNM-oxi-GDC shows satisfactory stability at 1.0-1.4 V, while the SOEC experiences a rapid degradation when the voltage exceeds 1.6 V. For the long-term stability, the SOEC experiences a significant degradation at the initial stage, followed by a steady *j*-V profile. In fact, the *j* also experiences a slight increase, peaking at 31 h with a maximum *j* of 0.18 A cm⁻². Then, the SFNM-oxi-GDC shows a visible degradation after 60 h electrolysis, and finally destabilized.”

Comment #2. The HR-TEM results for SFNM+0.0Fe-red and SFNM+0.0Fe-red were clearly showed the difference between the conventional and TIE exsolved nanoparticles. What are the morphologies like for each of them after stability tests?

Authors' response to Comment #2:

As suggested by the reviewer, the SEM, SE-STEM and STEM-EDS images are added to characterize the cathode surface morphologies of SFNM+0.0Fe-red-GDC and SFNM+1.2Fe-red-GDC after the long-term stability tests (Figs. R6 and R7 [also Figs. 5d-g in the revised manuscript]). The obvious morphology changes and coarsening of exsolved particles can be observed on both substrates, and the coarsening of surface nanoparticles on SFNM+1.2Fe-red-GDC is less severe than that on SFNM+0.0Fe-red-GDC (Fig. R6). Furthermore, more pronounced Sr- and Mo-derived phase separation in the form of irregular particles appear on the surfaces of SFNM+0.0Fe-red-GDC, (Fig. R7), suggesting that SFNM+0.0Fe-red-GDC undergoes the more significant phase decomposition than SFNM+1.2Fe-red-GDC under the synergistic effect of the high voltage and CO₂/CO mixed atmosphere during the CO₂ electrolysis.

Figure R6 (also Figs. 5d-e). SEM images of cathode surface microstructures of (a) SFNM+0.0Fe-red-GDC and (b) SFNM+1.2Fe-red-GDC after the long-term stability at 1.6 V and 850 °C.

Figure R7 (also Figs. 5f-g). SE-STEM and STEM-EDS images of cathode surface microstructures of (a) SFNM+0.0Fe-red-GDC and (b) SFNM+1.2Fe-red-GDC after long-term stability at 1.6 V and 850 °C.

We have revised the manuscript (L1-11, P21) accordingly as shown below:

Revised manuscript (L1-11, P21): “After the long-term stability tests, the significant coarsening and decrease in the population of nanoparticles can be observed on the cathode surfaces of SFNM+0.0Fe-red-GDC and SFNM+1.2Fe-red-GDC (Figs. 5d-e). Both cathodes were scratched from the cells for the element distribution detection by secondary electron-scanning TEM (SE-STEM) and STEM-EDS. More pronounced Sr- and Mo-derived phase separation in the form of irregular particles appear on the surfaces of SFNM+0.0Fe-red-GDC, (Figs. 5f-g), suggesting that SFNM+0.0Fe-red-GDC undergoes the more significant phase decomposition than SFNM+1.2Fe-red-GDC under the synergistic effect of the high voltage and CO₂/CO mixed atmosphere during the CO₂ electrolysis.”

Comment #3. (minor comment) The concept of vacancy trapping in fig. 3e needs experimental or theoretical verifications. Diffusion barrier calculation using DFT would be best to verify it.

Authors' response to Comment #3:

Thanks for this suggestion. We have verified the concept of oxygen vacancy trapping and compared the oxygen ion diffusion capacity by calculating the binding energy of $V_{Fe}'' - V_O''$ defect pair (Fig. R8 (also Supplementary Fig. 15 in the revised SI) and Table R4 [also Supplementary Table 5 in the revised SI]) and by conducting the electrical conductivity relaxation experiments (ECR) (Fig. R9 (also Fig. 4a in the revised manuscript)). To simplify the model, we assume that all Ni elements are exsolved from the SFNM substrate after the reduction. Therefore, the binding energy between the Fe-site vacancy and the oxygen vacancy is calculated in the Sr₂Fe_{1.5}Mo_{0.5}O₆ (SFM). The three defective configurations and corresponding formation energy are provided in Fig. R8 (also Supplementary Fig. 15) and Table R4 (also Supplementary Table 5). The calculated binding energy of $V_{Fe}'' - V_O''$ is -1.73 eV indicates that the additional association barrier needs to be overcome to achieve the jumping of oxygen vacancies¹⁵. In addition, the bulk diffusion constant (D_{chem}) of SFNM+1.2Fe-red obtained by fitting the ECR curve is $2.716 \times 10^{-5} \text{ cm}^2 \text{ s}^{-1}$, much higher than $1.208 \times 10^{-5} \text{ cm}^2 \text{ s}^{-1}$ of SFNM+0.0Fe-red¹⁶. It suggests that there are more available oxygen ion transfer pathways in SFNM+1.2Fe-red, which lead to the higher oxygen ion conductivity and lower charge transport resistance.

Figure R8 (also Supplementary Figure 15). Three defective configurations of (a) $V_{Fe}'' - V_O''$, (b) V_{Fe}'' and (c) V_O'' in SFM. The gray balls represent Sr atoms, the red balls represent Fe atoms, the pink balls represent Mo atoms, the blue balls represent O atoms.

Table R4 (also Supplementary Table 5). The corresponded energies of three defective configurations and perfect SFM.

Configuration	Energy (eV)
$V_{Fe}'' - V_O''$ in SFM	-258.45
V_{Fe}'' in SFM	-262.85
V_O'' in SFM	-268.81
Perfect SFM	-274.94

Figure R9 (also Figure 4a). Normalized electrical conductivity relaxation curves for SFNM+0.0Fe-red-GDC and SFNM+1.2Fe-red-GDC at 850 °C with switching of the gas stream from 5% H₂/95% Ar to 50% CO₂/50% CO.

We have revised the manuscript (L11-19, P16 and L1-3, P17) and SI (P20-21) accordingly, as shown below:

Revised manuscript (L11-19, P16 and L1-3, P17): “The calculated binding energy of the $V_{Fe}'' - V_O^{\bullet\bullet}$ defect pair is - 1.73 eV (Supplementary Fig. 15 and Supplementary Table 5), indicating that the additional association barrier needs to be overcome to achieve the jumping of oxygen vacancies¹⁵. In contrast, the full occupation of B-sites on SFNM+1.2Fe-red allows the elimination of constraint of B-site defect to surrounding oxygen vacancies. The electrical conductivity relaxation (ECR) experiments for SFNM+0.0Fe-red and SFNM+1.2Fe-red were carried out to study the oxygen ion bulk diffusion^{17,18} (Fig. 4a). The bulk diffusion constant (D_{chem}) of SFNM+1.2Fe-red obtained by fitting the ECR curve is $2.716 \times 10^{-5} \text{ cm}^2 \text{ s}^{-1}$, much higher than $1.208 \times 10^{-5} \text{ cm}^2 \text{ s}^{-1}$ of the SFNM+0.0Fe-red¹⁶. It suggests that there are more available oxygen ion transfer pathways in SFNM+1.2Fe-red, which lead to the higher oxygen ion conductivity and lower charge transport resistance.”

Revised SI (P20-21): “It has been reported that the association (trapping) of oxygen vacancy caused by other defects in perovskite lattice is detrimental to oxygen ion diffusion^{15,19,20}. And it is widely accepted that the association of oxygen vacancy can be determined by calculating the binding energy between the two adjacent point defects^{19,21} ($V_{Fe}'' - V_O^{\bullet\bullet}$ pair in our manuscript). The binding energy (E_{bind}) can be calculated by following equation²⁰:

$$E_{bind} = E_{defects\ pair} - \left(\sum_{component} E_{isolated\ defect} \right)$$

In our case,

$$E_{bind} = E_{(V_{Fe}''-V_O'')} - (E_{(V_{Fe}'')} + E_{(V_O'')})$$

To simplify the model, we assume that all Ni elements are exsolved from the SFNM substrate after the reduction. Therefore, the binding energy between the Fe-site vacancy and the oxygen vacancy is calculated in the $Sr_2Fe_{1.5}Mo_{0.5}O_6$ (SFM). The three defective configurations and the corresponding formation energy are provided in Supplementary Fig. 15 and Supplementary Table 5. As a result, the calculated binding energy of $V_{Fe}'' - V_O''$ is -1.73 eV. It indicates that the existence of B-site vacancies would hinder the transport of oxygen vacancies, thus resulting in the reduction of the free oxygen vacancy population.”

Comment #4. (minor comment) The caption of Figure 4d and 4e was reversed.

Authors' response to Comment #4:

We thank the reviewer for pointing this out. The captions of Figs. 4d and 4e have been corrected in Figs. 6a and 6c in the revised manuscript.

Comment #5. (minor comment) The number of figures is too small. Move important supplementary figures to main figures.

Authors' response to Comment #5:

As suggested by the reviewer, we have moved the following spectra from SI to the revised manuscript including the FTIR spectra of CO₂ chemisorption and physisorption for SFNM+0.0Fe-red and SFNM+1.2Fe-red at 600 °C (Fig. 3d), the Raman spectra collected from cathode surface of SFNM+0.0Fe-red-GDC whose stability test was interrupted before entering the Phase II (Fig. 6b). In the revised manuscript, we have also added the normalized electrical conductivity relaxation curves (Fig. 4a), the temperature-dependent electrical conductivities (Fig. 4d), the SEM, SE-STEM and STEM-EDS images of cathode surface microstructure of SFNM+0.0Fe-red-GDC and SFNM+1.2Fe-red-GDC after long-term stability tests (Figs. 5d-5g).

Reviewer #2 (Remarks to the Author):

In this work titled “B-site supplement mechanism: A new approach to boosting stability of perovskites with exsolved nanoparticles”, the authors describe the effect of B-site supplement (or topotactic ion exchange) of Fe nitrate guest in double perovskite $\text{Sr}_2\text{Fe}_{1.3}\text{Ni}_{0.2}\text{Mo}_{0.5}\text{O}_{6-\delta}$ (SFMN). Unfortunately, I have to say that the electrochemical part of this paper is poor and there are too many flaws in this manuscript. Thus, I can't recommend this manuscript to be published in Nature Communications. Some major comments and questions are listed as below:

Authors' response:

We thank the reviewer very much for carefully reviewing our manuscript and providing insightful comments. To address the concerns raised by the reviewer, we have carefully revised the manuscript according to the reviewer's suggestions, as shown in the following point-to-point responses. We believe that the tendency of our electrochemical experimental results are in line with the fundamental principles of CO_2 electrocatalysis in SOEC and the overall quality are consistent with previous work, such as the typical nonlinear profile at the initial region of j - V curve (Question #4), the open-circuit voltage as a function of the reactant gas composition (Question #4), the values of Faradic efficiency and CO productivity (Question # 18). Our detailed responses to each comment are listed below.

Comment #1. The main point of this work is to promote topotactic ion exchange (TIE) of Fe guest source in $\text{Sr}_2\text{Fe}_{1.3}\text{Ni}_{0.2}\text{Mo}_{0.5}\text{O}_{6-\delta}$ (SFMN) double perovskite. In my opinion, this work lacks novelty since the topotactic exchange has been already reported in reference 25 and 26 in the authors' work. Moreover, Figure 1c and 1d in this work is too much similar with the Figure 1a and 1b in the reference 26. Reference 26: Joo, S. et al. Cation-swapped homogeneous nanoparticles in perovskite oxides for high power density. Nat. Commun. 10, 697 (2019).

Authors' response to Comment #1:

It might be possible that the novelty of our work might not have been well perceived and we would like to respectfully reiterate that the significance of our work is to provide the instruction for designing P-eNs with high stability for CO_2 reduction and more broadly, for other energy storage and conversion systems. This is of critical importance because the rapid degradation of P-eNs for CO_2 electrocatalysis at high voltages greatly impedes their practical applications and this issue has not been well elucidated so far. In this study, we firstly reveal that the structural instability of perovskite scaffold may be the main reason for the rapid degradation of P-eNs for CO_2 electrocatalysis at high voltages. Then, the B-site supplement strategy is proposed to yield a robust perovskite scaffold, which leads to higher catalytic activity and stability of the P-eNs at high voltages. Lastly, the detailed degradations of P-eNs with and without B-site supplement are demonstrated. Therefore, the innovation

in our work lies in enhancing the stability of P-eNs for CO₂ electrocatalysis in SOEC by reinforcing the structural stability of perovskite scaffold rather than the investigation on topotactic ion exchange (TIE) technology, which is a synthetic method for us to verify the feasibility of B-site supplement strategy.

In the past few decades, exsolution has received extensive attentions as an emerging technique for in-situ growing nanoparticles, we have summarized the landmark work of exsolution technology on perovskites in the past ten years^{5,22,23 24,25,26,27,28,29,30} in Fig. R10. Most studies have focused on the enhancement of catalytic activity by promoting the exsolution of the nanoparticles and tuning the features of exsolved nanoparticles, while relatively few studies on the stability of P-eNs.

Figure R10. The landmark studies of the exsolution on perovskites in the past decade.

Currently, the researches on the stability of P-eNs are mainly focused on the unique socketed interface^{5,31}. Neagu et al. demonstrated that the exsolved nanoparticles are closely embedded into the parent perovskite scaffold, resulting in the high resistance to agglomeration and hydrocarbon coking⁵, thus maintaining the catalytic activity of nanoparticles during the long-term operation. However, the stability of P-eNs is still unsatisfactory for CO₂ electrocatalysis at high voltages and has not been resolved so far. The significant current density loss has been mentioned in previous studies and is suggested to be caused by the carbon dioxide starvation at high reaction rates, coke formation by CO disproportionation/ electrochemical reduction and mass transport limitation^{32,33,34}.

To address this challenge, we focus on developing a strategy to inhibit the structural evolution of the perovskite scaffold. Taking into account the promoting effect of the applied potentials on exsolution, the continuous exsolution occurs at high voltages²³, which may cause the accumulation of B-site vacancies and the consequent phase decomposition of perovskite substrate. Therefore, we infer that the structural stability would be greatly

improved after supplementing the B-site vacancies with the redox-stable cations. To achieve B-site supplement for boosting the stability of P-eNs, there are two synthesis methods to prepare the P-eNs with sufficient B-site occupation: the A-site deficiency/B-site excess and the topotactic ion exchange. However, it has been reported that pure $\text{Sr}_{1.9}\text{Fe}_{1.3}\text{Ni}_{0.2}\text{Mo}_{0.5}\text{O}_{6-\delta}$ cannot be obtained with 5% mol Sr-site deficiency³⁵, while almost all Ni elements can be exsolved from SFNM after reduction treatment at 800 °C and a 5% H_2/N_2 atmosphere for 2 h. In this case, less than 5% mol Sr-site deficiency cannot effectively retain the occupation of the B-site vacancies after exsolution. Therefore, we choose the ion exchange assisted exsolution method to achieve B-site supplement of perovskite scaffold of reduced SFNM.

Although the ion exchange method is not the novelty of this work and we only used the ion exchange method to enhance structure stability, thereby to facilitate stable CO_2 electrolysis performance at high voltage, we have optimized the precursor deposition method for initiating this synthesis route. The simple combination of lyophilization and TIE not only achieves the B-site supplement, but also shows great application prospects in decorating the perovskite scaffold of P-eNs based on the demand of specific application.

Figs. 1c and 1d are the schematic diagrams based on the specific SFNM double perovskite structure, intended to help readers gain a better understanding on the DFT calculation and the influence of B-site supplement on the structure of the reduced SFNM perovskite scaffold. Because the similar synthetic method was used as in the Reference 26, it is reasonable that the schematic representations would be similar. We believe that combining 2D and 3D schematics in a generally recognized way is helpful for readers to understand the work of the DFT and synthetic parts.

We have revised the manuscript (L10-19, P4) accordingly, as shown below:

Revised manuscript (L10-19, P4): “In this study, the promising double perovskite $\text{Sr}_2\text{Fe}_{1.3}\text{Ni}_{0.2}\text{Mo}_{0.5}\text{O}_{6-\delta}$ (SFNM) was selected as a prototype example to elaborate the effects of the structure evolution of the perovskite scaffold on the stability of P-eNs³. Either controlling the A-site deficiency or implementing the topotactic ion exchange (TIE) is expected to be a pathway to regulate the concentration of the B-site vacancies in the reduced SFNM (Eqs. 2 and 3)^{22,26,36,37}. However, the limited Sr-site deficiency (less than 5% mol) in the SFNM makes it fail to refill the B-site vacancies after the exsolution by controlling the A-site deficiency³⁵. Therefore, the TIE-assisted exsolution was employed to fine-tune the B-site occupation of perovskite scaffold while promoting the formation of nanoparticles.”

Comment #2. In Supplementary figure 2, there seems to be unknown peaks in my position. I recommend designating all the peaks for Supplementary Figure 2 to remove confusion. In addition, why did the authors only put SrMoO_4 as the PDF card for air-sintered SFM? If SrMoO_4 exists, where are the Fe source?

Authors' response to Comment #2:

We thank the reviewer for pointing this out. As suggested by the reviewer, all the characteristic peaks of Fig. R11 (also Supplementary Fig. 2) have been designated in the revised manuscript. In addition, it is expected that the trace amount of secondary phase SrMoO_4 will be detected in the air-sintered SFM³⁸, so the corresponding PDF card has been added. From an electroneutrality viewpoint, the high oxidation state of Fe (+2/+3) and Mo (+5/+6) at the B-site of SFM would result in an unbalanced positive charge, thus leading to a limited solubility of Mo at B-site of SFM. Consequently, Mo exists as a secondary phase SrMoO_4 by combining with Sr while Fe is completely dissolved in the perovskite lattice. Instead, the pure SFM can be synthesized in a reducing environment by reducing the B-site average valence³⁸.

Figure R11 (also Supplementary Figure 2). XRD pattern of SFNM+1.5Fe-red.

Comment #3. The same or similar host material has been already reported (reference 27 and the below □ additional reference). Reference 27: Lv, H. et al. In Situ Investigation of Reversible Exsolution/Dissolution of CoFe Alloy Nanoparticles in a Co-Doped $\text{Sr}_2\text{Fe}_{1.5}\text{Mo}_{0.5}\text{O}_{6-\delta}$ Cathode for CO_2 Electrolysis. *Adv. Mater.* 32, e1906193 (2020) Additional reference: Wang, Y., Liu, T., Li, M., Xia, C., Zhou, B., & Chen, F. (2016). Exsolved Fe–Ni nano-particles from $\text{Sr}_2\text{Fe}_{1.3}\text{Ni}_{0.2}\text{Mo}_{0.5}\text{O}_6$ perovskite oxide as a cathode for solid oxide steam electrolysis cells. *Journal of Materials Chemistry A*, 4(37), 14163-14169.

Authors' response to Comment #3:

It is true that similar host material has already been reported by other researchers. The main achievement of our work is to propose and verify the effectiveness of B-site supplement strategy for stability enhancement of

the P-eNs. For this purpose, a typical P-eNs should be selected as a prototype for elucidation. $\text{Sr}_2\text{Fe}_{1.5}\text{Mo}_{0.5}\text{O}_{6-\delta}$ (SFM) has great potentials in solid oxide cells due to its satisfactory redox stability and conductivity under both reducing and oxidizing conditions^{39,40,41}. SFM with Fe-Ni alloy nanoparticles have been regarded as the promising candidate for CO_2 reduction in SOEC³. Like other P-eNs, the reduced $\text{Sr}_2\text{Fe}_{1.3}\text{Ni}_{0.2}\text{Mo}_{0.5}\text{O}_{6-\delta}$ (SFNM) with exsolved Fe-Ni nanoparticles suffers significant degradation for CO_2 electrocatalysis at high voltages, which greatly limits its further application. Therefore, SFNM was selected as a prototype example.

Comment #4. Why is there non-linearity from the voltage range of 0 to about 0.8 V in the I-V curves in Supplementary Figure 9? Also, why is the open-circuit voltage too low?

Authors' response to Comment #4:

The change in the slope of the j -V from the non-linearity to linearity at approximately 0.8 V separates the curve into two major cell processes. The nonlinear part (below 0.8 V) is the activation region, i.e., the SOEC mainly undergoes activation polarization and the electrolysis process is controlled by the reduction reaction on the cathode and the oxidation reaction on the anode; while the linear segment is ohmic loss region, the electrolysis process is controlled by the resistance to the flow of current through the cell when the applied potential enters into ohmic region^{42,43,44,45,46,47}.

The measured open-circuit voltage (OCV) is expected to be slightly lower than the theoretical reversible voltage (E_{rev}) required for CO_2 splitting. E_{rev} at fixed pressure and temperature can be expressed by the Nernst equation:

$$E_{rev} = E_{rev}^{\theta} - \frac{RT}{nF} \ln \frac{p_{\text{CO}_2}}{(p_{\text{CO}} \cdot p_{\text{O}_2}^{1/2})}$$

In our experiments, the SOECs were running at 800 and 850 °C, the anode was exposed to ambient air while the cathode side was fed with pure CO_2 . Although the E_{rev} in pure CO_2 environment cannot be obtained by the above Equation, E_{rev} value shows a decreasing trend with decreasing CO concentration. To verify the rationality of the OCVs obtained in our experiments, we tested the changes in OCV of SFNM+0.0Fe-red-GDC at 800 °C and atmosphere pressure when the cathodic atmosphere was switched from 70% CO_2 /30% CO to pure CO_2 . The measured OCV in the 70% CO_2 /30% CO atmosphere is 0.88 V (Fig. R12), very close to the $E_{rev} = 0.895$ V calculated from the Nernst Equation ($E_{rev}^{\theta} = 0.97$ V at atmosphere pressure and 800 °C⁴⁸). The E_{rev} remarkably decreases to 0.08 V after switching the atmosphere from 70% CO_2 /30% CO to pure CO_2 . Based on above discussion, the measured OCVs for CO_2 electrolysis in our work are all within a reasonable range calculated by the Nernst equation and similar to the OCVs reported in the literatures measured under the same conditions^{49,50}.

Figure R12. Change in OCV of SFNM+0.0Fe-red-GDC at 800 °C when switching the atmosphere from 70% CO₂/30% CO to pure CO₂.

Comment #5. Are there any Nyquist plots for the EIS measurements in Supplementary figure 9 and 10? In general, the DRT plot (Supplementary figure 10) is originated from the Nyquist plot for the EIS measurement.

Authors' response to Comment #5:

The Nyquist plots are shown in Fig. 3b in the manuscript.

Comment #6. There are too many slope changes in the TGA plot in Supplementary Figure 12. What could be the possible reason for many slope changes? Also, the TGA measurement condition is not listed in the caption of Supplementary figure 12.

Authors' response to Comment #6:

We thank the reviewer for the valuable comment. It is common for TGA plot to have multiple slopes in this system, and each individual slope can be ascribed to different process, such as the loss of the lattice oxygen caused by the reduction of B-site cations and exsolution of Fe-Ni alloy, as shown in the following equations⁵¹:

Loss of lattice oxygen caused by reducing oxidation states of Fe and Ni:

Loss of lattice oxygen caused by exsolution of Fe and Ni:

To reveal the detailed processes responsible for the slope changes from 400 to 800 °C, the ex-situ XRD and SEM results of SFNM+0.0Fe-red and SFNM+1.2Fe-red treated at 400, 500, 600, 700, 800 °C in 5% H₂/95% N₂ atmosphere are provided in Figs. R13 and R14. For SFNM+0.0Fe-red, there is only a sharp weight loss between 400 and 500 °C. From the XRD results, we can see that the main peak at about 32° shifts to the right slightly. It may be ascribed to the reduction of reducible Fe and Ni cations, which leads to the lattice expansion. The SEM results of SFNM+0.0Fe-red-500 °C-0h show that trace amount of exsolved nanoparticles has formed on the surface. It indicates that the onset temperature of nanoparticle exsolution on SFNM is below 500 °C. For SFNM+xFe ($x=0.5, 0.8, 1.2$), the ramps of weight loss curves appear at slightly lower temperatures (below 400 °C), indicating that the reduction and exsolution may happen at the lower temperatures, which is consistent with the promoting effect of ion exchange on exsolution. It is also verified by the XRD results that the main peak of SFNM+1.2Fe-red shifts slightly to lower angle from 400 to 500 °C. As temperature ramping, there is a sharp weight loss between 600 and 700 °C for SFNM+xFe ($x=0.5, 0.8, 1.2$). It may be ascribed to the promotion effect of TIE on nanoparticle exsolution, which results in the formation of Fe-Ni nanoparticles and loss of oxygen vacancies. The exsolved nanoparticles can be easily observed on the surfaces of SFNM+0.0Fe-red and SFNM+1.2Fe-red.

Figure R13. Ex-situ XRD results of (a₁-a₂) SFNM+0.0Fe-red and (b₁-b₂) SFNM+1.2Fe-red treated at 400, 500, 600, 700, 800 °C in 5% H₂/95% N₂ atmosphere.

Figure R14. Ex-situ SEM results of (a) SFNM+0.0Fe-red and (b) SFNM+1.2Fe-red treated at 400, 500, 600, 700, 800 °C in 5% H₂/95% N₂ atmosphere.

We have revised the Supplementary information (SI) (L12-14, P18 and L1-7, P19) accordingly, as shown below:

Revised SI (L12-14, P18 and L1-7, P19): “The gradual weight loss below 400 °C can be ascribed to the detachment of surface-absorbed H₂O molecule and decomposition of nitrate¹⁴. Further raising the temperature induces a continuous decline in sample weight for all samples, which was attributed to the formation of oxygen vacancy caused by the reduction of B-site cations and exsolution of Fe-Ni alloy, as shown in the following equations⁵¹:

Loss of lattice oxygen caused by reduced oxidation states of Fe and Ni:

Loss of lattice oxygen caused by exsolution of Fe and Ni:

Additionally, as suggested by the reviewer, the TGA measurement condition has been added in the caption of Supplementary Fig. 14 in the revised SI.

Comment #7. Are there any additional supporting data (e.g., X-ray absorption fine structure (XAFS) measurement) to further support the XPS data in Supplementary figure 13?

Authors' response to Comment #7:

Thanks to the reviewer for the suggestion. As suggested by the reviewer, the Fe and Ni K-edge X-ray absorption near edge structure (XANES) spectra among air-sintered SFNM-oxi, SFNM+0.0Fe-red and SFNM+1.2Fe-red have been provided in Fig. R15 (also Supplementary Fig. 17)^{52,53}. The shifts of pre-edge of Fe and Ni K-edge for SFNM+0.0Fe-red and SFNM+1.2Fe-red to lower energy with respect to that for SFNM-oxi are associated with the decrease of valence states of Fe and Ni after reduction^{54,55}. It is clear that the valence states of Ni are almost identical in SFNM+0.0Fe-red and SFNM+1.2Fe-red, while the Fe valence state for SFNM+1.2Fe-red is apparently lower, which is consistent with the Fe and Ni 2p XPS spectra. It suggests that the amount of the Ni exsolution has almost reached the peak value via conventional exsolution due to its lower co-segregation energy.

Figure R15 (also Supplementary Figure 17). Fe and Ni K-edge XANES spectra of sintered SFNM, SFNM+0.0Fe-red and SFNM+1.2Fe-red.

We have revised the SI (L5-14, P23) accordingly, as shown below:

Revised SI (L5-14, P23): “The Fe and Ni K-edge X-ray absorption near edge structure (XANES) spectra among air-sintered SFNM-oxi, SFNM+0.0Fe-red and SFNM+1.2Fe-red are provided in Supplementary Fig. 17^{52,53}. The shifts of pre-edge of Fe and Ni K-edge for SFNM+0.0Fe-red and SFNM+1.2Fe-red to lower energy with respect to that for SFNM-oxi are associated with the decrease of valence states of Fe and Ni after the reduction^{54,55}. It is clear that the valence states of Ni are almost identical in SFNM+0.0Fe-red and SFNM+1.2Fe-red, while the Fe valence state for SFNM+1.2Fe-red is apparently lower, which is consistent with the Fe and Ni 2p XPS spectra. It suggests that the amount of the Ni exsolution has almost reached the peak value via conventional exsolution due to its lower co-segregation energy.”

Comment #8. In my opinion, this work did not follow the Nature Communications format. Is the font for this work “times” or “times new roman”?

Authors' response to Comment #8:

The format of the revised manuscript follows the requirement of Nature Communications and the font is “Calibri”.

Comment #9. Why did the authors use electrode-GDC composite for SOEC measurements? Since the GDC could significantly affect the exsolution and/or TIE properties, the authors should compare the electrochemical performance measurements with only electrode material (SOEC cathode).

Authors' response to Comment #9:

We thank the reviewer for pointing this out. For electrodes in SOEC, the perovskite-based materials are usually mixed with a pure oxygen ion conductor to prolong the triple-phase boundaries and reaction sites, thereby resulting in a higher electrocatalytic performance than their pure perovskite cathode counterpart^{56,57}. The common pure ion conductors include gadolinium-doped ceria (GDC) and samarium-doped ceria (SDC). The CO₂ electrocatalysis process can be described using Kröger-Vink notation:

The reduction of GDC ($Ce^{4+} \rightarrow Ce^{3+}$) in the composite electrodes under cathodic polarization would provide additional electrical conduction paths^{58,59}.

Most importantly, since there is a GDC barrier layer between the electrode material and YSZ electrolyte in the cell assembly (LSCF-GDC|GDC|YSZ|GDC|SFNM+xFe-red-GDC), the composite cathodes are more chemically compatible with the GDC buffer layer. It would reduce the difference of thermal expansion coefficient and interfacial resistance between cathode and GDC buffer layer⁵⁷. For the same reason, LSCF-GDC composite was selected as anode electrode. Furthermore, for the material preparation and cell assembly, the SFNM+xFe-red ($x = 0.0, 0.5, 0.8, 1.2$) have been prepared by pre-reduction in 5% H₂/95% N₂ before mixing with GDC, so it would not affect the conventional and TIE-assisted exsolution processes.

Comment #10. In page 7 line 15, the authors stated as “X-ray diffraction (XRD) analysis confirms that secondary Fe-Ni alloy phases have emerged ...”. In general, secondary phase could also imply unnecessary or

impurity phase. However, the Fe-Ni alloy exsolution would positively affect the electro-catalytic properties, as also listed in this work.

Authors' response to Comment #10:

Thanks to the reviewer for this valuable suggestion, the “secondary Fe-Ni alloy phases” has been corrected to “Fe-Ni alloy phases” on L15, P8 of the revised manuscript.

Comment #11. What is the possible reason for ultrasound (ultrasonication) and freeze-drying Fe nitrate nonahydrate in the host material (Supplementary figure 1)? In addition, how much weight percent of Fe guest did the authors immerse in the SFNM material to promote TIE phenomenon?

Authors' response to Comment #11:

The water solubility of ferric nitrate and the high temperature decomposition of nitrate after heat treatment to 800 °C are the main reasons why we choose it as the Fe source to be loaded on the surface of perovskite scaffold. The ultrasound treatment followed by freeze-drying treatment allows the external Fe source to be uniformly deposited on the perovskite substrates. The principle of freeze-drying is to utilize the sublimation of objects. In a vacuum environment, the frozen water can directly become gas without going through the intermediate liquid phase, leaving the Fe source on the perovskite substrates.

The Fe loading amount on the surface of perovskite substrates has been defined on L14, P8 in the manuscript “a series of SFNM+xFe-red samples ($x=0.0, 0.5, 0.8, 1.2$, which refer to the molar ratios of guest Fe to host Ni)”. After determining the weight of SFNM powder, the weight ratio of ferric nitrate for preparing SFNM+xFe-red samples can be calculated by simple unit conversion (mole to gram). For instance, if using 0.3 g SFNM powder to prepare SFNM+xFe-red ($x=0.0, 0.5, 0.8, 1.2$), the weight of ferric nitrate required is shown in Table R5:

Table R5. Amount of deposited Fe nitrate in weight.

Samples (weighting 0.3 g SFNM powders)	Mole ratio of the guest Fe and host Ni	Mol. of guest Fe for deposition (mmol)	Weight of Fe(NO ₃) ₃ ·9H ₂ O (g)
SFNMo+0.5 Fe	0.5	0.0743	0.0300
SFNMo+0.8 Fe	0.8	0.1191	0.0481
SFNMo+1.2 Fe	1.2	0.1785	0.0721
SFNMo+1.5 Fe	1.5	0.2230	0.0901

Comment #12. In Figure 2, there seems to be secondary phases (NiO, NiFe₂O₄, and SrMoO₄) after re-oxidation in air. The formation of secondary phases could negatively affect the electrochemical properties. Moreover, the reason for conducting re-oxidation in air is not clearly explained in this work.

Authors' response to Comment #12:

Thanks to the reviewer for raising the question. The reoxidation experiments is to further verify the reoccupation of B-site by the guest Fe for SFNM+*x*Fe-red samples (*x*=0.5, 0.8, 1.2).

It has been reported that the exsolution of nanoparticles from perovskites is reversible/partially reversible^{24,60}, i.e., the exsolved nanoparticles could completely/partially dissolve into the lattice of parent perovskite by the re-oxidation treatment. To verify this, the XRD analyses were conducted. The results show that the additional NiO peaks have clearly emerged in re-oxidized SFNM+0.0Fe-red, indicating that the redissolution of the exsolved Fe-Ni nanoparticles on the SFNM is partially reversible. Moreover, Fe atoms in Fe-Ni nanoparticles preferentially dissolve into perovskite compared to Ni atoms, which is in accord with the fact that Ni exsolves more favourably than Fe. However, NiFe₂O₄ or SrMoO₄ can be detected on re-oxidized SFNM+ *x*Fe-red (*x*=0.5, 0.8, 1.2). The appearance of SrMoO₄ indicates that the B-sites have been almost occupied by the high-valence Fe and Mo³⁸, the exsolved Fe and Ni on surface are prone to self-assembly into binary oxide NiFe₂O₄ in air.

Comment #13. Why did the authors conduct reduction at 5% H₂/N₂ balance instead of real experimental conditions for SOEC cathode (Figure 2)?

Authors' response to Comment #13:

The pre-reduction treatment of sintered perovskites aims to produce the highly active metal nanoparticles on the surface by exposing the perovskite to a reducing atmosphere at elevated temperatures, thus enabling it to have a higher initial reactivity for CO₂ electrocatalysis. Fig. R16 shows the surface morphologies of exsolved nanoparticles on SFNM+0.0Fe-red and SFNM+1.2Fe-red after heating at 800 °C for 2h in 70% CO₂/30% CO atmosphere. The populations of the exsolved nanoparticles on both samples treated in 70% CO₂/30% CO environment are far less than those treated in 5% H₂/95% N₂. It suggests that the 5% H₂/95% N₂ atmosphere is more potent for nanoparticle exsolution than 70% CO₂/30% CO environment³⁴, which delivers a higher catalytic activity for CO₂ reduction.

Figure R16. Surface morphologies of (a) SFNM+0.0Fe-red and (b) SFNM+1.2Fe-red after heating at 800 °C for 2h in 70% CO₂/30% CO atmospheres.

This stark difference in the surface features of reduced perovskites may be dominated by the different oxygen partial pressure and surface defect reactions in different reducing atmospheres^{17,34}. The exsolution process of Fe and Ni from SFNM perovskite in the two types of atmospheres (H₂/N₂ and CO/CO₂) can be expressed by the following equations:

Exsolution at H₂/N₂ atmosphere:

Exsolution at CO/CO₂ atmosphere:

The exsolved Fe-Ni alloy nanoparticles have been regarded as the efficient CO₂ adsorption sites on the surface of perovskites. Consequently, it would block the binding of CO molecules with the surface of perovskites, which may inhibit the continuous exsolution³⁴, as shown from the above reaction process. Therefore, the H₂/N₂ reducing atmospheres have been widely applied to produce the P-eNs in the literatures and our work.

Comment #14. There seems to be SrO segregation in Figure 3d. In general, the A-site segregation hampers the exsolution capability and could also negatively affect the electro-catalytic properties.

Authors' response to Comment #14:

We agree that the A-site segregation would hamper the exsolution capability and could also negatively affect the electro-catalytic properties. As seen from Figure 3d, SrO segregation occurs at around the nanoparticles of the two samples, and the undesirable Sr-segregation of SFNM+1.2Fe-red is less severe than that of SFNM+0.0Fe-red. It is mainly due to the B-site supplement, as explained by Eqs. 6 and 7 on P14-15 in the revised manuscript.

Comment #15. The degradation is observable in Figure 4c. In the case of SFNM+1.2Fe-red-GDC (Figure 4c), there seems to be about 30% degradation after 100-hour stability test.

Authors' response to Comment #15:

We agree with the reviewer. The observable degradation is mainly ascribed to the harsh testing conditions (1.6 V and 850 °C), which cause the degradation of not only cathode, but also the anode and electrolyte. As mentioned on L13-18, P19 in the revised manuscript, the high cathode potential and temperature are applied to intentionally accelerate the degradation of SOECs to help us understand the cathode evolution and further shed lights on the degradation mechanisms. The SEM results of cathode surface morphologies on the SFNM+0.0Fe-red-GDC and SFNM+1.2Fe-red-GDC as well as the cross-section of electrolysis cells after the long-term stability are characterized, as shown in Figs. R17 and R18 (also Figs. 5d-e and Supplementary Fig. 24). The obvious morphology changes and coarsening of exsolved particles can be observed on both cathodes, and the coarsening of surface nanoparticles on SFNM+1.2Fe-red-GDC is less severe than that on SFNM+0.0Fe-red-GDC (Fig. R17). Furthermore, the significant grain coarsening of YSZ electrolyte and the delamination of anode from the electrolyte can be observed on both cells after the long-term stability tests and both factors also contributed to the degradation (Fig. R18).

Figure R17 (also Figures 5d-e). SEM images of cathode surface microstructures of (a) SFNM+0.0Fe-red-GDC and (b) SFNM+1.2Fe-red-GDC after long-term stability at 1.6 V and 850 °C.

Figure R18 (also Supplementary Figure 24). SEM images of cross section of (a) SFNM+0.0Fe-red-GDC and (b) SFNM+1.2Fe-red-GDC after long-term stability at 1.6 V and 850 °C.

We have revised the manuscript (L1-5, P21, L14-18, P25) and SI (L5-14, P31) accordingly, as shown below:

Revised manuscript (L1-5, P21): “After the long-term stability tests, the significant morphology changes and coarsening of exsolved particles can be observed on the both cathode surfaces, and the coarsening of surface nanoparticles on SFNM+1.2Fe-red-GDC is less severe than that on SFNM+0.0Fe-red-GDC (Figs. 5d-e).”

(L14-18, P25): “In addition, the deteriorations of the electrolyte and the anode under the harsh conditions seem to be inevitable (Supplementary Fig. 24), which are also responsible for the significant attenuation of the current density during CO₂ electrocatalysis at 1.6 V and 850 °C.”

Revised SI (L5-14, P31): “The degradation of the electrolyte and the anode under the harsh conditions are analyzed⁶¹. Supplementary Fig. 24 shows the SEM images of the cross section of SFNM+0.0Fe-red-GDC and SFNM+1.2Fe-red-GDC after the long-term stability tests at 1.6 V and 850 °C. The significant grain coarsening of YSZ electrolyte can be observed near the anode|GDC|YSZ interfaces of both cells, and this grain growth propagates along the electrolyte and progresses towards the cathode side. Furthermore, the pore formation near the anode|GDC|YSZ interfaces and even the delamination of anode from the electrolyte can be observed, which is supposed to originate from the high oxygen partial pressures formed at the interfaces during the long-term CO₂ electrolysis at high voltage.”

Comment #17. In the abstract part, the authors stated as “In this study, we reveal that the formation of B-site vacancies in perovskite scaffold is the major contributor to the degradation of P-eNs; ...”. Instead of B-site vacancies in perovskite scaffold, A-site segregation and/or the formation of impurity phases (SrCO₃ and SrMoO₄) in Figure 4e appears to be much significant factor for the degradation properties. This part should be clearly elucidated.

Authors' response to Comment #17:

We thank the reviewer for pointing this out. From our work, we conclude that the structural decomposition of perovskite scaffold is caused by the accumulation of B-site vacancies during the CO₂ electrolysis at high voltages. During exsolution, the reducible B-site ions migrate to the surface to form nanoparticles, accompanied by the formation of B-site vacancies. When a small number of B-site reducible cations exsolve out of the perovskite lattice, the concentration of B-site vacancies is not enough to reach the threshold of destructive structural decomposition since perovskite has a certain structural tolerance. When the created B-site vacancies exceeds the threshold, the perovskite partially decomposes into Sr-based derivatives on the surface. Therefore, the A-site segregation and the formation of impurity phases (SrCO₃ and SrMoO₄) during exsolution are the significant factor for the degradation properties. As discussed in the revised manuscript (L3-4, P15): “The Sr-rich surface would reorganize into SrCO₃ at a high CO₂ concentration, which has a detrimental effect on the surface catalytic activity for CO₂ reduction⁶².” and the revised manuscript (L16-18, P21): “The emergence of Sr-containing impurities on both spectra confirms the structure decomposition and surface reconstruction on SFNM+0.0Fe-red and SFNM+1.2Fe-red.” and the revised manuscript (L5-8, P22): “As shown in Fig. 6b, only the Raman feature peaks of SrCO₃ and SrMoO₄ were detected but no peaks of carbon, which confirms that the perovskite scaffold of SFNM+0.0Fe-red primarily underwent the bulk structural decomposition before the exposure of newly grown nanoparticles and carbon deposition.”

Comment #18. The current density for SFNM+0.0Fe-red-GDC at 1.8 V much decreases just after 15 minutes. Hence, the production rate of CO and the Faradaic efficiency of CO should be much different between 60 min and 75 min.

Authors' response to Comment #18:

Thanks for this question. The Faradaic efficiency of CO is calculated by the following equation^{63,32}:

$$FE_{CO} = \frac{0.1315 \times V\left(\frac{mL}{min}\right) \times v(vol\%)}{I_{total}(A)} \times 100\%$$

where $V\left(\frac{mL}{min}\right)$ = Gas flow rate measured by a flow meter at the exit of the cell at room temperature and under ambient pressure.

$v(vol\%)$ = Volume concentration of CO in the exhaust gas from the cell (obtained by gas chromatography).

$I_{total}(A)$ = cell current during short-term stability experiments.

As shown in the equation, FE_{CO} depends on the ratio of $v(vol\%)$ to $I_{total}(A)$. The current density for SFNM+0.0Fe-red-GDC at 1.8 V drops remarkably after 15 minutes, but in the meantime, the $v(vol\%)$ decreases at the same degree accordingly. Therefore, the value of FE_{CO} would not change significantly.

The following equation is used to calculate the production rate of CO⁶⁴:

$$Production\ rate\ of\ CO = \frac{q_{total} \times FE_{CO} \times 22.4(L/mol) \times 1000 \times 60(\frac{h}{min})}{FntS \times 100}$$

where q_{total} = total charge passed.

FE_{CO} = Faradaic efficiency of CO.

F = Faraday constant.

n = the number of exchanged electrons to produce CO from CO₂.

t = electrolysis time (h).

S = the geometric area of the electrode (cm²).

The production rate of CO is proportional to the q_{total} which is the integral of cell current versus time and is an average value over 15 min. Fig. 5b in the revised manuscript shows the comparison of the average CO productivity of short-term stability tests at different potentials. It can be seen that the current density for SFNM+0.0Fe-red-GDC at 1.8 V drops remarkably during the stability test, the total charge passing through SFNM+0.0Fe-red-GDC at 1.8 V is much smaller than that at 1.6 V during the 15 min of test. Therefore, the CO productivity at 1.8 V is smaller than that at 1.6 V during the 15 min of test, as shown in Fig. 5b. In addition, the instantaneous production rate of CO should decrease with decreasing the current density. Hence, the CO productivity for SFNM+0.0Fe-red-GDC at 1.8 V decreases significantly from 60 min to 75 min.

Comment #19. How did the authors calculate the Faradaic efficiency of CO and the production rate of CO via gas chromatography measurements? I couldn't find clear explanations and equations on how the authors calculated the Faradaic efficiency of CO and the production rate of CO.

Authors' response to Comment #19:

We thank the reviewer for this comment. We have added the equations of the Faradaic efficiency of CO and the production rate of CO to the "Method section" in the revised manuscript (P33-34).

Comment #20. Did the authors calculate the Gibbs free energy for Fe-Ni alloy formation? Furthermore, is there any Gibbs free energy calculations on whether the Fe-Ni alloy formation is favorable at the surface or the bulk? This part should be also explained in this paper since the title in this work is “B-site supplement mechanism: A new approach to boosting stability of perovskites with exsolved nanoparticles”.

Authors' response to Comment #20:

Thanks for the suggestion. The Gibbs free energy for Fe-Ni alloy formation is not calculated in the manuscript because we can know that the Fe-Ni alloy can be formed from the XRD and TEM-EDS results in this work. The formation location of alloy in perovskites is a very interesting topic. It is a challenging work to unveil the formation location of alloy nanoparticles by DFT calculation since it is not easy to define the accurate configuration of alloy formed in the perovskite bulk from the perspective of energy. Therefore, it may not be possible to correctly judge the preferred sites of alloy formation by DFT method. To the best of our knowledge, only the work of Kwon et al. was to explore the bulk Co-Ni alloy formation using DFT calculation⁶⁵. They defined the configuration in which Co and Ni share the same oxygen vacancy in bulk, but this definition does not really represent the actual situation.

Alternatively, it has been recently reported that the surface exsolved nanoparticles from $(\text{La}_{0.3}\text{Ca}_{0.7})_{0.95}\text{Fe}_{0.7}\text{Cr}_{0.25}\text{Ni}_{0.05}\text{O}_{3-\delta}$ grown in a $\text{H}_2\text{-N}_2$ atmosphere exhibit a compositional progression from an initially Ni-rich phase to an Fe-rich phase during the heat treatment³⁴. It indicates that the formation of surface Fe-Ni alloys have undergone the process of diffusion of Fe and Ni from the bulk to the surface followed by alloying at surface. In addition, two other typical studies on the metallic nanoparticle nucleation mechanism demonstrated that the exsolved metallic nanoparticles would nucleate at surface or subsurface by in-situ TEM, atomic force microscopy (AFM) and strain field modeling^{25,31}. These two metallic nanoparticle formation scenarios are more acceptable compared to the bulk nanoparticle formation due to the high strain from the surrounding lattice and high diffusion energy barrier of the nanoparticles formed in the perovskite bulk. Besides, Kousi et al. reported the endogenous nanoparticles nucleation in perovskite bulk by selecting a titanate perovskite with relatively low cation transport²⁸. They proposed that the design principle of bulk nucleation is to make the diffusivity of B-site reducible cations in the bulk as low as possible, so that the B-site reducible cations favour local particle nucleation rather than transport to the surface followed nucleation. All these evidences demonstrate that the favourable location for the formation of exsolved metallic nanoparticle is at surface rather than in bulk. Furthermore, the main focus of our work is to enhance the stability of P-eNs rather than to investigate the formation location of alloy exsolution Thus, our goal of this work is only focused on calculating the co-segregation energy of different B-site cations of SFNM to judge the external B-site supplemental element.

Comment #21. The overall thesis design and/or English proficiency should be much complemented in this work.

Authors' response to Comment #21:

We appreciate the reviewer's comment and have carefully revised the manuscript to improve its readability.

Reviewer #3 (Remarks to the Author):

This work proposed B site supplement strategy to compensate the occurrence of B site vacancies caused by exsolution process before and during the operation, especially at high reduction potential. Mostly, nanoparticle exsolution of B site cations in perovskite lattice leaves many B site vacancies and causes detrimental A site segregation, leading to the losses of $B^{(n+1)}-O-B^{n+}$ pathway. In this work, authors have insisted that the guest Fe metal coated on $Sr_2Fe_{1.3}Ni_{0.2}Mo_{0.5}O_{6-\delta}$ (SFNM) would fill the B site vacancies by topotactic ion exchange (TIE) mechanism and mitigate the Sr segregation on the surface, thus enhancing structure stability and facilitating stable CO_2 electrolysis performance at high voltage. Several characterizations such as XRD, SEM, TEM, and elemental mappings were performed to verify the role of guest Fe, filling the B site vacancies during exsolution process and suppressing Sr segregation on perovskite surface. In potential step chronoamperometry, SFNM without guest Fe (SFNM+0.0Fe-red) showed significant decay of current density (j), whereas SFNM with guest Fe metals (SFNM+1.2Fe-red) showed stable j profiles, especially under high operating voltage (1.8V). In this work, the results are noteworthy and would be useful to develop a highly active CO_2 electrolysis cathode catalyst with superior stability in SOEC system. Authors have provided detailed mechanisms of perovskite structure evolution during the exsolution and TIE-assisted exsolution, and most of conclusions are supported by the results. It might be suitable to be published in this journal after addressing below issues and comments:

Authors' response:

We thank Reviewer #3 for the highly positive comments and recommendation for publication. Our detailed responses to each comment are as follows.

Comment #1. In page 3~4, authors have mentioned that the exsolution process on the perovskite leaves many B site vacancies within perovskite bulk and thus A site segregation on the surface as below equation:

However, it is well known that these defects (i.e., $B_{(1-\alpha)}$ and AO) can be relieved by controlling A site non-stoichiometry on perovskite lattice as followed:

A-site deficient design has been adopted in numerous other researches of SOEC CO_2 cathodes, especially in [D. Neagu et al., Nat. Chem., 2013, 5, 11, 916-923]. It might be helpful to mention on your manuscript that what is the strong point of your strategy to coat additional guest Fe metals on perovskite surface to induce TIE-assisted exsolution compared to the one, which controls the A site non-stoichiometry of perovskite oxide.

Authors' response to Comment #1:

Thanks to the reviewer for this valuable suggestion. Indeed, as mentioned by the reviewer, A-site deficiency/B-site excess is an alternative way to reduce the concentration of the B-site vacancies of perovskite scaffold in addition to the ion exchange. These two schemes had been considered when we proposed the B-site supplement mechanism. However, the adjustable range of A-site deficiency in $\text{Sr}_2\text{Fe}_{1.3}\text{Ni}_{0.2}\text{Mo}_{0.5}\text{O}_{6-\delta}$ (SFNM) is relatively limited. It has been reported that pure $\text{Sr}_{1.9}\text{Fe}_{1.3}\text{Ni}_{0.2}\text{Mo}_{0.5}\text{O}_{6-\delta}$ cannot be obtained with 5% mol Sr-site deficiency³⁵. Moreover, almost all Ni elements can be exsolved from SFNM after reduction treatment at 800 °C and a 5% H_2/N_2 atmosphere for 2 h. In this case, less than 5% mol Sr-site deficiency cannot fully retain the occupation of the B-site vacancies after exsolution. Therefore, we choose the ion exchange assisted exsolution method to achieve B-site supplement of perovskite scaffold of reduced SFNM. As suggested by the reviewer, we have added the following paragraph on L10-19, P4 in the revised manuscript to explain the strong point of the strategy to coat additional guest Fe sources on the perovskite surface to induce TIE-assisted exsolution compared to the one controlling the A site non-stoichiometry of perovskite oxide.

Revised manuscript (L10-19, P4) : "In this study, the promising double perovskite $\text{Sr}_2\text{Fe}_{1.3}\text{Ni}_{0.2}\text{Mo}_{0.5}\text{O}_{6-\delta}$ (SFNM) was selected as a prototype example to elaborate the effects of the structure evolution of the perovskite scaffold on the stability of P-eNs³. Either controlling the A-site deficiency or implementing the topotactic ion exchange (TIE) is expected to be a pathway to regulate the concentration of the B-site vacancies in the reduced SFNM (Eqs. 2 and 3)^{22,26,36,37}. However, the limited Sr-site deficiency (less than 5% mol) in the SFNM makes it fail to refill the B-site vacancies after the exsolution by controlling the A-site deficiency³⁵. Therefore, the TIE-assisted exsolution was employed to fine-tune the B-site occupation of perovskite scaffold while promoting the formation of nanoparticles."

Comment #2. In page 6, co-segregation energies of Ni, Fe and Mo at B site of SFNM were calculated by DFT method. From the calculation results, it seems to be reasonable to choose Ni as a guest ions to induce vigorous TIE-assisted exsolution because of its relatively low segregation energies (Ni: -1.46 eV, Fe: -1.06 eV, Mo: 2.26 eV). However, authors have chosen Fe as guest metals due to its relatively high redox stability (in page 5). Is there any additional calculation results that can support the choice of Fe as a guest ions?

Authors' response to Comment #2:

The following factors are mainly considered when choosing the B-site supplement agent: 1) the doping availability of deposited cations into SFNM perovskite lattice; 2) the exchangeability of deposited ions with the easily reducible Fe and Ni in the parent SFNM lattice.

Firstly, the $\text{Sr}_2\text{Fe}_{1.3}\text{Ni}_{0.2}\text{Mo}_{0.5}\text{O}_{6-\delta}$ (SFNM) can be seen as the $\text{Sr}_2\text{Fe}_{1.5}\text{Mo}_{0.5}\text{O}_{6-\delta}$ (SFM) with the partial Ni substitution at Fe-site, which causes the structural instability of perovskite scaffold and yields the highly active

Fe-Ni alloy after the reduction. Therefore, the redox-stable Fe, instead of the highly reducing Ni, is expected to supplement into B-site vacancies, thus leading to a relatively robust perovskite scaffold.

We clearly know that Fe and Ni can be exsolved from SFNM, leaving the Fe- and Ni-site vacancies, so the external Fe source must be able to exchange with the host Fe and Ni to achieve B-site full occupation. The exchangeability is determined by comparing the co-segregation energies of B-site elements^{26,37}. The elements with lower co-segregation energy tend to exsolve on the surface, in turn, the elements with higher co-segregation energy tend to dissolve into the perovskite lattice. The calculated co-segregation energy trend of B-site elements is Ni>Fe>Mo. Thus, the deposited Fe can exchange with host Ni and Fe but not the host Mo, resulting in the supplement of Fe and Ni-site vacancies of reduced SFNM with foreign Fe. However, the deposited Ni can only exchange with the host Ni, leaving a large number of Fe-site vacancies in SFNM. Therefore, the relatively redox-stable Fe is selected to supplement into the B-site vacancies during exsolution.

Comment #3. Authors have insisted that TIE-assisted exsolution would decrease B site vacancies that could be produced during exsolution, thus help to maintain the electrical conductivity of bulk perovskite. XPS analysis, which showed more Fe²⁺-Fe³⁺ and Mo⁵⁺-Mo⁶⁺ charge pairs in SFNM+1.2Fe-red, was conducted to support the opinions. Is it possible to measure electrical conductivities of SFNM+0.0Fe-red and SFNM+1.2Fe-red under reducing and operating atmosphere?

Authors' response to Comment #3:

We thank the reviewer for this valuable suggestion. As suggested by the reviewer, the temperature-dependent electrical conductivities of SFNM+0.0Fe-red and SFNM+1.2Fe-red in the 50% CO₂/50% CO and 5% H₂/95% Ar atmospheres were measured and the results are shown in Fig. R19 (also Fig. 4d and Supplementary Fig. 18). In the 50% CO₂/50% CO atmosphere, the electrical conductivities of both SFNM+0.0Fe-red and SFNM+1.2Fe-red increase as the temperature increases, reaching 16.5 and 24.6 S cm⁻¹ at 850 °C, respectively (Fig. 4d). The results clearly show that SFNM+1.2Fe-red has the higher conductivity in the prospective operation atmosphere of CO₂ electrolysis. The electrical conductivities of SFNM+0.0Fe-red and SFNM+1.2Fe-red in the 5% H₂/95% Ar atmosphere at 850 °C reach 26.0 and 48.2 S cm⁻¹, respectively, which are higher than those in the 50% CO₂/50% CO atmosphere. It can be ascribed to the contribution of the more exsolved Fe-Ni alloy nanoparticles formed under the more reducing condition in addition to the conduction pathways in perovskite scaffold⁶⁶. A large number of metallic nanoparticles on SFNM+1.2Fe-red leads to the slightly reduced conductivity as the temperature increases in the 5% H₂/95% Ar atmosphere.

Figure R19 (also Figure 4d and Supplementary Figure 18). Temperature-dependent electrical conductivities of SFNM+0.0Fe-red and SFNM+1.2Fe-red under 50% CO_2 /50% CO and 5% H_2 /95% Ar atmospheres.

We have revised the manuscript (L16-19, P17 and L1-4, P18) and SI (L5-12, P24) accordingly, as shown below:

Revised manuscript (L16-19, P17 and L1-4, P18): “The temperature-dependent electrical conductivity results show that SFNM+1.2Fe-red performs better than SFNM+0.0Fe-red in the 50% CO_2 /50% CO and 5% H_2 /95% Ar atmospheres (Fig. 4d and Supplementary Fig. 18). In the 50% CO_2 /50% CO atmosphere, the electrical conductivities of SFNM+0.0Fe-red and SFNM+1.2Fe-red increase as the temperature increases, reaching 16.5 and 24.6 S cm^{-1} at 850 $^{\circ}\text{C}$, respectively (Fig. 4d). SFNM+1.2Fe-red shows the higher conductivity in the prospective operation atmosphere of CO_2 electrolysis.”

Revised SI (L5-12, P24): “As depicted in Supplementary Fig. 18, the electrical conductivities of SFNM+0.0Fe-red and SFNM+1.2Fe-red in the 5% H_2 /95% Ar atmosphere at 850 $^{\circ}\text{C}$ reach 26.0 and 48.2 S cm^{-1} , respectively, which are higher than those in the 50% CO_2 /50% CO atmosphere. It can be ascribed to the contribution of the more exsolved Fe-Ni alloy nanoparticles formed under the more reducing condition in addition to the conduction pathways in perovskite scaffold⁶⁶. A large number of metallic nanoparticles on SFNM+1.2Fe-red leads to the slightly reduced conductivity as the temperature increases in the 5% H_2 /95% Ar atmosphere.”

Comment #4. In Figure 3, two electrodes, SFNM+0.0Fe-red and SFNM+1.2Fe-red, have shown different behaviors in ohmic resistance when the external voltage is applied. It would be better to explain the change of ohmic resistance under applied voltage conditions for each electrodes.

Authors' response to Comment #4:

We thank the reviewer for this suggestion. As suggested, the changes in the ohmic resistance (R_s) with the applied voltages have been analyzed in the revised SI (P12-13), as shown below:

Revised SI (P12-13): "From 1.0 to 1.2 V, the cathode reaction rate accelerates as the voltage increases, leading to an increase of the conductivity, i.e., the oxygen ion transfer resistance is reduced at cathode side and so is R_s . However, the R_s values of both cells increase as the applied voltage increases to 1.4 V, which may be ascribed to the increased resistance at the anode/electrolyte interface. Because the oxygen ions produced at the anode/electrolyte interface gradually accumulate when the cathode reaction kinetics speeds up, this accumulation would hinder the transfer of oxygen ions. This explains why the R_s of the SFNM+1.2Fe-red-GDC with superior oxygen ion conduction at cathode side increases by a larger magnitude than that of SFNM+0.0Fe-red-GDC. As the applied voltage further increases to 1.6-1.8 V (above the thermally neutral voltage $E_H=1.46$ V at 850 °C^{67,68}), the net heat is produced because the entropic heat consumption rate becomes slower than the production rate of the irreversible heat (due to activation, ohmic, and mass transport losses in the electrolyzer⁶⁷). This net heat would cause the promoted transport of oxygen ions, consequently leading to a greatly reduced R_s ."

Figure R20 (also Supplementary Figure 10). Change in R_s of SFNM+0.0Fe-red-GDC and SFNM+1.2Fe-red-GDC with voltages.

Comment #5. Performance comparison tables of cathode catalysts with exsolved-metal nanoparticles from other researches under similar conditions would be proper to show novelty of your works. Below papers would be helpful to use.

S. Park et al., Energy Technol., 2021, 9, 2100116.

L. Houfu et al., J. Mater. Chem. A, 2019, 7, 11967

J. Choi et al., J. Mater. Chem. A, 2021, 9, 8740

Authors' response to Comment #5:

Thanks to the reviewer for this constructive suggestion and important information. Performance comparison tables are helpful for highlighting the novelty of our work. We have cited all the references suggested by the reviewer in our revised manuscript. As discussed in our response to Reviewer #1, Comment #1A, the Nyquist plots and the fitted R_s , R_p (equivalent circuit model $LR(Q_H R_H)(Q_L R_L)$) of SFNM+1.2Fe-red-GDC at 800 °C are provided. The current density, polarization resistance, CO productivity and short-term stability of SFNM+1.2Fe-red-GDC in our work are compared with other state-of-art P-eNs based SOECs. Please refer our response to Comment 1A, Reviewer #1 for more details.

Comment #6. What is pre-reduction conditions of SFNM+1.2-red electrode before CO₂ electrolysis test?

Authors' response to Comment #6:

Sorry for having missed this information. All the pre-reduction treatments for SFNM+ x Fe-red-GDC ($x=0.0, 0.5, 0.8$) cathode before CO₂ electrocatalysis are the same. We have revised the manuscript (L11-15, P32), as shown below:

Revised manuscript (L11-15, P32): "The 5% H₂/N₂ reducing gas flow is continuously pumped into the SFNM+ x Fe-red-GDC ($x=0.0, 0.5, 0.8, 1, 2$) cathode compartment during the temperature ramping of SOEC. The temperature is maintained at 800 °C for 2 h to complete the further reduction and exsolution of SFNM+ x Fe-red-GDC."

References

1. Kousi, K., Tang, C., Metcalfe, I. S. & Neagu, D. Emergence and future of exsolved materials. *Small* **17**, 2006479 (2021).
2. Sun, X. *et al.* Progress of exsolved metal nanoparticles on oxides as high performance (electro)catalysts for the conversion of small molecules. *Small* **17**, 2005383 (2021).
3. Lv, H. *et al.* In situ exsolved FeNi₃ nanoparticles on nickel doped Sr₂Fe_{1.5}Mo_{0.5}O_{6-δ} perovskite for efficient electrochemical CO₂ reduction reaction. *J. Mater. Chem. A* **7**, 11967-11975 (2019).
4. Kim, J. H. *et al.* Nanoparticle ex-solution for supported catalysts: materials design, mechanism and future perspectives. *ACS Nano* **15**, 81–110 (2020).
5. Neagu, D. *et al.* Nano-socketed nickel particles with enhanced coking resistance grown in situ by redox exsolution. *Nat. Commun.* **6**, 8120 (2015).
6. Li, Y., Zhan, Z. & Xia, C. Highly efficient electrolysis of pure CO₂ with symmetrical nanostructured perovskite electrodes. *Catal. Sci. Technol.* **8**, 980-984 (2018).
7. Chen, L., Xu, J., Wang, X. & Xie, K. Sr₂Fe_{1.5+x}Mo_{0.5}O_{6-δ} cathode with exsolved Fe nanoparticles for enhanced CO₂ electrolysis. *Int. J. Hydrog. Energy.* **45**, 11901-11907 (2020).
8. Bai, L. *et al.* New insight into the doped strontium titanate cathode with in situ exsolved nickel nanoparticles for electrolysis of carbon dioxide. *Adv. Mater. Interfaces* **8**, 2001598 (2020).
9. Lv, H. *et al.* In situ investigation of reversible exsolution/dissolution of CoFe alloy nanoparticles in a Co-doped Sr₂Fe_{1.5}Mo_{0.5}O_{6-δ} Cathode for CO₂ Electrolysis. *Adv. Mater.* **32**, e1906193 (2020).
10. Ye, L. *et al.* Enhancing CO₂ electrolysis through synergistic control of non-stoichiometry and doping to tune cathode surface structures. *Nat. Commun.* **8**, 14785 (2017).
11. Choi, J. *et al.* Highly efficient CO₂ electrolysis to CO on Ruddlesden–Popper perovskite oxide with in situ exsolved Fe nanoparticles. *J. Mater. Chem. A* **9**, 8740-8748 (2021).
12. Tian, Y. *et al.* Boosting CO₂ electrolysis performance by calcium oxide-looping combined with in situ exsolved Ni-Fe nanoparticles based on symmetrical solid oxide electrolysis cell. *J. Mater. Chem. A* **8**, 14895-14899 (2020).
13. Xi, X., Fan, Y., Zhang, J., Luo, J.-L. & Fu, X.-Z. In situ construction of hetero-structured perovskite composites with exsolved Fe and Cu metallic nanoparticles as efficient CO₂ reduction electrocatalysts for high performance solid oxide electrolysis cells. *J. Mater. Chem. A* **10**, 2509-2518 (2022).
14. Ding, S. *et al.* A-site deficient perovskite with nano-socketed Ni-Fe alloy particles as highly active and durable catalyst for high-temperature CO₂ electrolysis. *Electrochim. Acta* **335**, 135683 (2020).
15. Neagu, D. & Irvine, J. T. S. Enhancing electronic conductivity in strontium titanates through correlated A and B-site doping. *Chem. Mater.* **23**, 1607-1617 (2011).
16. Na, B. T. *et al.* Enhanced accuracy of electrochemical kinetic parameters determined by electrical conductivity relaxation. *Solid State Ion.* **361**, 115561 (2021).
17. Jiang, Y., Yang, Y., Xia, C. & Bouwmeester, H. J. M. Sr₂Fe_{1.4}Mn_{0.1}Mo_{0.5}O_{6-δ} perovskite cathode for highly efficient CO₂ electrolysis. *J. Mater. Chem. A* **7**, 22939-22949 (2019).
18. Li, Y., Chen, X., Yang, Y., Jiang, Y. & Xia, C. Mixed-Conductor Sr₂Fe_{1.5}Mo_{0.5}O_{6-δ} as Robust Fuel Electrode for Pure CO₂ Reduction in Solid Oxide Electrolysis Cell. *ACS Sustain. Chem. Eng.* **5**, 11403-11412 (2017).
19. Islam, M. S. & Davies, R. A. Atomistic study of dopant site-selectivity and defect association in the lanthanum gallate perovskite. *J. Mater. Chem.* **14**, 86-93 (2004).
20. Islam, A. J. M. S. Atomic-scale insight into LaFeO₃ perovskite defect nanoclusters and ion migration. *J Phys. Chem. C* **112**, 4455–4462 (2008).

21. Mogensen, M., Lybye, D., Bonanos, N., Hendriksen, P. & Poulsen, F. Factors controlling the oxide ion conductivity of fluorite and perovskite structured oxides. *Solid State Ion.* **174**, 279-286 (2004).
22. Neagu, D., Tsekouras, G., Miller, D. N., Menard, H. & Irvine, J. T. In situ growth of nanoparticles through control of non-stoichiometry. *Nat. Chem.* **5**, 916-923 (2013).
23. Myung, J. H., Neagu, D., Miller, D. N. & Irvine, J. T. Switching on electrocatalytic activity in solid oxide cells. *Nature* **537**, 528-531 (2016).
24. Neagu, D. *et al.* Demonstration of chemistry at a point through restructuring and catalytic activation at anchored nanoparticles. *Nat. Commun.* **8**, 1855 (2017).
25. Neagu, D. *et al.* In situ observation of nanoparticle exsolution from perovskite oxides: from atomic scale mechanistic insight to nanostructure tailoring. *ACS Nano* **13**, 12996-13005 (2019).
26. Joo, S. *et al.* Cation-swapped homogeneous nanoparticles in perovskite oxides for high power density. *Nat. Commun.* **10**, 697 (2019).
27. Han, H. *et al.* Lattice strain-enhanced exsolution of nanoparticles in thin films. *Nat. Commun.* **10**, 1471 (2019).
28. Kalliopi Kousi, D. N., Leonidas Bekris, Evangelos I. Papaioannou and Ian S. Metcalfe. Endogenous nanoparticles strain perovskite host lattice providing oxygen capacity and driving oxygen exchange and CH₄ conversion to syngas. *Angew. Chem. Int. Ed.* **132**, 2531-2540 (2019).
29. Chen, Z. *et al.* Organic photochemistry-assisted nanoparticle segregation on perovskites. *Cell Rep. Phys. Sci.* **1**, 100243 (2020).
30. Lv, H. *et al.* Promoting exsolution of RuFe alloy nanoparticles on Sr₂Fe_{1.4}Ru_{0.1}Mo_{0.5}O_{6-δ} via repeated redox manipulations for CO₂ electrolysis. *Nat. Commun.* **12**, 5665 (2021).
31. Oh, T. S. *et al.* Evidence and model for strain-driven release of metal nanocatalysts from perovskites during exsolution. *J. Phys. Chem. Lett.* **6**, 5106-5110 (2015).
32. Liu, S., Liu, Q. & Luo, J.-L. Highly stable and efficient catalyst with in situ exsolved Fe–Ni alloy nanospheres socketed on an oxygen deficient perovskite for direct CO₂ electrolysis. *ACS Catal.* **6**, 6219-6228 (2016).
33. Opitz, A. K. *et al.* Surface chemistry of perovskite-type electrodes during high temperature CO₂ electrolysis investigated by operando photoelectron spectroscopy. *ACS Appl. Mater. Interfaces* **9**, 35847-35860 (2017).
34. Ansari, H. M., Bass, A. S., Ahmad, N. & Birss, V. I. Unraveling the evolution of exsolved Fe–Ni alloy nanoparticles in Ni-doped La_{0.3}Ca_{0.7}Fe_{0.7}Cr_{0.3}O_{3-δ} and their role in enhancing CO₂–CO electrocatalysis. *J. Mater. Chem. A* (2021).
35. Yu, N. *et al.* Understanding the A-site non-stoichiometry in perovskites: promotion of exsolution of metallic nanoparticles and the hydrogen oxidation reaction in solid oxide fuel cells. *Sustain. Energy Fuels* **5**, 401-411 (2021).
36. Bak, J., Bin Bae, H. & Chung, S. Y. Atomic-scale perturbation of oxygen octahedra via surface ion exchange in perovskite nickelates boosts water oxidation. *Nat. Commun.* **10**, 2713 (2019).
37. Joo, S. *et al.* Highly active dry methane reforming catalysts with boosted in situ grown Ni-Fe nanoparticles on perovskite via atomic layer deposition. *Sci. Adv.* **6**, eabb1573 (2020).
38. Xi, X. *et al.* Reducing d-p band coupling to enhance CO₂ electrocatalytic activity by Mg-doping in Sr₂FeMoO_{6-δ} double perovskite. *Nano Energy* **82**, 105707 (2020).
39. Liu, Q., Dong, X., Xiao, G., Zhao, F. & Chen, F. A novel electrode material for symmetrical SOFCs. *Adv. Mater.* **22**, 5478-5482 (2010).
40. Liu, Q., Yang, C., Dong, X. & Chen, F. Perovskite Sr₂Fe_{1.5}Mo_{0.5}O_{6-δ} as electrode materials for symmetrical solid oxide electrolysis cells. *Int. J. Hydrog. Energy* **35**, 10039-10044 (2010).

41. Munoz-Garcia, A. B. *et al.* Unveiling structure-property relationships in $\text{Sr}_2\text{Fe}_{1.5}\text{Mo}_{0.5}\text{O}_{6-\delta}$, an electrode material for symmetric solid oxide fuel cells. *J. Am. Chem. Soc.* **134**, 6826-6833 (2012).
42. Ruan, C. & Xie, K. A redox-stable chromate cathode decorated with in situ grown nickel nanocatalyst for efficient carbon dioxide electrolysis. *Catal. Sci. Technol.* **5**, 1929-1940 (2015).
43. Qi, W. *et al.* Remarkable chemical adsorption of manganese-doped titanate for direct carbon dioxide electrolysis. *J. Mater. Chem. A* **2**, 6904-6915 (2014).
44. Li, Y., Li, P., Hu, B. & Xia, C. A nanostructured ceramic fuel electrode for efficient $\text{CO}_2/\text{H}_2\text{O}$ electrolysis without safe gas. *J. Mater. Chem. A* **4**, 9236-9243 (2016).
45. Duan, N. *et al.* Exploring $\text{Ni}(\text{Mn}_{1/3}\text{Cr}_{2/3})_2\text{O}_4$ spinel-based electrodes for solid oxide cells. *J. Mater. Chem. A* **8**, 3988-3998 (2020).
46. Duan, N. *et al.* Multi-functionalities enabled fivefold applications of $\text{LaCo}_{0.6}\text{Ni}_{0.4}\text{O}_{3-\delta}$ in intermediate temperature symmetrical solid oxide fuel/electrolysis cells. *Nano Energy* (2020).
47. Cao, Z. *et al.* Efficient electrolysis of CO_2 in symmetrical solid oxide electrolysis cell with highly active $\text{La}_{0.3}\text{Sr}_{0.7}\text{Fe}_{0.7}\text{Ti}_{0.3}\text{O}_3$ electrode material. *Electrochem. commun.* **69**, 80-83 (2016).
48. K ngas, R. Review—Electrochemical CO_2 Reduction for CO Production: Comparison of Low- and High-Temperature Electrolysis Technologies. *J. Electrochem. Soc.* **167** (2020).
49. Deka, D. J. *et al.* Investigation of hetero-phases grown via in-situ exsolution on a Ni-doped (La,Sr)FeO₃ cathode and the resultant activity enhancement in CO_2 reduction. *Appl. Catal. B* **286**, 119917 (2021).
50. Liu, C. *et al.* Enhancing CO_2 Catalytic Adsorption on an Fe Nanoparticle-Decorated $\text{LaSrFeO}_{4+\delta}$ Cathode for CO_2 Electrolysis. *ACS Appl. Mater. Interfaces* **13**, 8229–8238 (2021).
51. Hou, N. *et al.* A-Site Ordered Double Perovskite with in Situ Exsolved Core–Shell Nanoparticles as Anode for Solid Oxide Fuel Cells. *ACS Appl. Mater. Interfaces* **11**, 6995-7005 (2019).
52. Ravel, B. & Newville, M. ATHENA, ARTEMIS, HEPHAESTUS: data analysis for X-ray absorption spectroscopy using IFEFFIT. *J. Synchrotron Rad.* **12**, 537-541 (2005).
53. D’Orazio, A. C. *et al.* High temperature X-ray absorption spectroscopy of the local electronic structure and oxide vacancy formation in the $\text{Sr}_2\text{Fe}_{1.5}\text{Mo}_{0.5}\text{O}_{6-\delta}$ solid oxide fuel cell anode catalyst. *ACS Appl. Energy Mater.* **2**, 3061-3070 (2019).
54. Oishi, M., Sakuragi, T., Ina, T., Oshima, N. & Fujishiro, F. In situ evaluation of the electronic/local structure in B-site mixed perovskite-type oxide $\text{SrFe}_{0.6}\text{Mn}_{0.4}\text{O}_{3-\delta}$. *J. Solid State Chem.* **294**, 121893 (2021).
55. Park, S. *et al.* Ruddlesden–Popper oxide $(\text{La}_{0.6}\text{Sr}_{0.4})_2(\text{Co,Fe})\text{O}_4$ with exsolved CoFe nanoparticles for a solid oxide fuel cell anode catalyst. *Energy Technol.* **9**, 2100116 (2021).
56. Lv, H. *et al.* Infiltration of $\text{Ce}_{0.8}\text{Gd}_{0.2}\text{O}_{1.9}$ nanoparticles on $\text{Sr}_2\text{Fe}_{1.5}\text{Mo}_{0.5}\text{O}_{6-\delta}$ cathode for CO_2 electroreduction in solid oxide electrolysis cell. *J. Energy Chem.* **35**, 71-78 (2019).
57. Yue, X. & Irvine, J. T. S. Alternative Cathode Material for CO_2 Reduction by High Temperature Solid Oxide Electrolysis Cells. *J. Electrochem. Soc.* **159**, F442-F448 (2012).
58. Litzelman, S. J. & Tuller, H. L. Measurement of mixed conductivity in thin films with microstructured Hebb–Wagner blocking electrodes. *Solid State Ionics* **180**, 1190-1197 (2009).
59. Xie, Y., Xiao, J., Liu, D., Liu, J. & Yang, C. Electrolysis of carbon dioxide in a solid oxide electrolyzer with silver-gadolinium-doped ceria cathode. *J. Electrochem. Soc.* **162**, F397-F402 (2015).
60. Lai, K.-Y. & Manthiram, A. Evolution of exsolved nanoparticles on a perovskite oxide surface during a redox process. *Chem. Mater.* **30**, 2838-2847 (2018).
61. Laguna-Bercero, M. A., Campana, R., Larrea, A., Kilner, J. A. & Orera, V. M. Electrolyte degradation in anode supported microtubular yttria stabilized zirconia-based solid oxide steam electrolysis cells at high voltages of operation. *J. Power Sources* **196**, 8942-8947 (2011).

62. Hu, S. *et al.* Alkaline-earth elements (Ca, Sr and Ba) doped LaFeO_{3-δ} cathodes for CO₂ electroreduction. *J. Power Sources* **443**, 227268 (2019).
63. Liu, S., Liu, Q. & Luo, J.-L. CO₂-to-CO conversion on layered perovskite with in situ exsolved Co–Fe alloy nanoparticles: an active and stable cathode for solid oxide electrolysis cells. *J. Mater. Chem. A* **4**, 17521-17528 (2016).
64. Ma, W. *et al.* Promoting electrocatalytic CO₂ reduction to formate via sulfur-boosting water activation on indium surfaces. *Nat Commun.* **10**, 892 (2019).
65. Kwon, O. *et al.* Self-assembled alloy nanoparticles in a layered double perovskite as a fuel oxidation catalyst for solid oxide fuel cells. *J. Mater. Chem. A* **6**, 15947-15953 (2018).
66. Yang, X. *et al.* Enhancing stability and catalytic activity by in situ exsolution for high-performance direct hydrocarbon solid oxide fuel cell anodes. *Ind. Eng. Chem. Res.* **60**, 7826-7834 (2021).
67. O'hayre, R., Cha, S.-W., Colella, W. & Prinz, F. B. *Fuel cell fundamentals*. (John Wiley & Sons, 2016).
68. Park, S. *et al.* In situ exsolved Co nanoparticles on Ruddlesden-Popper material as highly active catalyst for CO₂ electrolysis to CO. *Appl. Catal. B* **248**, 147-156 (2019).

REVIEWER COMMENTS

Reviewer #1 (Remarks to the Author):

Reply to revision comment #1-A:

The authors' response to Comment #1-A satisfied the requirements. They explained the advantages of exsolved NPs on SOEC well. And the addition of the Table S3 highlighted the advantage of exsolution SOEC.

Reply to revision comment #1-B:

The authors' response to Comment #1-B nearly satisfied the requirements. They compared the SOEC stability of SFNM-red and SFNM-oxi in Figure S20. At first, the authors claimed that exsolution make support unstable, but SFNM-oxi was more unstable than SFNM+0.0Fe-red despite the absence of exsolution. It shows there are different degradation mechanisms for SFNM-red and SFNM-oxi. I think they should have discussed and compared the two degradation mechanisms more. But, at least, they just revealed the showed the degradation mechanism of SFNM-oxi in Figure S21, S22, S23.

Reply to revision comment #2:

The authors' response to Comment #2 satisfied the requirements. The addition of the SE-STEM and STEM-EDS images well supported the authors' point of view on the better stability for SFNM+1.2Fe-red.

Reply to revision comment #3:

The authors' response to Comment #3 satisfied the requirements. The idea of binding energy calculation was proper to verify [vacancy trapping] concept. They calculated the binding energy by DFT calculation and verified [vacancy trapping] concept well.

Reply to revision comment #4, 5:

The authors' response to Comment #4, 5 satisfied the requirements. They corrected their manuscript well.

Reviewer #2 (Remarks to the Author):

I have thoroughly checked each response to comments, and it seems that the authors have finely replied most of the comments. But, since Nature Communications publishes scientifically rigorous works with novel ideas, I feel much sorry for the harsh judgement, yet the detailed comments below should be clearly addressed or would not be suitable for the publication to Nature Communications.

1. The main point of my comment #1 is not well-addressed in my position. The Fig. 1 in this work seems to be much similar with other reported papers. Moreover, the topotactic ion exchange (TIE, or i.e., B-site supplement) has been already widely studied in previous works. So again, I want to ask the author whether the novelty of this work is just focused on developing SOEC cathode material via TIE method. The response to comment #1 part seems to be a lengthy but not clear response in my position.

2. The authors explained that the Fe source is not fully incorporated as B-site of perovskite oxides for SFM. This also indicates that B-site supplementary mechanism or TIE process can't be fully performed because of SrMoO₄ formation.

3. The OCV value seems to be about ~0.88 V at 70% CO₂/30% CO condition and ~0.1 V at pure CO₂ condition (Fig. R12). However, the OCV value in Fig. 3a and Supplementary Fig. 9 seems to be much low considering the experimental conditions for SOEC measurements (70% CO₂/30% CO condition), hence much different with other reported literatures regarding SOEC measurements.

4. The shape of the XAFS figure (Supplementary Fig. 17) is too weird in my position since the characteristic line for both Fe K-edge and Ni K-edge is not in coincidence with other previously reported papers. In addition, the references for XAFS measurements is not given in this work (e.g., Fe foil, Fe₂O₃, NiO, etc...). Furthermore, the fourier-transformed k³-weighted plot for the XAFS measurements is not plotted in this work.

5. Where are the XRD patterns of SFNM+1.2Fe-red after reducing at 70% CO₂/30% CO atmosphere, which is the real experimental conditions (Related with Comment #13)?

5. The authors stated in the "Authors' response to Comment #20" that the Fe-Ni alloy formation can be formed from the XRD and TEM-results in this work. However, in Fig. 2a₂, for the SFNM + 1.2 Fe-red, the small portion of Fe peak can be observed.

6. For Fig. 5c, the SFNM+0.0Fe-red-GDC (without B-site supplementary mechanism) seems to be much more stable than the SFNM+1.2Fe-red-GDC (with B-site supplementary mechanism) in my position since

the electrochemical performance for SFNM+1.2Fe-red-GDC linearly decreases. Nevertheless, the title is written as “boosting the stability of perovskites with exsolved nanoparticles by B-site supplement mechanism”. In my opinion, the title and the Fig. 5c does not correlate with each other.

7. I can't find the main purpose of “reoxidation” process in this work since many secondary phases are observed. In addition, the authors did not conduct any reoxidation process in terms of electrochemical performances.

8. Why did the authors conduct ECR measurements from 5% H₂/95% Ar to 50% CO₂/50% CO? The ECR measurements are usually conducted to measure the bulk diffusion of oxygen (D) and surface oxygen exchange coefficient (k) values, in which the oxygen partial pressure value should be precisely controlled to determine D and k values. Moreover, the D value between SFNM+0.0Fe-red and SFNM+1.2Fe-red seems to be similar in my position. Furthermore, the authors' main objective is to develop SOEC cathode (fuel electrode), not SOEC anode (air electrode), thus ECR measurement is not an important part in this study.

9. In my position, the designated planes in Supplementary Fig. 2 is not correct. In addition, in Supplementary Fig. 1, dispersion is written as ‘disepersion’.

10. In Supplementary Fig. 5, the exsolved particle diameter (or particle size) becomes larger with increasing ‘x’ value in SFNM + ‘x’ Fe-red. This would also affect the electrochemical performances of SOFC and/or SOEC.

11. In the TGA part (Supplementary Figure 14), the authors stated as “the gradual weight loss below 400 oC can be ascribed to ... and decomposition of nitrate”. Can the nitrates still exist after synthesis via sol-gel method plus reduction? Moreover, even though the authors wrote Kroger-Vink notation to explain the slope variations with increasing temperature, yet the clear reasons on each slope variations are not clearly elucidated in my position.

12. How did the authors confirm the exsolved nanoparticle as having Fe:Ni = 1:3 except for the XRD patterns? More clear evidence is required to confirm the ratio of Fe/Ni in Fe-Ni alloy.

13. At the DFT computational details in the methods part, the Gaussian smearing factor is 0.2 eV, which is much high compared to other DFT-conducted literatures. Also, what is the related orbital for Ni and Fe for U calculation? The U values for Ni and Fe seems to be different with other reported literatures.

14. I think that the equation for the calculation of CO production and FECO did not precisely consider the GC measurements.

Therefore, these points should be clearly elucidated or would not be suitable for publication in Nature Communications.

Reviewer #3 (Remarks to the Author):

In this version of manuscript, the authors have taken adequate reply for the questions I suggested in the previous review. A number of new explanations, and figures were added, which could improve the quality of the manuscript. This improved paper could be now accepted for publication in the Nature Communications.

Responses to Reviewer#2's comments

Note: Line X and Page Y are abbreviated as LX and PY in the following responses for simplicity.

Remarks to the Author:

I have thoroughly checked each response to comments, and it seems that the authors have finely replied most of the comments. But, since Nature Communications publishes scientifically rigorous works with novel ideas, I feel much sorry for the harsh judgement, yet the detailed comments below should be clearly addressed or would not be suitable for the publication to Nature Communications.

Authors' response:

We appreciate the amount of time and effort the reviewer #2 has put into helping us to refine this manuscript. We highly value each and every comment from the reviewer with our full respect and gratitude, since these suggestive comments are truly helpful in terms of improving the overall quality of our manuscript and enhancing its readability. Our detailed responses to each and every comment are listed as below.

Comment #1. The main point of my comment #1 is not well-addressed in my position. The Fig. 1 in this work seems to be much similar with other reported papers. Moreover, the topotactic ion exchange (TIE, or i.e., B-site supplement) has been already widely studied in previous works. So again, I want to ask the author whether the novelty of this work is just focused on developing SOEC cathode material via TIE method. The response to comment #1 part seems to be a lengthy but not clear response in my position.

Authors' response to Comment #1:

To avoid the similarity with other reported paper, the Fig. 1c (Fig. R1c) and the 3D schematics in Fig. 1d (Fig. R1d) in the original manuscript have been removed. The revised Fig. 1 in the revised manuscript is shown in Fig. R2 which mainly emphasises our new contribution of this work. As can be seen from the revised Fig. 1c, the difference between the scaffold structures of reduced SFNM with and without B-site supplement has been demonstrated.

Figure R1. Feasibility of B-site supplement calculated by DFT calculations and schematic illustrations of two exsolution process. (a) Co-segregation energy and schematic illustrations of the DFT models for co-segregation by conventional exsolution. (b) Exchange energy comparison of B-site cations of SFNM with guest Fe. (c) Schematic illustrations of the DFT models for exchange energy calculations and alloy formation. (d) Schematic illustration of TIE-assisted and conventional exsolution processes on SFNM.

Figure R2. (also Figure 1 in revised manuscript) Feasibility of B-site supplement calculated by DFT calculations and schematic illustrations of two exsolution process. (a) Co-segregation energy and schematic illustrations of the DFT models for co-segregation by conventional exsolution. (b) Exchange energy comparison of B-site cations of SFNM with guest Fe. (c) Schematic illustration of exsolution with and without B-site supplement on SFNM.

As we have summarized in the first round of point-by-point response to the comment #1, the novelty of this work is not just focused on the ion exchange method. Compared with the innovation of the specific synthetic methods that the reviewer is concerned about, our work focuses more on the creative ideas and solutions to address the stability issues of P-eNs. As far as we know, as of this writing, the concept of B-site supplement strategy has not been reported in any of the previous studies.

To make the answer simple and straightforward, we would like to re-summarize the significance of our work: Firstly, the robust structure of the perovskite scaffold is identified as the main factor to enhance the stability of P-eNs, which has been neglected in the previous literatures when investigating the stability of P-eNs^{1,2,3}. In this work, on the other hand, the B-site supplement strategy is proposed to enhance the structural stability of perovskite substrate, consequently leading to the higher stability of SFNM-based P-eNs. The degradation mechanism of conventional P-eNs and the role of B-site supplement strategy in boosting the stability of P-eNs have both been revealed by the intentionally accelerated degradation tests (ADT) of SFNM-based P-eNs with and without B-site supplement at high voltages. The outputs from this work can help to broaden the strategic thinking from the perspective of exsolved nanoparticles and the surface side reactions to the perovskite support

in order to improve the stability of P-eNs. This bears great significance for developing the advanced P-eNs with high catalytic activity and stability for CO₂ reduction and more broadly, for other energy storage and conversion systems.

Comment #2. The authors explained that the Fe source is not fully incorporated as B-site of perovskite oxides for SFM. This also indicates that B-site supplementary mechanism or TIE process can't be fully performed because of SrMoO₄ formation.

Authors' response to Comment #2:

We would like to clarify the B-site supplement and re-oxidation treatment on SFNM+xFe-red in this work.

Firstly, the surface-deposited Fe source is not only involved in the B-site supplement of perovskite scaffold, but also involved in the formation of the surface nanoparticles.

Secondly, the secondary SrMoO₄ phase is not formed during the ion exchange assisted exsolution, but is formed after the further reoxidation treatment, because it has been reported that the secondary phase SrMoO₄ would form in air-sintered Sr₂Fe_{1.5}Mo_{0.5}O_{6-δ} (SFM)^{4,5}. From an electroneutrality viewpoint, the high oxidation state of Fe (+2/+3) and Mo (+5/+6) at the B-site of SFM would result in an unbalanced positive charge, thus leading to a limited solubility of Mo at B-site of SFM. Consequently, Mo exists as a secondary phase SrMoO₄ by combining with Sr while Fe is completely dissolved in the perovskite lattice. To verify the extent to which the B-site of SFNM+xFe-red ($x=0.5, 0.8, 1.2$) was incorporated by the external Fe source, the SFNM+xFe-red ($x=0.0, 0.5, 0.8, 1.2$) was subjected to the reoxidation treatment. It demonstrates that SFNM+xFe-red has been sufficiently filled with foreign Fe and has formed a structure close to SFM when SrMoO₄ forms after the reoxidation treatment of SFNM+xFe-red. Fig.2b clearly shows that the SrMoO₄ forms during the reoxidation process of SFNM+0.8Fe-red and SFNM+1.2Fe-red, which in turn confirms that the B-site supplementary mechanism works out well on the SFNM+0.8Fe-red and SFNM+1.2Fe-red.

Comment #3. The OCV value seems to be about ~0.88 V at 70% CO₂/30% CO condition and ~0.1 V at pure CO₂ condition (Fig. R12). However, the OCV value in Fig. 3a and Supplementary Fig. 9 seems to be much low considering the experimental conditions for SOEC measurements (70% CO₂/30% CO condition), hence much different with other reported literatures regarding SOEC measurements.

Authors' response to Comment #3:

We did not utilize 70% CO₂/30% CO condition in this work. For Fig. 3a and Supplementary Fig. 9, the pure CO₂ was fed to the cathodic side during the CO₂ electrocatalysis process, as shown on L17, P32 in the Methods part

of the revised manuscript. Therefore, the OCV values shown in Fig. 3a and Supplementary Fig. 9 should be close to 0.1 V and are in consistence with those from previous work performed with pure CO₂ supply. Please refer to Fig. 5a in Reference 6 published in Nature communications, Fig. 4a in Reference 7 published in Advanced Energy Materials and Fig. 10a in Reference 8 published in Journal of Materials Chemistry A.

Comment #4. The shape of the XAFS figure (Supplementary Fig. 17) is too weird in my position since the characteristic line for both Fe K-edge and Ni K-edge is not in coincidence with other previously reported papers. In addition, the references for XAFS measurements is not given in this work (e.g., Fe foil, Fe₂O₃, NiO, etc...). Furthermore, the fourier-transformed k³-weighted plot for the XAFS measurements is not plotted in this work.

Authors' response to Comment #4:

The specific references mentioned as "other previously reported papers" are not included in the reviewer's Comment. Thus, we reviewed the literatures reporting the Fe *K*-edge and Ni *K*-edge X-ray absorption near-edge structure (XANES) spectra for the similar materials. We have found that the shape of Fe *K*-edge and Ni *K*-edge XANES spectra in our work are in consistence with those in the previous literatures^{5, 9-13}, please refer to the Fe *K*-edge data (Fig. 4) in Reference 12 and the Ni *K*-edge data (Fig. 3) in Reference 13, both published in Nature communications.

Per the reviewer's comments, the XANES data of the references (Fe foil, FeO, Fe₂O₃, Ni foil, NiO) have been added (Figs. R3a-b). The Fe *K*-edge and Ni *K*-edge extended X-ray absorption fine structure (EXAFS) data are also presented (Figs. R3c-d). From the Fe *K*-edge EXAFS spectra (Fig. R3c), it can be seen that the overlapping area between the peak of SFNM+1.2Fe-red and the peak of Fe foil is much larger than that between SFNM+0.0Fe-red and Fe foil, which is consistent with the XPS and XANES data. This can be ascribed to the participation of external Fe sources in the formation of Fe-Ni alloy nanoparticles. As seen from the Ni *K*-edge EXAFS spectra (Fig. R3d), the valence state of Ni element in SFNM-oxi is +2, while the peaks of SFNM+0.0Fe-red and SFNM+1.2Fe-red overlap with those of Ni foil and NiO. The Ni peaks of both reduced samples are similar, which is in consistence with the XPS and XANES data. It demonstrates that the amount of the Ni exsolution has almost reached the peak value via conventional exsolution due to its lower co-segregation energy. Therefore, combining the XANES, EXAFS results with the XPS data, the Fe and Ni exsolution of both samples can be well discussed.

Figure R3 (also Supplementary Figure 17). (a) Fe K-edge and (b) Ni K-edge X-ray absorption near-edge structure (XANES) spectra of SFNM-oxi, SFNM+0.0Fe-red and SFNM+1.2Fe-red, together with reference samples Fe-foil, FeO, Fe₂O₃, Ni-foil, and NiO. Fourier-transformed (c) Fe K-edge and (d) Ni K-edge extended X-ray absorption fine structure (EXAFS) spectra of SFNM-oxi, SFNM+0.0Fe-red and SFNM+1.2Fe-red, together with reference samples Fe-foil, FeO, Fe₂O₃, Ni-foil, and NiO.

We have revised the manuscript accordingly, as shown on L6-8, P17 of the revised manuscript and P23-24 of the revised supplementary information (SI).

Comment #5. Where are the XRD patterns of SFNM+1.2Fe-red after reducing at 70% CO₂/30% CO atmosphere, which is the real experimental conditions (Related with Comment #13)?

Authors' response to Comment #5:

Figure R4. XRD results of SFNM+0.0Fe and SFNM+1.2Fe after reduction at 70% CO₂/30% CO and 800 °C for 2h.

To address this comment, we conducted XRD analyses of SFNM+0.0Fe and SFNM+1.2Fe in 70% CO₂/30% CO atmosphere. The results are shown in Figure R4. The secondary phases of SrCO₃, SrMoO₄, magnetite, Fe₂O₃ and NiFe₂O₄ are detected, indicating that both materials have undergone severe phase decomposition. In particular, the formation of SrCO₃ may be attributed to the adsorption of CO₂ on the surface of perovskites, thereby causing other side reactions. These secondary phases are undesirable pre-reduction products and thus have a detrimental effect on the initial surface-activity of reduced perovskite. It is difficult to determine the composition of the exsolved nanoparticles from the XRD results, mainly due to the severe phase decomposition and the small amount of exsolution. On the contrary, the nanoparticles are well dispersed on the surface and the phase structure of perovskite scaffold can be well preserved after the exsolution in 5% H₂/95% N₂ atmosphere, which can deliver a higher initial catalytic activity. Therefore, the reducing atmospheres containing H₂ rather than the mixed CO₂/CO reducing atmospheres are selected as the pre-reduction condition of perovskite materials in this work and previous work.

Comment #5. The authors stated in the “Authors’ response to Comment #20” that the Fe-Ni alloy formation can be formed from the XRD and TEM-results in this work. However, in Fig. 2a2, for the SFNM + 1.2 Fe-red, the small portion of Fe peak can be observed.

Authors’ response to Comment #5:

Thanks to the reviewer for pointing this out. From the Fig. 2a₂, there may be a small amount of Fe that was present on the surface of the SFNM + 1.2 Fe-red, but it has not yet been determined that it existed in the form

of particles because no significant Fe nanoparticles were found by TEM-EDS results. It is likely due to the inevitable uneven Fe deposition during the lyophilization process or the inhomogeneous diffusion of external Fe source involved in Fe-Ni alloy nanoparticle growth process during ion exchange. Nonetheless, we believe that Fe as an active site would not affect the stability investigation of perovskite scaffold in this work^{14, 15}. Furthermore, compared to the guest ion deposition via infiltration and atom layer deposition (ALD) introduced in previous studies, the simple lyophilization process to deposit the guest ions on the surface offers its own unique advantages, i.e., more time-saving, low-cost and high yield, therefore, would expand the application of ion exchange technology in exsolution and show a great potential for its industrial utilizations in the future.

Comment #6. For Fig. 5c, the SFNM+0.0Fe-red-GDC (without B-site supplementary mechanism) seems to be much more stable than the SFNM+1.2Fe-red-GDC (with B-site supplementary mechanism) in my position since the electrochemical performance for SFNM+1.2Fe-red-GDC linearly decreases. Nevertheless, the title is written as “boosting the stability of perovskites with exsolved nanoparticles by B-site supplement mechanism”. In my opinion, the title and the Fig. 5c does not correlate with each other.

Authors' response to Comment #6:

The degradations of electrolyte and anode are inevitable during CO₂ electrocatalysis at 1.6 V and 850 °C, which may be the major reason for the linear decrease of the $j - t$ profile of SFNM+1.2Fe-red-GDC. In contrast to the linear decline in $j - t$ profile of SFNM+1.2Fe-red-GDC, the $j - t$ profile for SFNM+0.0Fe-red-GDC experienced a “degradation–reactivation–degradation” process. This change can be attributed to the fact that SFNM+0.0Fe-red-GDC undergoes a more severe phase decomposition with continuous exsolution, coke formation and formation of the secondary phases. This trend was not found in the $j - t$ profile of SFNM+1.2Fe-red-GDC, indicating that the phase decomposition of SFNM+1.2Fe-red-GDC has been significantly prevented by the B-site supplement.

Furthermore, the $j - t$ profile for SFNM+0.0Fe-red-GDC can be roughly categorized to Phase I (0-8 h), Phase II (8-35 h) and Phase III (35-55 h). Within Phase I, j experiences dramatic decrease with a degradation rate of 44 mA cm⁻² h⁻¹. For the Phase III, the j drops again until it becomes unstable at around 55 h. Especially, j declines at an attenuation rate of 6 mA cm⁻² h⁻¹ from 50 to 55 h. In contrast, SFNM+1.2Fe-red-GDC manifests a rather simple and mild degradation profile throughout the long-term stability measurement without observable j fluctuations, only having an average degradation rate of 3 mA cm⁻² h⁻¹. These results have been discussed on L1-14, P20 of the manuscript.

More importantly, the post-mortem SEM and TEM results in Fig. 5d-g, along with the Raman results in Fig. 6a-b, also prove that the structural stability of SFNM+1.2Fe-red-GDC is better than that of SFNM+0.0Fe-red-GDC after the accelerated degradation testing at 1.6 V and 850 °C.

Comment #7. I can't find the main purpose of "reoxidation" process in this work since many secondary phases are observed. In addition, the authors did not conduct any reoxidation process in terms of electrochemical performances.

Authors' response to Comment #7:

As discussed in our response to the Comment #2, considering the feature that the secondary phase SrMoO₄ would form in the air-sintered Sr₂Fe_{1.5}Mo_{0.5}O_{6-δ} (SFM)^{4,5}, the reoxidation treatment was used as a means of characterizing the extent of the B-site in SFNM+xFe-red (x=0.5, 0.8, 1.2) incorporated by the external Fe source. The results show that SFNM+xFe-red (x=0.5, 0.8, 1.2) have been sufficiently filled with foreign Fe and formed a structure close to SFM when SrMoO₄ forms after reoxidation treatment of SFNM+xFe-red (x=0.5, 0.8, 1.2). The representative literatures cited in the revised manuscript also explain the purpose of the re-oxidation treatment in details^{16,17}.

Three secondary phases have been shown in Fig. 2b, and the XRD PDF-card information of all secondary phases has also been provided.

The reoxidation treatment is not a sample treatment process prior to electrochemical performance measurements. Therefore, the electrochemical performances of re-oxidized samples were not conducted.

Comment #8. Why did the authors conduct ECR measurements from 5% H₂/95% Ar to 50% CO₂/50% CO? The ECR measurements are usually conducted to measure the bulk diffusion of oxygen (D) and surface oxygen exchange coefficient (k) values, in which the oxygen partial pressure value should be precisely controlled to determine D and k values. Moreover, the D value between SFNM+0.0Fe-red and SFNM+1.2Fe-red seems to be similar in my position. Furthermore, the authors' main objective is to develop SOEC cathode (fuel electrode), not SOEC anode (air electrode), thus ECR measurement is not an important part in this study.

Authors' response to Comment #8:

We thank the reviewer for this important comment. CO₂ electrolysis in SOEC consists of many steps, among which the transport of oxygens ion in the perovskite bulk is an important step. ECR experiment has become an effective way in evaluating the oxygen ions diffusion rate of the perovskite-based cathode candidates for CO₂ electrocatalysis^{8, 18}. The simulated bulk diffusion coefficient D_{chem} has been arguably regarded as an important indicator for comparing the diffusion of oxygen ions in various cathode candidates¹⁹.

As the reviewer pointed out, the oxygen partial pressure value should be precisely controlled. ECR measurements for perovskite-based cathodic candidates should be performed by switching between two mixed CO₂/CO atmospheres where the difference in oxygen partial pressures is less than 10 times. Therefore, the ECR

measurements in this work have been re-examined by changing the atmosphere from 2:1 CO-CO₂ ($P_{O_2} = 1.246 \times 10^{-18} \text{ atm}$ at 850 °C) to 1:1 CO-CO₂ ($P_{O_2} = 4.985 \times 10^{-18} \text{ atm}$) (Fig. R5). This atmosphere switching has been widely used in the ECR experiment of cathode materials for CO₂ electrolysis^{8,18}. The D_{chem} of SFNM+1.2Fe-red obtained by fitting the ECR curve is $3.747 \times 10^{-4} \text{ cm}^2 \text{ s}^{-1}$, much higher than $1.439 \times 10^{-5} \text{ cm}^2 \text{ s}^{-1}$ of the SFNM+0.0Fe-red. We have revised the manuscript, as shown on L18-19, P16, L1, P17 and L12-15, P31 of the revised manuscript.

Figure R5 (also Figure 4a). Electrical conductivity relaxation curves of SFNM+0.0Fe-red and SFNM+1.2Fe-red bars at 850 °C.

Comment #9. In my position, the designated planes in Supplementary Fig. 2 is not correct. In addition, in Supplementary Fig. 1, dispersion is written as ‘disepersion’.

Authors’ response to Comment #9:

We reviewed those literatures that had reported Sr₂Fe_{1.5}Mo_{0.5}O_{6-δ} based perovskites to confirm that the crystal plane information in this work is correct. We found that there are two main types of specified crystal plane information for Sr₂Fe_{1.5}Mo_{0.5}O_{6-δ} based perovskites in the previous literatures, one is Sr₂Fe_{1.5}Mo_{0.5}O_{6-δ} (ICSD-238948)^{7, 18,20,14, 21} and the other is Sr₂FeMoO_{6-δ} (PDF#01-070-4093)^{5,22}. The difference between the two types of crystal plane information has been shown in Fig. R6. Because the starting perovskite Sr₂Fe_{1.3}Ni_{0.2}Mo_{0.5}O_{6-δ} in this work can be seen as the Ni doped Sr₂Fe_{1.5}Mo_{0.5}O_{6-δ}, we choose the crystal plane information of Sr₂Fe_{1.5}Mo_{0.5}O_{6-δ} (ICSD-238948) as the reference.

Besides, we thank the reviewer for pointing out our typo. The wrong spelling, “disepersion”, has been corrected in the revised SI.

Figure R6. The difference in plane information between references $\text{Sr}_2\text{Fe}_{1.5}\text{Mo}_{0.5}\text{O}_{6-\delta}$ (ICSD-238948) and $\text{Sr}_2\text{FeMoO}_{6-\delta}$ (PDF#01-070-4093).

Comment #10. In Supplementary Fig. 5, the exsolved particle diameter (or particle size) becomes larger with increasing 'x' value in SFNM + 'x' Fe-red. This would also affect the electrochemical performances of SOFC and/or SOEC.

Authors' response to Comment #10:

We agree with the reviewer's observation. As shown in Supplementary Figs. 5a-d, the TIE-assisted procedure yields an average particle size of around 22 nm within a distribution range of 10-40 nm, which are evidently larger than the corresponding values of 15 nm and 8-22 nm in the SFNM+0.0Fe-red. The increase in the size of exsolved nanoparticles indeed has an impact on the catalytic performance, however, the population of exsolved nanoparticles also increases significantly by ion exchange (Supplementary Fig. 6), thereby expanding the surface-active area. From the current density-voltage profiles in Fig. 3a, the overall electrochemical performance of SFNM+1.2Fe-red is superior to that of SFNM+0.0Fe-red. It is in consistency with the results reported in the previous literatures on improving catalytic activity by ion exchange^{23,24}.

We believe that the changes in both the size and population of exsolved nanoparticles due to the ion-exchange-promoted exsolution contribute to the improved electrochemical performances of SOEC. The effects of the size and population of exsolved nanoparticles from SFNM+x Fe-red ($x=0.5, 0.8, 1.2$) on the electrochemical performance have been discussed on P8 of the revised SI.

Comment #11. In the TGA part (Supplementary Figure 14), the authors stated as “the gradual weight loss below 400 °C can be ascribed to ... and decomposition of nitrate”. Can the nitrates still exist after synthesis via sol-gel method plus reduction? Moreover, even though the authors wrote Kroger-Vink notation to explain the slope variations with increasing temperature, yet the clear reasons on each slope variations are not clearly elucidated in my position.

Authors' response to Comment #11:

As discussed on the P18-19 of the revised SI, the weight losses of all samples can be ascribed to the detachment of the surface adsorbates and loss of lattice oxygen of P-eNs. In particular, the lattice oxygen loss (formation of oxygen vacancy) above 400 °C is balanced by the valence state change of the B-site transition cations of perovskites²⁵. To show the valence state change of B-site cations due to the loss of lattice oxygen, the TGA curves of SFNM+0.0Fe-red and SFNM+1.2Fe-red are respectively segmented into two stages (Stage I :400-500 °C and Stage II: 500-800 °C) and four stages (Stage I: 400-450 °C, Stage II: 450-600 °C, Stage III: 600-700 °C, Stage IV:700-800 °C), respectively (Fig. R7). The XPS analyses are performed on Fe, Ni and Mo of both samples (Fe:0, +2 and +3, Ni: 0 and +2, Mo: +5 and +6) at the corresponding temperatures (Figs. R8 and R9), and the ratios of Fe, Ni, Mo in different valence states are calculated from the integral area ratios of the respective XPS spectra (Tables R1 and R2). According to the valence state changes of Fe, Ni and Mo, the reason for the loss of lattice oxygen at each stage for both samples can be clearly expressed by the Kroger-Vink notation (O_O^X is lattice oxygen, $V_O^{\cdot\cdot}$ is oxygen vacancy, Fe_{Fe}^X is Fe^{3+} , Fe'_{Fe} is Fe^{2+} , Ni_{Ni}^X is Ni^{2+} , Mo_{Mo}^X is Mo^{6+} , Mo'_{Mo} is Mo^{5+}):.

Figure R7. Segment of TGA curves of SFNM+0.0Fe-red and SFNM+1.2Fe-red for XPS analysis.

For SFNM+0.0Fe-red, the loss of lattice oxygen (formation of oxygen vacancies) can be mainly compensated by the valence reduction of Fe at the Stage I (400-500 °C), as the following:

Accordingly, at the Stage II (500-800 °C), it can be expressed as:

Figure R8. XPS results of (a) Fe, (b) Ni, and (c) Mo for SFNM+0.0Fe-red at 400, 500 and 800 °C.

Table R1. Relative composition of B-site cations in SFNM+0.0Fe-red calculated from respective XPS spectra.

B-site cations		Fe		Ni		Mo	
Valence states		+2	+3	0	+2	+5	+6
Temperatures (°C)	400	49%	51%	0	100%	0	100%
	500	54%	46%	0	100%	0	100%
	800	50%	50%	24%	76%	9%	91%

For SFNM+1.2Fe-red, the loss of lattice oxygen can be mainly compensated by the valence reduction of Fe at the Stage I (400-450 °C) as:

Accordingly, at the Stage II (450-600 °C), it can be expressed as:

At the Stage III (600-700 °C), it can be expressed as:

At the Stage IV (700-800 °C), it can be expressed as:

Figure R9. XPS results of (a) Fe (b) Ni (c) Mo for SFNM+1.2Fe-red at 400, 450, 600, 700 and 800 °C.

Table R2. Relative composition of B-site cations in SFNM+1.2Fe-red calculated from respective XPS spectra.

B-site cations Valence states	Fe			Ni		Mo		
	0	+2	+3	0	+2	+5	+6	
Temperatures (°C)	400	0	49.8%	50.2%	0	100%	0	100%
	450	0	56%	44%	0	100%	0	100%
	600	2%	60%	38%	32%	68%	0	100%
	700	3%	59%	38%	38%	62%	6%	94%
	800	6%	45%	49%	50.5%	49.5%	12%	88%

In addition, we thank the reviewer for pointing out the inappropriate term “decomposition of nitrate”. We have deleted the phrase of “and decomposition of nitrate” in the revised SI.

Comment #12. How did the authors confirm the exsolved nanoparticle as having Fe:Ni = 1:3 except for the XRD patterns? More clear evidence is required to confirm the ratio of Fe/Ni in Fe-Ni alloy.

Authors’ response to Comment #12:

In addition to the XRD results from this work, we have also presented the TEM (with EDS) results of the randomly selected nanoparticles on SFNM+0.0Fe-red (Supplementary Fig. 4a-c); the calculated average atom ratio of Fe/Ni in the alloy nanoparticles is 0.37 (Supplementary Table 1), which is very close to 1:3 (0.33).

The SFNM-similar perovskites have also been reported in the previous literatures, the exsolved FeNi₃ alloy nanoparticles have been reported after a similar reducing treatment in the Reference 26 titled “In situ exsolved FeNi₃ nanoparticles on Ni doped Sr₂Fe_{1.5}Mo_{0.5}O_{6-δ} perovskite for efficient electrochemical CO₂ reduction reaction” and Reference 22.

Comment #13. At the DFT computational details in the methods part, the Gaussian smearing factor is 0.2 eV, which is much high compared to other DFT-conducted literatures. Also, what is the related orbital for Ni and Fe for U calculation? The U values for Ni and Fe seems to be different with other reported literatures.

Authors’ response to Comment #13:

Our thanks to the reviewer for raising this question. The sigma values adopted by most literatures for the calculations of metal oxides/perovskites are 0.05, 0.1 and 0.2^{27,28, 29,30}. We performed three tests with the sigma values being 0.05, 0.1, and 0.2, to firstly optimize the geometric structure of Sr₄Fe₂Mo₂O₁₂ (SFMO), and then to perform the self-consistent calculations (calculation of the total energy), and the results are shown in the Table R3.

For all three sigma values tested (0.05, 0.01, and 0.2), the calculated total energy of the SFMO bulk is very similar (around -274.94 eV), however, the computation time increases as the sigma value decreases. Of note, for sigma = 0.05, the computation time increased 2.4-fold compared to the that of the sigma = 0.2. Based on the results, we chose sigma = 0.2 to obtain reliable value with the reasonable amount of time used. It is also worth mentioning that, in this study, our goal is to calculate the “energy difference” (i.e., exsolution energy, formation of oxygen vacancy) in order to help us to understand the entire reaction process. Therefore, choosing the sigma value of 0.2 and keeping the value consistent throughout the calculations are a reasonable and safe choice for the calculations performed in this study.

Table R3. Summary of the calculation results using different sigma values.

Sigma	energy	Time 1	Time 2	total time
0.05	-274.9364536	1576.708	857.614	2434.322
0.1	-274.9358404	625.569	483.113	1108.682
0.2	-274.9355251	550.028	460.322	1010.35

Note 1: Time 1 stands for the computational time for geometric optimization; Time 2 stands for the computational time for self-consistent calculations. Total time is the sum of Time 1 and Time 2. The unit for time is second. The pristine model used for the test was directly downloaded from Materials Project database.

For the U value, we considered the d-orbitals of Fe and Ni, and we chose the U values based on multiple previous literature results: ^{31,32,33} For example: Fe in LaFeO₃ (U_{Fe} = 5.1 eV) ³⁴, Fe in Li_xFeSiO₄ (U_{Fe} = 5 eV) ³¹, Fe in Sr₃Fe_{1.8}Co_{0.2}O_{7-δ} (U_{Fe} = 5.3) ³⁵, etc. In another study, Bouhafs et al. tested the U value from 2 to 8 eV in PrFeO₃, which is also a perovskite, and they showed that the calculational results match well with the experimental data (lattice parameters, magnetic moment, band gap, etc.) when using U_{Fe} = 5 eV, close to the value of 5.3 that we adopted in this study³³.

For Ni atom, the literatures with Ni in LaNiO₃ (U_{Ni} = 6.4 eV) ³², Ni in Li_xNiSiO₄ (U_{Ni} = 6 eV) ³¹, doped NiO (U_{Ni} = 6.4 eV) ³⁶ all showed a similar U value of Ni to the one adopted in this study.

Furthermore, not only from the previous literature results, Materials Project (Note 2) also uses the same U value for Fe and Ni as we adopted in this study (<https://materialsproject.org/tasks/mp-1173284#mp-1173284>, <https://materialsproject.org/tasks/mp-18940#mp-1292866>).

Therefore, we adopted U=5.3 eV and U=6.2 eV for Fe and Ni, respectively, and believe that this is suitable for the calculations performed in this study.

Note 2: The Materials Project is an open-access database offering material properties with the structures of more than 35,000 molecules and over 130,000 inorganic compounds. Details for DFT calculations, including the calculations input files for the geometrical optimizations and static calculations, are also provided.

Comment #14. I think that the equation for the calculation of CO production and FE_{CO} did not precisely consider the GC measurements.

Authors' response to Comment #14:

The equations we listed for the calculations of CO production and FE_{CO} are generally accepted ^{1, 37}. The volume concentration of CO (*v*) in the formula for calculating the FE_{CO} is obtained by GC measurement. To ensure the accuracy of this value obtained by GC testing, we had calibrated the GC measurement with standard gas (70% CO₂/30% CO) before the tests.

References

1. Liu, S.; Liu, Q.; Luo, J.-L., Highly stable and efficient catalyst with in situ exsolved Fe–Ni alloy nanospheres socketed on an oxygen deficient perovskite for direct CO₂ electrolysis. *ACS Catal.* **2016**, *6* (9), 6219-6228.
2. Opitz, A. K.; Nenning, A.; Rameshan, C.; Kubicek, M.; Gotsch, T.; Blume, R.; Havecker, M.; Knop-Gericke, A.; Rupprechter, G.; Klotzer, B.; Fleig, J., Surface chemistry of perovskite-type electrodes during high temperature CO₂ electrolysis investigated by operando photoelectron spectroscopy. *ACS Appl. Mater. Interfaces* **2017**, *9* (41), 35847-35860.
3. Ansari, H. M.; Bass, A. S.; Ahmad, N.; Birss, V. I., Unraveling the evolution of exsolved Fe–Ni alloy nanoparticles in Ni-doped La_{0.3}Ca_{0.7}Fe_{0.7}Cr_{0.3}O_{3-δ} and their role in enhancing CO₂–CO electrocatalysis. *J. Mater. Chem. A* **2021**.
4. Xi, X.; Liu, J.; Fan, Y.; Wang, L.; Li, J.; Li, M.; Luo, J.-L.; Fu, X.-Z., Reducing d-p band coupling to enhance CO₂ electrocatalytic activity by Mg-doping in Sr₂FeMoO_{6-δ} double perovskite. *Nano Energy* **2020**, *82*, 105707.
5. Xi, X.; Liu, J.; Luo, W.; Fan, Y.; Zhang, J.; Luo, J. L.; Fu, X. Z., Unraveling the enhanced kinetics of Sr₂Fe_{1+x}Mo_{1-x}O_{6-δ} electrocatalysts for high-performance solid oxide cells. *Adv. Energy Mater.* **2021**, *11* (48), 2102845.
6. Ye, L.; Zhang, M.; Huang, P.; Guo, G.; Hong, M.; Li, C.; Irvine, J. T.; Xie, K., Enhancing CO₂ electrolysis through synergistic control of non-stoichiometry and doping to tune cathode surface structures. *Nat. Commun.* **2017**, *8*, 14785.
7. Li, Y.; Li, Y.; Wan, Y.; Xie, Y.; Zhu, J.; Pan, H.; Zheng, X.; Xia, C., Perovskite oxyfluoride electrode enabling direct electrolyzing carbon dioxide with excellent electrochemical performances. *Adv. Energy Mater.* **2019**, *9* (3), 1803156.
8. Jiang, Y.; Yang, Y.; Xia, C.; Bouwmeester, H. J. M., Sr₂Fe_{1.4}Mn_{0.1}Mo_{0.5}O_{6-δ} perovskite cathode for highly efficient CO₂ electrolysis. *J. Mater. Chem. A* **2019**, *7*, 22939-22949.
9. D’Orazio, A. C.; Marshall, T.; Sultana, T.; Gerardi, J. K.; Segre, C. U.; Carlo, J. P.; Eigenbrodt, B. C., High temperature X-ray absorption spectroscopy of the local electronic structure and oxide vacancy formation in the Sr₂Fe_{1.5}Mo_{0.5}O_{6-δ} solid oxide fuel cell anode catalyst. *ACS Appl. Energy Mater.* **2019**, *2* (5), 3061-3070.
10. Oishi, M.; Sakuragi, T.; Ina, T.; Oshima, N.; Fujishiro, F., In situ evaluation of the electronic/local structure in B-site mixed perovskite-type oxide SrFe_{0.6}Mn_{0.4}O_{3-δ}. *J. Solid State Chem.* **2021**, *294*, 121893.
11. Beppu, K.; Hosokawa, S.; Teramura, K.; Tanaka, T., Oxygen storage capacity of Sr₃Fe₂O_{7-δ} having high structural stability. *J. Mater. Chem. A* **2015**, *3* (25), 13540-13545.
12. Kim, H.; Lim, C.; Kwon, O.; Oh, J.; Curnan, M. T.; Jeong, H. Y.; Choi, S.; Han, J. W.; Kim, G., Unveiling the key factor for the phase reconstruction and exsolved metallic particle distribution in perovskites. *Nat Commun* **2021**, *12* (1), 6814.
13. Li, Y.; Tan, X.; Hocking, R. K.; Bo, X.; Ren, H.; Johannessen, B.; Smith, S. C.; Zhao, C., Implanting Ni-O-VOx sites into Cu-doped Ni for low-overpotential alkaline hydrogen evolution. *Nat. Commun.* **2020**, *11* (1), 2720.
14. Chen, L.; Xu, J.; Wang, X.; Xie, K., Sr₂Fe_{1.5+x}Mo_{0.5}O_{6-δ} cathode with exsolved Fe nanoparticles for enhanced CO₂ electrolysis. *Int. J. Hydrog. Energy.* **2020**, *45* (21), 11901-11907.
15. Jiang, Y.; Ye, L.; Zhang, S.; Xia, C., Doped ceria with exsolved Fe⁰ nanoparticles as a Sr-free cathode for CO₂ electrolysis in SOECs at reduced temperatures. *J. Mater. Chem. A* **2022**, *10*, 9380-9383.
16. Neagu, D.; Papaioannou, E. I.; Ramli, W. K. W.; Miller, D. N.; Murdoch, B. J.; Menard, H.; Umar, A.; Barlow, A. J.; Cumpson, P. J.; Irvine, J. T. S.; Metcalfe, I. S., Demonstration of chemistry at a point through restructuring and catalytic activation at anchored nanoparticles. *Nat. Commun.* **2017**, *8* (1), 1855.
17. Lai, K.-Y.; Manthiram, A., Evolution of exsolved nanoparticles on a perovskite oxide surface during a redox process. *Chem. Mater.* **2018**, *30* (8), 2838-2847.

18. Li, Y.; Chen, X.; Yang, Y.; Jiang, Y.; Xia, C., Mixed-Conductor $\text{Sr}_2\text{Fe}_{1.5}\text{Mo}_{0.5}\text{O}_{6-\delta}$ as Robust Fuel Electrode for Pure CO_2 Reduction in Solid Oxide Electrolysis Cell. *ACS Sustain. Chem. Eng.* **2017**, *5* (12), 11403-11412.
19. Li, Y.; Yu, L.; Yu, Y.; Maliutina, K.; Wu, Q.; He, C.; Fan, L., Understanding CO_2 electrochemical reduction kinetics of mixed-conducting cathodes by the electrical conductivity relaxation method. *Int. J. Hydrog. Energy* **2020**, *46* (15), 9646-9652.
20. Li, Y.; Singh, M.; Zhuang, Z.; Jing, Y.; Li, F.; Maliutina, K.; He, C.; Fan, L., Efficient reversible CO/CO_2 conversion in solid oxide cells with a phase-transformed fuel electrode. *Sci. China Mater.* **2020**, *64* (5), 1114-1126.
21. Dai, N.; Feng, J.; Wang, Z.; Jiang, T.; Sun, W.; Qiao, J.; Sun, K., Synthesis and characterization of B-site Ni-doped perovskites $\text{Sr}_2\text{Fe}_{1.5-x}\text{Ni}_x\text{Mo}_{0.5}\text{O}_{6-\delta}$ ($x = 0, 0.05, 0.1, 0.2, 0.4$) as cathodes for SOFCs. *J. Mater. Chem. A* **2013**, *1* (45).
22. Du, Z.; Zhao, H.; Yi, S.; Xia, Q.; Gong, Y.; Zhang, Y.; Cheng, X.; Li, Y.; Gu, L.; Swierczek, K., High-performance anode material $\text{Sr}_2\text{FeMo}_{0.65}\text{Ni}_{0.35}\text{O}_{6-\delta}$ with in situ exsolved nanoparticle catalyst. *ACS Nano* **2016**, *10* (9), 8660-9.
23. Joo, S.; Kwon, O.; Kim, K.; Kim, S.; Kim, H.; Shin, J.; Jeong, H. Y.; Sengodan, S.; Han, J. W.; Kim, G., Cation-swapped homogeneous nanoparticles in perovskite oxides for high power density. *Nat. Commun.* **2019**, *10* (1), 697.
24. Joo, S.; Seong, A.; Kwon, O.; Kim, K.; Lee, J. H.; Gorte, R. J.; Vohs, J. M.; Han, J. W.; Kim, G., Highly active dry methane reforming catalysts with boosted in situ grown Ni-Fe nanoparticles on perovskite via atomic layer deposition. *Sci. Adv.* **2020**, *6* (35), eabb1573.
25. Sun, Y. F.; Li, J. H.; Cui, L.; Hua, B.; Cui, S. H.; Li, J.; Luo, J. L., A-site-deficiency facilitated in situ growth of bimetallic Ni-Fe nano-alloys: a novel coking-tolerant fuel cell anode catalyst. *Nanoscale* **2015**, *7* (25), 11173-81.
26. Lv, H.; Lin, L.; Zhang, X.; Gao, D.; Song, Y.; Zhou, Y.; Liu, Q.; Wang, G.; Bao, X., In situ exsolved FeNi_3 nanoparticles on nickel doped $\text{Sr}_2\text{Fe}_{1.5}\text{Mo}_{0.5}\text{O}_{6-\delta}$ perovskite for efficient electrochemical CO_2 reduction reaction. *J. Mater. Chem. A* **2019**, *7* (19), 11967-11975.
27. Kim, K.; Joo, S.; Huang, R.; Kim, H. J.; Kim, G.; Han, J. W., Mechanistic insights into the phase transition and metal ex-solution phenomena of $\text{Pr}_{0.5}\text{Ba}_{0.5}\text{Mn}_{0.85}\text{Co}_{0.15}\text{O}_{3-\delta}$ from simple to layered perovskite under reducing conditions and enhanced catalytic activity. *Energy Environ. Sci.* **2021**, *14* (2), 873-882.
28. Xing, J.; Zhao, Y.; Askerka, M.; Quan, L. N.; Gong, X.; Zhao, W.; Zhao, J.; Tan, H.; Long, G.; Gao, L.; Yang, Z.; Voznyy, O.; Tang, J.; Lu, Z. H.; Xiong, Q.; Sargent, E. H., Color-stable highly luminescent sky-blue perovskite light-emitting diodes. *Nat Commun* **2018**, *9* (1), 3541.
29. García Pintos, D.; Juan, A.; Irigoyen, B., Mn-Doped CeO_2 : DFT+U Study of a Catalyst for Oxidation Reactions. *The Journal of Physical Chemistry C* **2013**, *117* (35), 18063-18073.
30. Mao, L.; Kennard, R. M.; Traore, B.; Ke, W.; Katan, C.; Even, J.; Chabiny, M. L.; Stoumpos, C. C.; Kanatzidis, M. G., Seven-Layered 2D Hybrid Lead Iodide Perovskites. *Chem* **2019**, *5* (10), 2593-2604.
31. Wu, S. Q.; Zhu, Z. Z.; Yang, Y.; Hou, Z. F., Structural stabilities, electronic structures and lithium deintercalation in Li_xMSiO_4 ($M=\text{Mn, Fe, Co, Ni}$): A GGA and GGA+U study. *Comput. Mater. Sci.* **2009**, *44* (4), 1243-1251.
32. Huang, W. L.; Zhu, Q.; Ge, W.; Li, H., Oxygen-vacancy formation in LaMO_3 ($M=\text{Ti, V, Cr, Mn, Fe, Co, Ni}$) calculated at both GGA and GGA+U levels. *Comput. Mater. Sci.* **2011**, *50* (5), 1800-1805.
33. Rezaiguia, M.; Benstaali, W.; Abbad, A.; Bentata, S.; Bouhafs, B., GGA + U Study of Electronic and Magnetic Properties of $\text{Pr}(\text{Fe}/\text{Cr})\text{O}_3$ Cubic Perovskites. *J. Supercond. Nov. Magn.* **2017**, *30* (9), 2581-2590.

34. Hong, J.; Stroppa, A.; Íñiguez, J.; Picozzi, S.; Vanderbilt, D., Spin-phonon coupling effects in transition-metal perovskites: A DFT + U and hybrid-functional study. *Phys. Rev. B* **2012**, *85* (5).
35. Huan, D.; Wang, Z.; Wang, Z.; Peng, R.; Xia, C.; Lu, Y., High-Performanced Cathode with a Two-Layered R-P Structure for Intermediate Temperature Solid Oxide Fuel Cells. *ACS Appl. Mater. Interfaces* **2016**, *8* (7), 4592-9.
36. Kropp, T.; Lu, Z.; Li, Z.; Chin, Y.-H. C.; Mavrikakis, M., Anionic Single-Atom Catalysts for CO Oxidation: Support-Independent Activity at Low Temperatures. *ACS Catalysis* **2019**, *9* (2), 1595-1604.
37. Xi, X.; Fan, Y.; Zhang, J.; Luo, J.-L.; Fu, X.-Z., In situ construction of hetero-structured perovskite composites with exsolved Fe and Cu metallic nanoparticles as efficient CO₂ reduction electrocatalysts for high performance solid oxide electrolysis cells. *J. Mater. Chem. A* **2022**, *10*, 2509-2518.

Reviewers' comments

Reviewer #2 (Remarks to the Author):

I have thoroughly checked all the comments and felt that the authors have much endeavored to address all the comments in both the first and second rounds. Unfortunately, I am sorry for the harsh judgement, but since some of the comments are not well-addressed and flaws exist, I think this paper is not suitable for Nature Communications. Some of the points that I thought as not well-addressed are as follows:

1. The SrMoO₄ phase is formed before the reduction process (i.e., after sintered in air), which implies that the SFNM is not fully synthesized. In addition, there exists many impurities for both SFNM+1.2Fe-red and SFNM+0.0Fe-red (please refer to Figure R4) after reduction at CO-CO₂ mixed atmosphere. I think there are too many impurities to correlate with Figure 6c.
2. I agree that the OCV value should be close to 0.1 V at pure CO₂ atmosphere. However, the condition is not listed in the figure caption related to electrochemical performance measurements, making me more confused. In addition, as I have mentioned in comment #1, many impurities exist after reduction at 70% CO₂/30% CO atmosphere (much oxidizing atmosphere) only after 2 hours. Yet, the condition for electrochemical performance measurement is 100% CO₂, which is much close to oxidizing atmosphere compared to 70%CO₂/30% CO atmosphere, which may lead to more impurities.
3. I also know the purpose of re-oxidation process and agree that the reference 17 given in the rebuttal letter have well-discussed of the purpose of re-oxidation process. However, the purpose of re-oxidation process can't be fulfilled for samples with B-site supplement process in my position: B-site vacancies should exist for redox-reversible samples, but for the samples with B-site supplement process, the B-site vacancies does not exist, indicating that the re-oxidation process can't be fully performed for SFNM+1.2Fe-red. Therefore, since the main sample of this study with B-site supplement process (SFNM+1.2Fe-red) can't fulfill the re-oxidation process, the purpose of re-oxidation process can't be well-addressed for the main sample.
4. I think that the plane designation for Figure R6 is much weird in my position. How can the (610) plane exist before (110) plane for Sr₂Fe_{1.5}Mo_{0.5}O_{6-d}?
5. Even though there exists some discrepancies on the Fe/Ni ratio for SFNM+0.0Fe-red, I think that the authors can state that the Fe/Ni ratio as Fe:Ni = 1:3. However, for the SFNM+1.2Fe-red material, the Fe/Ni ratio seems to be much weird and different with SFNM+0.0Fe-red. There exists discrepancies for Fe/Ni ratio in SFNM+1.2Fe-red (from 1.98~3.05), implying that the Fe:Ni can be 2:1 to 3:1 (quite different from SFNM+0.0Fe-red). Moreover, the Fe/Ni ratio may significantly affect the electrochemical performance, but the reasons why the Fe/Ni ratio is different between SFNM+0.0Fe-red and SFNM+1.2Fe-red is not listed.
6. The authors stated that the equations listed for Faradaic efficiency calculation for CO production is generally accepted. Unfortunately, two references given for the Faradaic efficiency calculation of CO production are the works from your laboratory, in which the authors should be more careful in using the phrase "generally accepted". If the current density for the SOEC stability test decreases much, the Faradaic efficiency should also decrease in the same ratio assuming that the condition for the electrochemical measurement is well-maintained.
7. I have an inquiry about the oxygen partial pressure given for the 2:1 CO-CO₂ and 1:1 CO-CO₂ atmospheres. Where did the authors refer or calculate the oxygen partial pressure of CO-CO₂ mixed atmospheres?

In these regards, again, sorry for the harsh judgement, but I have to say that this paper is not

suitable for Nature Communications. It would be grateful to consider our comments for the next submission.

Reviewer #2 (Remarks to the Author):

I have thoroughly checked all the comments and felt that the authors have much endeavored to address all the comments in both the first and second rounds. Unfortunately, I am sorry for the harsh judgement, but since some of the comments are not well-addressed and flaws exist, I think this paper is not suitable for Nature Communications. Some of the points that I thought as not well-addressed are as follows:

Authors' response:

We thank the Reviewer #2 for these comments, our detailed responses to each comment are as follows.

Comment #1. The SrMoO_4 phase is formed before the reduction process (i.e., after sintered in air), which implies that the SFNM is not fully synthesized. In addition, there exists many impurities for both SFNM+1.2Fe-red and SFNM+0.0Fe-red (please refer to Figure R4) after reduction at CO-CO₂ mixed atmosphere. I think there are too many impurities to correlate with Figure 6c.

Authors' response to Comment #1:

We have never provided relevant data of characterization of the phase of SFNM (i.e., XRD) in the manuscript, supporting information, and all the responses in the 1st and 2nd rounds. We now provide the XRD results here which clearly show that no SrMoO_4 phase was formed on SFNM after sintering in air. (Fig. R1). Therefore, the statement "The SrMoO_4 phase is formed before the reduction process (i.e., after sintered in air)" is not fact-based.

Figure R1. XRD result of SFNM after sintering in air at 1000 °C for 5h (SFNM-oxi).

We would like to clarify once again that 5% H₂/95% N₂ was used to prepare our cathode samples for CO₂ electrocatalysis in this work. CO₂-CO mixed atmosphere was used only to answer to the reviewer’s previous question: “why the pre-reduction of all samples was carried out in 5% H₂/95% N₂ in this work”. The reviewer mixed up the results of these two systems.

Although the SFNM+0.0Fe and SFNM+1.2Fe produced many impurities after reduction in CO₂-CO mixed atmosphere (Fig. R4), the SFNM+0.0Fe-red and SFNM+1.2Fe-red used in this work were pre-reduced in 5% H₂/95% N₂ and no impurities were found in SFNM+0.0Fe-red and SFNM+1.2Fe-red (Fig. 2a). Therefore, it is illogical to correlate the samples after reduction in CO₂-CO mixed atmosphere in Fig. R4 with the samples treated in 5% H₂/95% N₂ in Figure 6c.

Comment #2. I agree that the OCV value should be close to 0.1 V at pure CO₂ atmosphere. However, the condition is not listed in the figure caption related to electrochemical performance measurements, making me more confused. In addition, as I have mentioned in comment #1, many impurities exist after reduction at 70% CO₂/30% CO atmosphere (much oxidizing atmosphere) only after 2 hours. Yet, the condition for electrochemical performance measurement is 100% CO₂, which is much close to oxidizing atmosphere compared to 70% CO₂/30% CO atmosphere, which may lead to more impurities.

Authors’ response to Comment #2:

The experimental condition (pure CO₂ atmosphere) has been added in the caption of Fig. 3a. Furthermore, only pure CO₂ was fed to cathode side for electrolysis in the entire article. On Lines 17-18, Page 32, we clearly stated that “During the electrolysis, pure CO₂ was fed

to the cell with a flow rate of 100 mL min⁻¹ via the cathode compartment located at the bottom”, which will not cause confusion to the readers.

The reviewer again mixed up the 5% H₂/95% N₂ pre-reduction condition in this work with 70% CO₂/30% CO reduction condition used to answer the reviewer's previous questions. There are indeed many impurities that appeared after the reduction in 70% CO₂/30% CO atmosphere. However, the cathode materials for CO₂ electrocatalysis in this work were prepared in 5% H₂/95% N₂, and no impurities were found in SFNM+0.0Fe-red and SFNM+1.2Fe-red. After pre-reduction in 5% H₂/95% N₂, a more stable perovskite scaffold was formed on SFNM+1.2Fe-red compared to SFNM+0.0Fe-red, which significantly alleviated the phase decomposition during the pure CO₂ electrocatalysis.

Comment #3. I also know the purpose of re-oxidation process and agree that the reference 17 given in the rebuttal letter have well-discussed of the purpose of re-oxidation process. However, the purpose of re-oxidation process can't be fulfilled for samples with B-site supplement process in my position: B-site vacancies should exist for redox-reversible samples, but for the samples with B-site supplement process, the B-site vacancies does not exist, indicating that the re-oxidation process can't be fully performed for SFNM+1.2Fe-red. Therefore, since the main sample of this study with B-site supplement process (SFNM+1.2Fe-red) can't fulfill the re-oxidation process, the purpose of re-oxidation process can't be well-addressed for the main sample.

Authors' response to Comment #3:

The statement “B-site vacancies should exist for redox-reversible samples, but for the samples with B-site supplement process, the B-site vacancies do not exist, indicating that the re-oxidation process cannot be fully performed for SFNM+1.2Fe-red” is logically reversed. The reviewer was using the conclusions drawn from the XRD results of the reoxidation treatment (no B-site vacancies on SFNM+1.2Fe-red) to refute the purpose of the reoxidation on SFNM+1.2Fe-red.

The purpose of reoxidation on all the reduced samples is to investigate at which concentration of guest Fe ion the B-site of the sample is fully supplemented after reduction, and whether the substrate has been transformed into a more stable Sr₂Fe_{1.5}Mo_{0.5}O_{6-δ} (SFM) structure. Because it has been reported that the secondary phase SrMoO₄ would form in air-sintered SFM¹, the presence of the SrMoO₄ phase on SFNM+1.2Fe-red after reoxidation indicates that SFNM+1.2Fe-red before reoxidation is similar to SFM. The results directly verify that no B-site vacancies exist on SFNM+1.2Fe-red, and the B-site is almost occupied by Fe and Mo, resulting in a more stable perovskite scaffold.

Comment #4. I think that the plane designation for Figure R6 is much weird in my position. How can the (610) plane exist before (110) plane for Sr₂Fe_{1.5}Mo_{0.5}O_{6-d}?

Authors' response to Comment #4:

We thank the reviewer for pointing out the typo. The plane has been corrected to (001) in the revised manuscript.

Comment #5. Even though there exists some discrepancies on the Fe/Ni ratio for SFNM+0.0Fe-red, I think that the authors can state that the Fe/Ni ratio as Fe:Ni = 1:3. However, for the SFNM+1.2Fe-red material, the Fe/Ni ratio seems to be much weird and different with SFNM+0.0Fe-red. There exists discrepancies for Fe/Ni ratio in SFNM+1.2Fe-red (from 1.98~3.05), implying that the Fe:Ni can be 2:1 to 3:1 (quite different from SFNM+0.0Fe-red). Moreover, the Fe/Ni ratio may significantly affect the electrochemical performance, but the reasons why the Fe/Ni ratio is different between SFNM+0.0Fe-red and SFNM+1.2Fe-red is not listed.

Authors' response to Comment #5:

The difference in the exsolved Fe-Ni alloy composition between SFNM+0.0Fe-red and SFNM+1.2Fe-red can be ascribed to the two different exsolution processes (conventional exsolution and ion exchange assisted exsolution). For the formed Fe-Ni alloy nanoparticles on SFNM+1.2Fe-red, the Fe sources are not only from the bulk diffusion of Fe cation, but also from the surface diffusion of guest Fe ions, which causes the higher Fe concentration in Fe-Ni alloy compared to that on SFNM+0.0Fe-red. From the point of view of the Fe-Ni phase diagram, the Fe-Ni alloys with an Fe/Ni ratio in the range of 2:1 to 3:1 are thermodynamically stable and exist in an FCC structure at 850 °C².

In addition, the reason for the difference in the Fe-Ni alloy composition between SFNM+0.0Fe-red and SFNM+1.2Fe-red has already been explained on Lines 10-12, Page 9 of the manuscript.

“Transmission electron microscopy (TEM) with energy dispersive X-ray spectroscopy (EDS) element mappings on randomly selected nanoparticles of SFNM+0.0Fe-red and SFNM+1.2Fe-red reveal the significant increase of Fe proportion in the exsolved nanoparticles of SFNM+1.2Fe-red (Supplementary Fig. 4 and Supplementary Table 1). It may be presumably explained by the involvement of guest Fe in the nanoparticle growth, and the slightly smaller electronegativity of Fe than that of Ni causes the lattice expansion of Fe-Ni alloy.”

Comment #6. The authors stated that the equations listed for Faradaic efficiency calculation for CO production is generally accepted. Unfortunately, two references given for the Faradaic efficiency calculation of CO production are the works from your laboratory, in which the authors should be more careful in using the phrase “generally accepted”. If the current density for the SOEC stability test decreases much, the Faradaic efficiency should also decrease in the same ratio assuming that the condition for the electrochemical measurement is well-maintained.

Authors’ response to Comment #6:

We thank the reviewer for pointing out the phrase “generally accepted”. The listed formula for calculating the Faraday efficiency of CO is derived from the general Faraday efficiency formula. The steps to convert from the general Faraday efficiency formula to the specific Faraday efficiency formula for producing CO from CO₂ reduction have been described extensively in the two references we cited (as shown in the following equations).

$$FE_j = \frac{nFvF_m}{I_{total}} \times 100\%,$$

Where n is the number of electrons exchanged, F is Faraday’s constant ($F = 96487 \text{ C/mol}$), v is the mole fraction of the product gas in the gaseous mixture analyzed (also equals to the volume fraction if gases are assumed to be ideal), F_m is the molar flow rate in mol/s, and I_{total} is the total current.

The molar flow rate is derived from the volume flow rate V by the relation $F_m = p_0V/RT$, with p_0 the atmospheric pressure (Pa), R the ideal gas constant ($R = 8.314 \text{ J/mol K}$), and T the temperature in K. Therefore, the Faraday efficiency for producing CO (FE_{CO}) can be expressed as follow:

$$FE_{CO} = \frac{2Fvp_0V}{RT_0I_{total}} \times 100\%$$

$$FE_{CO} = \frac{2 \times 96485 \left(\frac{C}{mol}\right) \times V \left(\frac{m^3}{s}\right) \times v(vol\%) \times 1.01 \times 10^5 \left(\frac{N}{m^2}\right)}{8.314 \left(\frac{N \cdot m}{mol \cdot K}\right) \times 298.15(K) \times I_{total} \left(\frac{C}{s}\right)} \times 100\%$$

$$FE_{CO} = \frac{2 \times 96485 \left(\frac{C}{mol}\right) \times V \left(\frac{mL}{min}\right) \times 10^{-6} \left(\frac{m^3}{mL}\right) \times v(vol\%) \times 1.01 \times 10^5 \left(\frac{N}{m^2}\right)}{8.314 \left(\frac{N \cdot m}{mol \cdot K}\right) \times 298.15(K) \times I_{total} \left(\frac{C}{s}\right) \times 60 \left(\frac{s}{min}\right)} \times 100\%$$

where the unit of V is mL/min. Then,

$$FE_{CO} = \frac{0.1315 \times V \left(\frac{mL}{min}\right) \times v(vol\%)}{I_{total}(A)} \times 100\%$$

where $V \left(\frac{mL}{min}\right)$ = Gas flow rate measured by a flow meter at the exit of the cell at room temperature and under ambient pressure.

$v(vol\%)$ = Volume concentration of CO in the exhaust gas from the cell (obtained by gas chromatography).

$I_{total}(A)$ = cell current during short-term stability experiments.

Faraday efficiency describes the efficiency with which charge (electrons) is transferred in a system facilitating an electrochemical reaction. Therefore, FE_{CO} is determined by many factors, including the current density, the concentration of CO among the products, as shown in the listed Faraday efficiency formula.

The statement of “If the current density for the SOEC stability test decreases much, the Faradaic efficiency should also decrease in the same ratio assuming that the condition for the electrochemical measurement is well-maintained” by the reviewer is lax and ignores the expression of objective facts. And we had already explained the relationship between the FE_{CO} and current density in Comment #18 of the first round of review.

Comment #7. I have an inquiry about the oxygen partial pressure given for the 2:1 CO-CO₂ and 1:1 CO-CO₂ atmospheres. Where did the authors refer or calculate the oxygen partial pressure of CO-CO₂ mixed atmospheres?

Authors’ response to Comment #7:

The oxygen partial pressure of mixed CO₂-CO atmosphere can be derived from the relationship between the change of the standard Gibbs free energy (ΔG°) and the reaction equilibrium constant (K)⁵. Because Δh° and Δs° determine the magnitude of ΔG° , and K is

a measure of the ratio of the concentrations of products to the concentrations of reactants, we can express K in terms of Δg° .

For reaction:

Table R1. Thermodynamic parameters for above reaction.

Species	Δh° (kJ/mol)	s° (J/mol · K)
CO	-110.53	197.66
CO ₂	-393.51	213.79
O ₂	0	205

$$\Delta h^\circ = -2 \times 110.53 + 0 + 2 \times 393.51 = 565.96 \text{ kJ/mol}$$

$$s^\circ = 2 \times 197.66 + 205 - 2 \times 213.79 = 172.74 \text{ J/mol} \cdot K$$

$$\Delta g^\circ(850^\circ\text{C}) = \Delta h^\circ - Ts^\circ = 565960 \text{ J/mol} - 1123 \times 172.74 \text{ J/mol} = 371972.98 \text{ J/mol}$$

The standard Gibbs free energy Δg° is related to the equilibrium constant K as follows:

$$\Delta g^\circ(T) = -RT \ln K = -RT \ln \frac{(p_{CO}/p^\theta)^2 \cdot (p_{O_2}/p^\theta)}{(p_{CO_2}/p^\theta)^2}$$

$$\Delta g^\circ(850^\circ\text{C}) = -8.314 \times 1123.15 \times \ln \frac{p_{CO}^2 \cdot p_{O_2}}{p_{CO_2}^2 \cdot p^\theta}$$

When 2:1 CO-CO₂,

$$371972.98 = -9337.8691 \times \ln \frac{4 \times p_{O_2}}{p^\theta}$$

$$-39.84 = \ln \frac{4 \times p_{O_2}}{p^\theta}$$

$$p_{O_2} = 1.246 \times 10^{-18} \text{ atm}$$

When 1:1 CO-CO₂,

$$-39.84 = \ln \frac{p_{O_2}}{p^\theta}$$

$$p_{O_2} = 4.985 \times 10^{-18} \text{ atm}$$

References:

1. Wang, Z.; Tian, Y.; Li, Y., Direct CH₄ fuel cell using Sr₂FeMoO₆ as an anode material. *Journal of Power Sources* **2011**, 196 (15), 6104-6109.

2. Chang, W.-S.; Wei, Y.; Guo, J.-M.; He, F.-J., Thermal Stability of Ni-Fe Alloy Foils Continuously Electrodeposited in a Fluoroborate Bath. *Open Journal of Metal* **2012**, *02* (01), 18-23.
3. Lv, H.; Lin, L.; Zhang, X.; Gao, D.; Song, Y.; Zhou, Y.; Liu, Q.; Wang, G.; Bao, X., In situ exsolved FeNi₃ nanoparticles on nickel doped Sr₂Fe_{1.5}Mo_{0.5}O_{6-δ} perovskite for efficient electrochemical CO₂ reduction reaction. *J. Mater. Chem. A* **2019**, *7* (19), 11967-11975.
4. Liu, S.; Liu, Q.; Luo, J.-L., Highly stable and efficient catalyst with in situ exsolved Fe–Ni alloy nanospheres socketed on an oxygen deficient perovskite for direct CO₂ electrolysis. *ACS Catal.* **2016**, *6* (9), 6219-6228.
5. O'hayre, R.; Cha, S.-W.; Colella, W.; Prinz, F. B., *Fuel cell fundamentals*. John Wiley & Sons: 2016.